# Unconstrained Stochastic CCA: Unifying Multiview and Self-Supervised Learning

**James Chapman**[*]**, Ana Lawry Aguila**
University College London
{james.chapman.19, ana.aguila.18}@ucl.ac.uk

**Lennie Wells**[*]
University of Cambridge
ww347@cam.ac.uk

## Abstract

The Canonical Correlation Analysis (CCA) family of methods is foundational in multiview learning. Regularised linear CCA methods can be seen to generalise Partial Least Squares (PLS) and be unified with a Generalized Eigenvalue Problem (GEP) framework. However, classical algorithms for these linear methods are computationally infeasible for large-scale data. Extensions to Deep CCA show great promise, but current training procedures are slow and complicated. First we propose a novel unconstrained objective that characterizes the top subspace of GEPs. Our core contribution is a family of fast algorithms for stochastic PLS, stochastic CCA, and Deep CCA, simply obtained by applying stochastic gradient descent (SGD) to the corresponding CCA objectives. Our algorithms show far faster convergence and recover higher correlations than the previous state-of-the-art on all standard CCA and Deep CCA benchmarks. These improvements allow us to perform a first-of-its-kind PLS analysis of an extremely large biomedical dataset from the UK Biobank, with over 33,000 individuals and 500,000 features. Finally, we apply our algorithms to match the performance of 'CCA-family' Self-Supervised Learning (SSL) methods on CIFAR-10 and CIFAR-100 with minimal hyper-parameter tuning, and also present theory to clarify the links between these methods and classical CCA, laying the groundwork for future insights.

## 1 Introduction

CCA methods learn highly correlated representations of multiview data. The original CCA method of Hotelling (1933) learns low-dimensional representations from linear transformations. Notable extensions to ridge-regularized CCA (Vinod, 1976), Partial Least Squares (PLS), and multiview CCA (Wong et al., 2021) allow one to use CCA in high dimensional regimes, and with three or more views of data. More recently, a variety of Deep CCA methods (Andrew et al., 2013) have been proposed which learn representations obtained from non-linear transformations of the data, parameterized by deep neural networks; Deep CCA has seen excellent empirical results, and is so foundational for deep multiview learning that it secured a runner-up position for the test-of-time award at ICML 2023 (ICML, 2023).

However, there are significant computational challenges when applying these CCA methods to large-scale data. Classical algorithms for linear CCA methods require computing full covariance matrices and so scale quadratically with dimension, becoming intractable for many datasets of practical interest. There is therefore great interest in approximating solutions for CCA in stochastic or data-streaming settings (Arora et al., 2012). Large-scale data also challenges existing full-batch algorithms for Deep CCA, and their stochastic counterparts are not only complex to implement but also difficult to train (Wang et al., 2015b).

Self-supervised learning (SSL) methods are now state-of-the-art in a range of domains, including image classification (Balestriero et al., 2023). They also learn useful representations of data, usually from pretext tasks or objectives that exploit some inherent structure or property of the data. Remarkably, SSL methods can even perform zero-shot classification: where the representations are learnt without any explicit labels or supervision. Of particular interest to us is the so-called CCA family of

---

[*]Equal contribution.

SSL methods (Balestriero et al., 2023). Like CCA, these methods aim to transform a pair of views of data to a pair of similar representations, and notably include Barlow twins (Zbontar et al., 2021) and VICReg (Bardes et al., 2021), which have become popular in light of empirical successes.

In section 2 we provide a unified approach to all the CCA methods introduced above, emphasizing objectives which are functions of the joint distributions of the transformed variables. Versions of all the linear CCA methods can be defined by solutions to certain Generalized Eigenvalue Problems (GEPs); this provides a particularly convenient way to relate a large number of equivalent objectives.

Section 3 outlines our core conceptual contributions. Firstly, with proposition 3.1 we present an unconstrained loss function that characterizes solutions to GEPs; this is based on the Eckhart–Young inequality and has appealing geometrical properties. We apply this to the GEP formulation of CCA and construct unbiased estimates of the loss and its gradients from mini-batches of data. These loss functions can therefore be optimized out-of-the-box using standard frameworks for deep learning. This immediately gives a unified family of algorithms for CCA, Deep CCA, and indeed SSL.

Our CCA algorithms dominate existing state-of-the-art methods across a wide range of benchmarks, presented in section 5. For stochastic CCA, our method not only converges faster but also achieves higher validation correlation scores than existing techniques. For Deep CCA and Deep Multiview CCA, our unbiased stochastic gradients yield significantly better validation correlations and allow the use of smaller mini-batches in memory constrained applications. We also demonstrate the practical utility of our algorithms with a pioneering real-world case study. We apply stochastic Partial Least Squares (PLS) to an extremely high-dimensional dataset from the UK Biobank – executing a biomedical analysis previously deemed intractable – all on a standard laptop.

Finally, our SSL method achieves comparable performance to VICReg and Barlow twins, despite having no hyperparameters in the objective. This frees computational resources to tune more critical hyperparameters, such as architecture, optimizer or augmentations. Our method also appears more robust to these other hyperparameters, has a clear theoretical foundation, and naturally generalizes to the multiview setting. In addition, we present theory in section 3.4 and appendix D which gives a more thorough description of how the existing SSL methods of Barlow twins and VICReg relate to CCA than in the previous work of Balestriero & LeCun (2022); we hope better understanding of these methods may lead to more principled empirical advances.

## 2 A UNIFIED APPROACH TO THE CCA FAMILY

Suppose we have a sequence of vector-valued random variables $X^{(i)} \in \mathbb{R}^{D^{(i)}}$ for $i \in \{1, \dots, I\}$[1]. We want to learn meaningful $K$-dimensional representations

$$Z^{(i)} = f^{(i)}(X^{(i)}; \theta^{(i)}). \tag{1}$$

For convenience, define $D = \sum_{i=1}^{I} D^{(i)}$ and $\theta = \left(\theta^{(i)}\right)_{i=1}^{I}$. We will consistently use the superscripts $i, j \in [I]$ for views and subscripts $l, k \in [K]$ for dimensions of representations - i.e. to subscript dimensions of $Z^{(i)}, f^{(i)}$. Later on, we will introduce total number of samples $N$ and mini-batch size $M$.

### 2.1 BACKGROUND: GEPs IN LINEAR ALGEBRA

A Generalized Eigenvalue Problem (GEP) is defined by two symmetric matrices $A, B \in \mathbb{R}^{D \times D}$ (Stewart & Sun, 1990)[2]. They are usually characterized by the set of solutions to the equation:

$$Au = \lambda Bu \tag{2}$$

with $\lambda \in \mathbb{R}, u \in \mathbb{R}^D$, called (generalized) eigenvalue and (generalized) eigenvector respectively. We shall only consider the case where $B$ is positive definite to avoid degeneracy. Then the GEP becomes equivalent to an eigen-decomposition of the symmetric matrix $B^{-1/2}AB^{-1/2}$. This is key to the proof of our new characterization. In addition, one can find a basis of eigenvectors spanning $\mathbb{R}^D$. We define a *top-K subspace* to be one spanned by some set of eigenvectors $u_1, \dots, u_K$ with

---

[1]A helpful mnemonic: there are $I$ (eye) views.

[2]More generally, $A, B$ can be Hermitian, but we are only interested in the real case.

the top-$K$ associated eigenvalues $\lambda_1 \geq \cdots \geq \lambda_K$. We say a matrix $U \in \mathbb{R}^{D \times K}$ *defines* a top-$K$ subspace if its columns span one.

## 2.2 THE CCA FAMILY

The classical notion CCA (Hotelling, 1992) considers two views, $I = 2$, and constrains the representations to be linear transformations

$$Z_k^{(i)} = \langle u_k^{(i)}, X^{(i)} \rangle. \tag{3}$$

The objective is to find the *weights* or *canonical directions* $u_k^{(i)}$ which maximize the *canonical correlations* $\rho_k = \text{Corr}(Z_k^{(1)}, Z_k^{(2)})$ sequentially, subject to orthogonality with the previous pairs of the transformed variables. It is well known that CCA is equivalent to a singular value decomposition (SVD) of the matrix $\text{Var}(X^{(1)})^{-1/2}\text{Cov}(X^{(1)}, X^{(2)})\text{Var}(X^{(2)})^{-1/2}$. It is slightly less well known (Borga, 1998) that this is equivalent to a GEP where:

$$A = \begin{pmatrix} 0 & \text{Cov}(X^{(1)}, X^{(2)}) \\ \text{Cov}(X^{(2)}, X^{(1)}) & 0 \end{pmatrix}, \quad B = \begin{pmatrix} \text{Var}(X^{(1)}) & 0 \\ 0 & \text{Var}(X^{(2)}) \end{pmatrix}, \quad u = \begin{pmatrix} u^{(1)} \\ u^{(2)} \end{pmatrix}. \tag{4}$$

CCA therefore has notions of uniqueness similar to those for SVD or GEPs: the weights are not in general unique, but the canonical correlations $1 \geq \rho_1 \geq \rho_2 \geq \cdots \geq 0$ are unique (Anderson, 2003). Therefore, we can write:

$$\text{CCA}_K(X^{(1)}, X^{(2)}) := (\rho_k)_{k=1}^K \tag{5}$$

**Sample CCA:** in practice we do not have access to the population distribution but to a finite number of samples; the classical estimator is defined by replacing the population covariances in eq. (4) with sample covariances. Unfortunately, this estimator breaks down when $N \leq \max(D^{(1)}, D^{(2)})$; giving arbitrary correlations of 1 and meaningless directions[3].

**Ridge-regularized CCA:** the most straightforward way to prevent this overfitting is to add a ridge regularizer (Vinod, 1976). Taking maximal ridge regularization recovers **Partial Least Squares PLS** (Mihalik et al., 2022), a widely used technique for multiview learning. Even these simple modifications to CCA can be very effective at preventing overfitting in high dimensions (Mihalik et al., 2022).

**multiview CCA (MCCA):** extends two-view CCA to deal with three or more views of data. Unfortunately, many of the different equivalent formulations of two-view CCA are no longer equivalent in the multiview setting, so there are many different extensions to choose from; see section 4. Of most interest to us is the formulation of Nielsen (2002); Wong et al. (2021) that extends the GEP formulation of eq. (4), which we next make precise and will simply refer to as MCCA from now.

**Unified GEP formulation:** this GEP formulation of MCCA can be presented in a unified framework generalizing CCA and ridge-regularized extensions. Indeed, we now take $A, B_\alpha \in \mathbb{R}^{D \times D}$ to be block matrices $A = (A^{(ij)})_{i,j=1}^I$, $B_\alpha = (B_\alpha^{(ij)})_{i,j=1}^I$ where the diagonal blocks of $A$ are zero, the off-diagonal blocks of $B_\alpha$ are zero, and the remaining blocks are defined by:

$$A^{(ij)} = \text{Cov}(X^{(i)}, X^{(j)}) \text{ for } i \neq j, \quad B_\alpha^{(ii)} = \alpha_i I_{D^{(i)}} + (1 - \alpha_i)\text{Var}(X^{(i)}) \tag{6}$$

Where $\alpha \in [0, 1]^I$ is a vector of ridge penalty parameters: taking $\alpha_i = 0 \, \forall i$ recovers CCA and $\alpha_i = 1 \, \forall i$ recovers PLS. We may omit the subscript $\alpha$ when $\alpha = 0$ and we recover the 'pure CCA' setting; in this case, following eq. (5) we can define $\text{MCCA}_K(X^{(1)}, \ldots, X^{(I)})$ to be the vector of the top-$K$ generalized eigenvalues.

**Deep CCA:** was originally introduced in Andrew et al. (2013); this was extended to an GEP-based formulation of **Deep multiview CCA** (DMCCA) in Somandepalli et al. (2019). This can be defined using our MCCA notation as maximizing

$$\|\text{MCCA}_K \left( Z^{(1)}, \ldots Z^{(I)} \right)\|_2 \tag{7}$$

over parameters $\theta$ of neural networks defining the representations $Z^{(i)} = f^{(i)}(X^{(i)}; \theta^{(i)})$ for $i \in [I]$.

---

[3] WLOG take $D^{(1)} \geq \max(N, D^{(2)})$. Then for any given observations $\mathbf{X}^{(1)} \in \mathbb{R}^{N \times K}, \mathbf{Z}_k^{(2)} \in \mathbb{R}^N$ there exists some $u_k^{(1)}$ such that $\mathbf{X}^{(1)\top} u_k^{(1)} = \mathbf{Z}_k^{(2)}$ - provided the observations of $\mathbf{X}^{(1)}$ are not linearly dependent - which is e.g. true with probability 1 when the observations are drawn from a continuous probability distribution.

## 3 NOVEL OBJECTIVES AND ALGORITHMS

### 3.1 UNCONSTRAINED OBJECTIVE FOR GEPS

First, we present proposition 3.1, a formulation of the top-$K$ subspace of GEP problems, which follows by applying the Eckhart–Young–Minsky inequality (Stewart & Sun, 1990) to the eigendecomposition of $B^{-1/2}AB^{-1/2}$. However, making this rigorous requires some technical care which we defer to the proof in supplement A.

**Proposition 3.1** (Eckhart–Young inspired objective for GEPs). *The top-$K$ subspace of the GEP* $(A, B)$ *can be characterized by minimizing the following objective over* $U \in \mathbb{R}^{D \times K}$:

$$\mathcal{L}_{\text{GEP-EY}}(U) := \text{trace}\left(-2\,U^\top A U + \left(U^\top B U\right)\left(U^\top B U\right)\right) \tag{8}$$

*Moreover, the minimum value is precisely* $-\sum_{k=1}^{K} \lambda_k^2$, *where* $(\lambda_k)$ *are the generalized eigenvalues.*

This objective also has appealing geometrical properties. It is closely related to a wide class of unconstrained objectives for PCA and matrix completion which have no spurious local optima (Ge et al., 2017), i.e. all local optima are in fact global optima. This implies that certain local search algorithms, such as stochastic gradient descent, should indeed converge to a global optimum.

**Proposition 3.2.** *The objective* $\mathcal{L}_{\text{GEP-EY}}$ *has no spurious local minima. That is, any matrix* $\bar{U}$ *that is a local minimum of* $\mathcal{L}_{\text{GEP-EY}}$ *must in fact be a global minimum.*

It is also possible to make this argument quantitative by proving a version of the strict saddle property from Ge et al. (2017; 2015); we state an informal version here and give full details in appendix B.

**Corollary 3.3** (Informal: Polynomial-time Optimization). *Under certain conditions on the eigenvalues and generalized eigenvalues of* $(A, B)$, *one can make quantitative the claim that: any* $U_K \in \mathbb{R}^{D \times K}$ *is either close to a global optimum, has a large gradient* $\nabla \mathcal{L}_{\text{GEP-EY}}$, *or has Hessian* $\nabla^2 \mathcal{L}_{\text{GEP-EY}}$ *with a large negative eigenvalue.*

*Therefore, for appropriate step-size sequences, certain local search algorithms, such as sufficiently noisy SGD, will converge in polynomial time with high probability.*

### 3.2 CORRESPONDING OBJECTIVES FOR THE CCA FAMILY

For the case of linear CCA we have $U^\top A U = \sum_{i \neq j} \text{Cov}(Z^{(i)}, Z^{(j)})$, $U^\top B U = \sum_i \text{Var}(Z^{(i)})$. To extend this to the general case of nonlinear transformations, eq. (1), we define the analogous matrices of total between-view covariance and total within-view variance

$$C(\theta) = \sum_{i \neq j} \text{Cov}(Z^{(i)}, Z^{(j)}), \quad V(\theta) = \sum_i \text{Var}(Z^{(i)}) \tag{9}$$

For linear transformations, eq. (3), it makes sense to add a ridge penalty so we can define

$$V_\alpha(\theta) = \sum_i \alpha_i U^{(i)\,\top} U^{(i)} + (1 - \alpha_i)\text{Var}(Z^{(i)}) \tag{10}$$

This leads to the following unconstrained objective for the CCA-family of problems.

**Definition 3.4** (Family of EY Objectives). Learn representations $Z^{(i)} = f^{(i)}(X^{(i)}; \theta^{(i)})$ minimizing

$$\mathcal{L}_{\text{EY}}(\theta) = -2\,\text{trace}\,C(\theta) + \|V_\alpha(\theta)\|_F^2 \tag{11}$$

**Unbiased estimates:** since empirical covariance matrices are unbiased, we can construct unbiased estimates to $C, V$ from a batch of transformed variables $\mathbf{Z}$.

$$\hat{C}(\theta)[\mathbf{Z}] = \sum_{i \neq j} \widehat{\text{Cov}}(\mathbf{Z}^{(i)}, \mathbf{Z}^{(j)}), \quad \hat{V}(\theta)[\mathbf{Z}] = \sum_i \widehat{\text{Var}}(\mathbf{Z}^{(i)}) \tag{12}$$

In the linear case we can construct $\hat{V}_\alpha(\theta)[\mathbf{Z}]$ analogously by plugging sample covariances into eq. (10). Then if $\mathbf{Z}, \mathbf{Z}'$ are two independent batches of transformed variables, the batch loss

$$\hat{\mathcal{L}}_{\text{EY}}[\mathbf{Z}, \mathbf{Z}'] := -2\,\text{trace}\,\hat{C}[\mathbf{Z}] + \langle \hat{V}_\alpha[\mathbf{Z}], \hat{V}_\alpha[\mathbf{Z}'] \rangle_F \tag{13}$$

gives an unbiased estimate of $\mathcal{L}_{\text{EY}}(\theta)$. This loss is a differentiable function of $\mathbf{Z}, \mathbf{Z}'$ and so also of $\theta$.

**Simple algorithms:** We first define a general algorithm using these estimates in Algorithm 1. In the next section we apply this algorithm to multiview stochastic CCA (**CCA-EY**) and PLS (**PLS-EY**), Deep CCA (**DCCA-EY**), and SSL (**SSL-EY**).

**Algorithm 1: GEP-EY**: General algorithm for learning correlated representations

---

**Input:** data stream of mini-batches $(\mathbf{X}(b))_{b=1}^{\infty}$ where each consists of $M$ samples from the original dataset. Learning rate $(\eta_t)_t$. Number of time steps $T$. Class of functions $f(\cdot; \theta)$ whose outputs are differentiable with respect to $\theta$.

**Initialize:** $\hat{\theta}$ with suitably random entries

**for** $t = 1$ **to** $T$ **do**
    Obtain two independent mini-batches $\mathbf{X}(b), \mathbf{X}(b')$ by sampling $b, b'$ independently
    Compute batches of transformed variables $\mathbf{Z}(b) = f(\mathbf{X}(b); \theta), \mathbf{Z}(b') = f(\mathbf{X}(b'); \theta)$
    Estimate loss $\hat{\mathcal{L}}_{\text{EY}}(\theta)$ using eq. (13)
    Obtain gradients by back-propagation and step with your favourite optimizer.
**end for**

---

### 3.3 APPLICATIONS TO MULTIVIEW STOCHASTIC CCA AND PLS, AND DEEP CCA

**Lemma 3.5** (Objective recovers GEP formulation of linear multiview CCA). *When the $f^{(i)}$ are linear, as in eq. (3), the population loss from eq. (11) recovers MCCA as defined in section 2.2.*

*Proof.* By construction, for linear MCCA we have $C = U^{\top}AU$, $V_{\alpha} = U^{\top}B_{\alpha}U$, where $(A, B_{\alpha})$ define the GEP for MCCA introduced in eq. (6). So $\mathcal{L}_{\text{EY}}(U) = \mathcal{L}_{\text{GEP-EY}}(U)$ and by proposition 3.1 the optimal set of weights define a top-$K$ subspace of the GEP, and so is a MCCA solution. $\square$

Moreover, by following through the chain of back-propagation, we obtain gradient estimates in $\mathcal{O}(MKD)$ time. Indeed, we can obtain gradients for the transformed variables in $\mathcal{O}(MK^2)$ time so the dominant cost is then updating $U$; we flesh this out with full details in appendix E.

**Lemma 3.6.** *[Objective recovers Deep multiview CCA] Assume that there is a final linear layer in each neural network $f^{(i)}$. Then at any local optimum, $\hat{\theta}$, of the population problem, we have*

$$\mathcal{L}_{EY}(\hat{\theta}) = -\|\text{MCCA}_K(\hat{Z})\|_2^2$$

*where $\hat{Z} = f_{\hat{\theta}}(X)$. Therefore, $\hat{\theta}$ is also a local optimum of objectives from Andrew et al. (2013); Somandepalli et al. (2019) as defined in eq. (7).*

*Proof sketch: see appendix C.2.1 for full details.* Consider treating the penultimate-layer representations as fixed, and optimising over the weights in the final layer. This is precisely equivalent to optimising the Eckhart-Young loss for linear CCA where the input variables are the penultimate-layer representations. So by proposition 3.2, a local optimum is also a global optimum, and by proposition 3.1 the optimal value is the negative sum of squared generalised eigenvalues. $\square$

### 3.4 APPLICATION TO SSL

We can directly apply Algorithm 1 to SSL. If we wish to have the same neural network transforming each view, we can simply tie the weights $\theta^{(1)} = \theta^{(2)}$. When the paired data are generated from applying independent, identically distributed (i.i.d.) augmentations to the same original datum, it is intuitive that tying the weights is a sensible procedure, and perhaps acts as a regulariser. We make certain notions of this intuition precise for CCA and Deep CCA in appendix C.

To provide context for this proposal, we also explored in detail how VICReg and Barlow twins are related to CCA. For now we focus on VICReg, whose loss can be written as

$$\mathcal{L}_{\text{VR}}(Z^{(1)}, Z^{(2)}) = \gamma\mathbb{E}\|Z^{(1)} - Z^{(2)}\|^2 + \sum_{i \in \{1,2\}} \left[ \alpha \sum_{k=1}^{K} \left(1 - \sqrt{\text{Var}(Z_i^{(i)})}\right)_+ + \beta \sum_{\substack{k,l=1 \\ k \neq l}}^{K} \text{Cov}(Z_k^{(i)}, Z_l^{(i)})^2 \right]$$

where $\alpha, \beta, \gamma > 0$ are tuning parameters and, as in the framework of section 2, the $Z^{(1)}, Z^{(2)}$ are $K$-dimensional representations, parameterised by neural networks in eq. (1). The heuristic behind this loss is that the $\gamma$-term encourages the pair of representations to be similar (Invariance), the $\beta$-term encourages different components of the representations to be uncorrelated (Covariance), and

the $\alpha$-term enforces strictly positive variance of each component (Variance). Our main conclusions regarding optima of the population loss are as follows.

- Consider the linear setting with untied weights. Then global optimisers of the VICReg loss define CCA subspaces, but may not be of full rank.
- Consider the linear setting with tied weights and additionally assume that the data are generated by i.i.d. augmentations. Then the same conclusion holds.
- In either of these settings, the optimal VICReg loss is a component-wise decreasing function of the vector of population canonical correlations $\mathrm{CCA}_K(X^{(1)}, X^{(2)})$.
- VICReg can therefore be interpreted as a formulation of Deep CCA, but one that will not in general recover full rank representations.

We give full mathematical details and further discussion in appendix D. The population loss for Barlow twins is motivated by similar heuristics, encouraging uncorrelated components and similarity between views, but is naturally viewed as explicitly constraining each component to have unit variance one. This makes the analysis for Barlow twins more difficult, but we present a combination of mathematical and empirical arguments which suggest all the same conclusions hold as for VICReg, again see appendix D for full details.

## 4 RELATED WORK

**Stochastic PLS and CCA:** To the best of our knowledge, the state-of-the-art in Stochastic PLS and CCA are the subspace Generalized Hebbian Algorithm (**SGHA**) of Chen et al. (2019) and $\gamma$-**EigenGame** from Gemp et al. (2020; 2021), which we use as benchmarks in the following section. Specifically, SGHA utilizes a Lagrange multiplier heuristic along with saddle-point analysis, albeit with limited convergence guarantees. EigenGame focuses on top-$K$ subspace learning but introduces an adaptive whitening matrix in the stochastic setting with an additional hyperparameter. Like our method, both can tackle other symmetric Generalized Eigenvalue Problems in principle.

**DCCA and Deep Multiview CCA:** The deep canonical correlation analysis (DCCA) landscape comprises three principal approaches with inherent limitations. The first, known as the full-batch approach, uses analytic gradient derivations based on the full sample covariance matrix (Andrew et al., 2013). The second involves applying the full batch objective to large mini-batches, an approach referred to as **DCCA-STOL** (Wang et al., 2015a). However, this approach gives biased gradients and therefore requires batch sizes much larger than the representation size in practice. This is the approach taken by both **DMCCA** (Somandepalli et al., 2019) and **DGCCA** (Benton et al., 2017) . The final set of approaches use an adaptive whitening matrix (Wang et al., 2015b; Chang et al., 2018) to mitigate the bias of the Deep CCA objective. However, the authors of **DCCA-NOI** highlight that the associated time constant complicates analysis and requires extensive tuning. These limitations make existing DCCA methods less practical and resource-efficient.

**Self-Supervised Learning:** We are most interested in comparisons to **Barlow twins** and **VICReg** because of their empirical success, and known relationship to CCA; we believe the most comprehensive existing theoretical analysis of this relationship is in Balestriero & LeCun (2022), however this only applies to a subset of possible VICReg parameters, appears incomplete for Barlow twins, and does not include discussion of the rank of representations. For a wider perspective on SSL methods and their applications we recommend the review of (Balestriero et al., 2023), and the efficient implementations available in solo-learn Da Costa et al. (2022).

## 5 EXPERIMENTS

### 5.1 EVALUATING CCA-EY FOR STOCHASTIC CCA

First, we compare our proposed method, CCA-EY, to the baselines of $\gamma$-EigenGame and SGHA. Our experimental setup is almost identical to that of Meng et al. (2021); Gemp et al. (2022). Unlike Gemp et al. (2022), we do not simplify the problem by first performing PCA on the data before applying the CCA methods, which explains the decrease in performance of $\gamma$-EigenGame compared to their work. All models are trained for a single epoch with varying mini-batch sizes ranging from

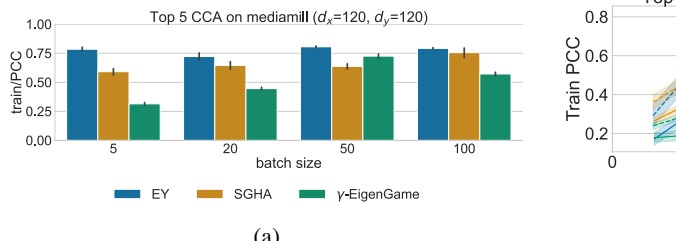
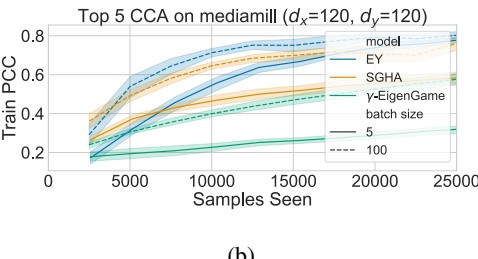

(a)                                                    (b)

Figure 1: Stochastic CCA on MediaMill using the Proportion of Correlation Captured (PCC) metric: (a) Across varying mini-batch sizes, trained for a single epoch, and (b) Training progress over a single epoch for mini-batch sizes 5, 100. Shaded regions signify $\pm$ one standard deviation around the mean of 5 runs.

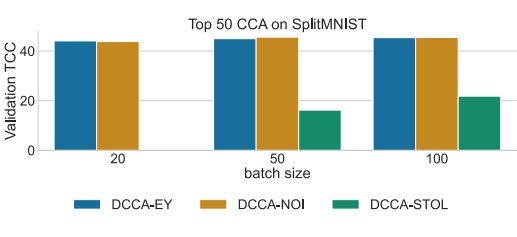
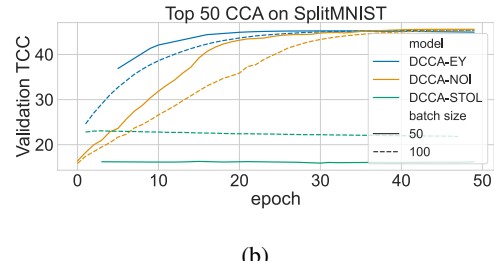

(a)                                                    (b)

Figure 2: Deep CCA on SplitMNIST using the Validation TCC metric: (a) after training each model for 50 epochs with varying batch sizes; (b) learning progress over 50 epochs.

5 to 100. We use Proportion of Correlation Captured (PCC) as our evaluation metric, defined as $\text{PCC} = (\sum_{i=k}^{K} \rho_k)/(\sum_{k=1}^{K} \rho_k^*)$ where $\rho_k$ are the full batch correlations of the learnt representations, and $\rho_k^*$ are the canonical correlations computed numerically from the full batch covariance matrices.

**Observations:** Figure 1 compares the algorithms on the MediaMill dataset. fig. 1a shows that CCA-EY consistently outperforms both $\gamma$-EigenGame and SGHA in terms of PCC across all evaluated mini-batch sizes. fig. 1b examines the learning curves for batch sizes 5 and 100 in more detail; CCA-EY appears to learn more slowly than SGHA at the start of the epoch, but clearly outperforms SGHA as the number of samples seen increases. $\gamma$-EigenGame significantly underperforms SGHA and CCA-EY, particularly for small batch sizes.

**Further experiments:** we conduct analogous experiments on the Split CIFAR dataset in supplementary material F and observe identical behaviour.

### 5.2   EVALUATING DCCA-EY FOR DEEP CCA

Second, we compare DCCA-EY against the DCCA methods described in section 4. The experimental setup is identical to that of Wang et al. (2015b). We learn $K = 50$ dimensional representations, using mini-batch sizes ranging from 20 to 100 and train for 50 epochs. Because there is no longer a ground truth, we have to use Total Correlation Captured (TCC), given by $\text{TCC} = \sum_{i=k}^{K} \rho_k$ where $\rho_k$ are now the empirical correlations between the representations on a validation set.

**Observations:** Figure 2 compares the methods on the splitMNIST dataset. DCCA-STOL captures significantly less correlation than the other methods, and breaks down when the mini-batch size is less than the dimension $K = 50$ due to low rank empirical covariances. DCCA-NOI performs similarly to DCCA-EY but requires careful tuning of an additional hyperparameter, and shows significantly slower speed to convergence (Figure 2b).

**Further experiments:** we conduct analogous experiments on the XRMB dataset in supplementary material G and observe identical behaviour.

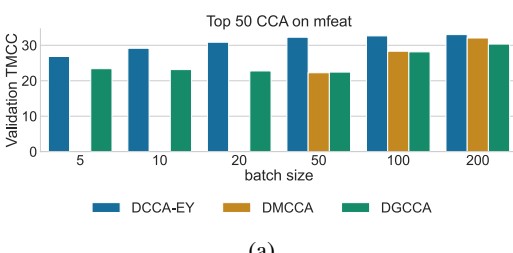
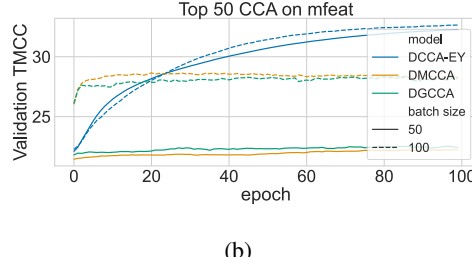

(a)                                                              (b)

Figure 3: Deep multiview CCA on mfeat using the Validation TMCC metric: (a) after training each model for 100 epochs with varying batch sizes; (b) learning progress over 100 epochs.

### 5.3 EXTENDING DCCA-EY TO THE MULTIVIEW SETTING

Third, we compare DCCA-EY to the existing DMCCA and DGCCA methods on the mfeat dataset; this contains 2,000 handwritten numeral patterns across six distinct feature sets, including Fourier coefficients, profile correlations, Karhunen-Love coefficients, pixel averages in $2 \times 3$ windows, Zernike moments, and morphological features. We again learn $K = 50$ dimensional representations, but now train for 100 epochs. We use a multiview extension of the TCC metric, which averages correlation across views; we call this Total Multiview Correlation Captured (TMCC), defined as TMCC $= \sum_{k=1}^{K} \frac{1}{I(I-1)} \sum_{i,j \leq I, i \neq j} \text{corr}(Z_k^{(i)}, Z_k^{(j)})$, using the notation of section 2.

**Observations:** Figure 3a shows that DCCA-EY consistently outperforms both DGCCA and DMCCA across various mini-batch sizes in capturing validation TMCC. Just like DCCA-NOI, DMCCA breaks down when the batch size is smaller than $K$. This is due to singular empirical covariances; DGCCA does not break down, but does significantly underperform with smaller batch sizes. This limits their practical applicability to large-scale data. Figure 3b shows learning curves for batch sizes 50 and 100. DMCCA and DGCCA both quickly learn significant correlations but then plateau out; our method consistently improves, and significantly outperforms them by the end of training.

### 5.4 SCALING PLS TO THE UK BIOBANK WITH PLS-EY

Next, we demonstrate the scalability of our methods to extremely high-dimensional data by applying stochastic PLS to imaging genetics data from the UK Biobank (Sudlow et al., 2015). PLS is typically used for imaging-genetics studies owing to the extremely high dimensionality of genetics data requiring lots of regularisation. PLS can reveal novel phenotypes of interest and uncover relationships between genetic mechanisms of disease and brain morphometry. Previous imaging genetics analyses using full-batch PLS were limited to much smaller datasets (Lorenzi et al., 2018; Taquet et al., 2021; Édith Le Floch et al., 2012). The only other analysis on the UK Biobank at comparable scale partitions the data into clusters and bootstraps local PLS solutions on these clusters (Lorenzi et al., 2017; Altmann et al., 2023). We ran PLS-EY with mini-batch size 500 on brain imaging (82 regional volumes) and genetics (582,565 variants) data for 33,333 subjects. See supplement (Section I.3.4) for data pre-processing details. To our knowledge, this is the largest-scale PLS analysis of biomedical data to-date.

**Observations:** We see strong validation correlation between all 10 corresponding pairs of vectors in the PLS subspace and weak cross correlation, indicating that our model learnt a coherent and orthogonal subspace of covariation (Figure 4a), a remarkable feat for such high-dimensional data. We found that the PLS brain subspace was associated with genetic risk measures for several disorders (Figure 4b), suggesting that the PLS subspace encodes relevant information for genetic disease risk, a significant finding for biomedical research.

### 5.5 APPLYING SSL-EY FOR PRINCIPLED SELF-SUPERVISED LEARNING

Finally, we benchmark our self-supervised learning algorithm, SSL-EY, with Barlow twins and VICReg on CIFAR-10 and CIFAR-100. Each dataset contains 60,000 labelled images, but these are over 10 classes for CIFAR-10 and 100 classes for CIFAR-100.

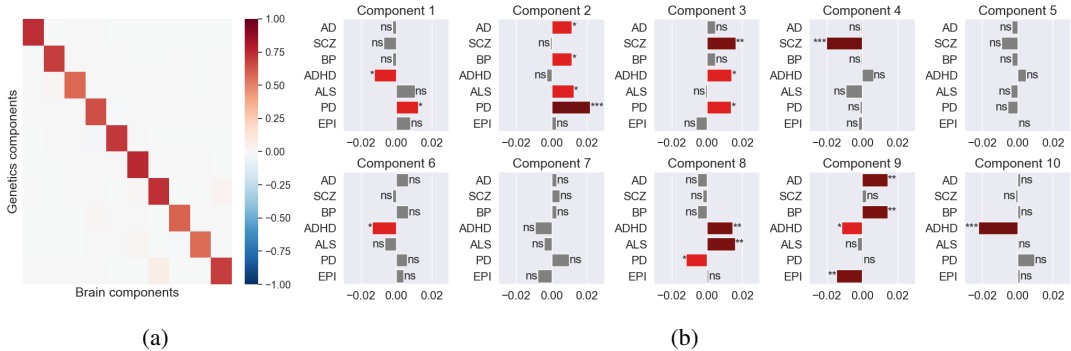

(a)                                                                    (b)

Figure 4: (a) Correlations between PLS components for UK Biobank. (b) Correlations between PLS brain components and genetic risk scores. AD=Alzheimer's disease, SCZ=Schizophrenia, BP=Bipolar, ADHD=Attention deficit hyperactivity disorder, ALS=Amyotrophic lateral sclerosis, PD=Parkinson's disease, EPI=Epilepsy. ns : $0.05 < p <= 1, * : 0.01 < p <= 0.05, ** : 0.001 < p <= 0.01, *** : 0.0001 < p <= 0.001$.

| Method | CIFAR-10 Top-1 | CIFAR-10 Top-5 | CIFAR-100 Top-1 | CIFAR-100 Top-5 |
|---|---|---|---|---|
| Barlow twins | **92.1** | 99.73 | **71.38** | **92.32** |
| VICReg | 91.68 | 99.66 | 68.56 | 90.76 |
| **SSL-EY** | 91.43 | **99.75** | 67.52 | 90.17 |

Table 1: Performance comparison of SSL methods on CIFAR-10 and CIFAR-100.

We follow a standard experimental design (Tong et al., 2023), and use solo-learn (Da Costa et al., 2022), which offers optimized setups particularly tailored for VICReg and Barlow twins. All methods utilize a ResNet-18 encoder coupled with a bi-layer projector network. Training spans 1,000 epochs with batches of 256 images. For SSL-EY, we use the hyperparameters optimized for Barlow twins, aiming not to outperform but to showcase the robustness of our method. We predict labels via a linear probe on the learnt representations and evaluate performance with Top-1 and Top-5 accuracies on the validation set. For more details, refer to the supplementary material I.3.

**Observations:** Table 1 shows that SSL-EY is competitive with Barlow twins and VICReg. This is remarkable because we used out-of-the-box hyperparameters for SSL-EY but used hyperparameters for Barlow twins and VICReg that had been heavily optimized in previous studies.

**Further experiments** included in appendix H show that the learning curves for all three methods are comparable, and that our method is much more stable when reducing the dimension of the learnt representations.

## 6 CONCLUSION

In this paper, we introduced a class of efficient, scalable algorithms for Canonical Correlation Analysis and Self-Supervised Learning, rooted in a novel unconstrained loss function. These algorithms are computationally lightweight, making them uniquely suited for large-scale problems where traditional methods struggle.

We have two distinct avenues for future research. Firstly, we aim to incorporate regularization techniques to improve both generalizability and interpretability, building upon existing sparse methods in CCA (Witten & Tibshirani, 2009). We also intend to investigate the utility of correlation as a metric for measuring the quality of learnt representations. This holds the potential to replace traditional validation methods like classification accuracy, especially in situations where validation labels are not available.

In summary, this paper sets a new benchmark for addressing large-scale CCA problems and opens new avenues in self-supervised learning, paving the way for more accessible and efficient solutions in various applications.

## AUTHOR CONTRIBUTIONS STATEMENT

JC conceived the original idea of developing new stochastic algorithms based on the GEP formulation CCA using the GEP formulation, and developed a number of methods that appeared to perform well empirically. LW proposed the core Eckhart–Young GEP framework of this paper in response, and proved all mathematical results. JC suggested the extension to SSL, and wrote all the code for the experiments. LW and JC wrote the paper collaboratively, focusing on theoretical and empirical contributions respectively. ALA performed the analysis on the UK Biobank dataset, wrote the corresponding commentary, and edited the main text.

## REPRODUCIBILITY STATEMENT

To ensure the reproducibility of our work, we have made the following resources publicly available:

- Code for stochastic CCA and deep CCA experiments: `https://github.com/jameschapman19/GEP-EY`.
- For Self-Supervised Learning experiments, our modified version of solo-learn is accessible at: `https://github.com/jameschapman19/solo-learn`.
- Standalone PyTorch implementation of the SSL-EY loss function: `https://github.com/jameschapman19/SSL-EY`.

## ACKNOWLEDGMENTS

The authors thank Sergio Bacallado for valuable advice on how to restructure an earlier version of this paper, and on various details of the theoretical contributions. LW is supported by the UK Engineering and Physical Sciences Research Council (EPSRC) under grant number EP/V52024X/1. JC and ALA are supported by the EPSRC-funded UCL Centre for Doctoral Training in Intelligent, Integrated Imaging in Healthcare (i4health) and the Department of Health's NIHR-funded Biomedical Research Centre at University College London Hospitals.

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

## A  ECKHART-YOUNG CHARACTERIZATION OF GEP SUBSPACE

Our characterisation of the top-$K$ subspace of GEPs with the GEP-EY loss is given in appendix A.4; the key to the proof is the algebraic manipulation in eq. (16), which reduces our GEP-EY loss to the form of the loss that appears in the Eckhart-Young inequality. However, we will need a non-standard formulation of the Eckhart-Young inequality to apply to this term; we state this as corollary A.10, and build machinery to prove it over the following two subsections.

### A.1  FORMAL DEFINITIONS

There are various different notations and conventions for GEPs and SVDs. We largely follow the standard texts on Matrix Analysis (Stewart & Sun, 1990; Bhatia, 1997) but seek a more careful handling of the equality cases of certain results. To help, we use the following non-standard definitions, largely inspired by Carlsson (2021).

**Definition A.1** (Top-$K$ subspace). Let the GEP $(A, B)$ on $\mathbb{R}^D$ have eigenvalues $\lambda_1 \geq \cdots \geq \lambda_D$. Then **a** top-$K$ subspace is that spanned by some $u_1, \ldots, u_K$, where $u_k$ is a $\lambda_k$-eigenvector of $(A, B)$ for $k = 1, \ldots, K$.

**Definition A.2** ($B$-orthonormality). Let $B \in \mathbb{R}^{D \times D}$ be strictly positive definite. Then we say a collection $u_1, \ldots, u_K \in \mathbb{R}^D$ of vectors is $B$-orthonormal if $u_k^\top B u_l = \delta_{kl}$ for each $k, l \in \{1, \ldots, K\}$.

**Definition A.3** (Top-$K$ matrix). We say $U \in \mathbb{R}^{D \times K}$ is a top-$K$ matrix for a GEP $(A, B)$ if the $k^{\text{th}}$ column $u_k$ of $U$ is a $\lambda_k$-eigenvector for each $k$ and the columns are $B$-orthonormal.

### A.2  STANDARD ECKHART–YOUNG INEQUALITY

**Theorem A.4** (Eckhart–Young). *Let $M \in \mathbb{R}^{p \times q}$. Then $\hat{M}$ minimises $\|M - \tilde{M}\|_F$ over matrices $\tilde{M}$ of rank at most $K$ if and only if $\hat{M} = A_K R_K B_K^\top$ where $(A_K, R_K, B_K)$ is some top-$K$ SVD of the target $M$.*

*Proof.* Let $M, \tilde{M}$ have singular values $\sigma_k, \tilde{\sigma}_k$ respectively. Since $\tilde{M}$ has rank at most $K$ we must have $\tilde{\sigma}_k = 0$ for $k > K$.

Then by von Neumann's trace inequality (Carlsson, 2021),

$$\langle M, \tilde{M} \rangle_F \leq \sum_{k=1}^K \sigma_k \tilde{\sigma}_k$$

with equality if and only if $M, \tilde{M}$ 'share singular vectors'; the notion of sharing singular vectors is defined as in Carlsson (2021) and in this case means that $\tilde{M} = A_K \tilde{R}_K B_K$ where $(A_K, R_K, B_K)$ is some top-$K$ SVD of $M$ and $\tilde{R}_K$ is a diagonal matrix with decreasing diagonal elements $\tilde{\sigma}_1 \geq \cdots \geq \tilde{\sigma}_K$.

Expanding out the objective and applying this inequality gives

$$\|\tilde{M} - M\|_F^2 \geq \sum_{k=1}^D \sigma_k^2 - 2\sum_{k=1}^K \sigma_k \tilde{\sigma}_k + \sum_{k=1}^K \tilde{\sigma}_k^2$$

$$= \sum_{k=K+1}^D \sigma_k^2 + \sum_{k=1}^K (\sigma_k - \tilde{\sigma}_k)^2$$

$$\geq \sum_{k=K+1}^D \sigma_k^2$$

so indeed to have equality in both cases requires $\sigma_k = \tilde{\sigma}_k$ for each $k \leq K$ so indeed $\tilde{R}_K = R_K$ and so $\hat{M}$, as defined in the statement of the theorem, minimises $\|M - \tilde{M}\|_F$ over matrices $\tilde{M}$ of rank at most $K$. $\qquad\square$

## A.3 Eckhart–Young for factored estimator of symmetric target

**Lemma A.5** (Matrix square root lemma). *Suppose we have two full rank matrices $E, F \in \mathbb{R}^{D \times K}$ where $K \leq D$ and such that $EE^\top = FF^\top$; then there exists an orthogonal matrix $O \in \mathbb{R}^{K \times K}$ with $E = FO$.*

*Proof.* Post multiplying the defining condition gives $EE^\top E = FF^\top E$. Then right multiplying by $(E^\top E)^{-1}$ gives

$$E = FF^\top E(E^\top E)^{-1} =: FO$$

to check that $O$ as defined above is orthogonal we again use the defining condition to compute

$$O^\top O = (E^\top E)^{-1}E^\top FF^\top E(E^\top E)^{-1} = (E^\top E)^{-1}E^\top EE^\top E(E^\top E)^{-1} = I_K$$

$\square$

**Remark A.6** (Tilde convention). In the rest of this section, the tildes above the quantity $\tilde{W}$ below is to indicate that said matrix may not have orthonormal columns, whereas quantities without tildes ($W_K, W_+, W_-$) implicitly do have orthonormal columns. A similar convention applies in the following subsection with the matrices $\tilde{U}, U_K$.

**Corollary A.7** (Eckhart–Young for factored estimator of PSD target). *Let $M \in \mathbb{R}^{D \times D}$ be symmetric positive semidefinite. Then*

$$\underset{\tilde{W} \in \mathbb{R}^{D \times K}}{\arg \min} \|M - \tilde{W}\tilde{W}^\top\|_F^2$$

*is precisely the set of $\tilde{W}$ of the form $\tilde{W} = W_K \Lambda_K^{1/2} O_K$ for some top-$K$ eigenvector-matrix $W_K$ of the GEP $(M, I)$ and some orthogonal $O_K \in \mathcal{O}(K)$, and where $\Lambda_K$ is a diagonal matrix of the top-$K$ eigenvalues.*

*Proof.* First note that when $M$ is positive semi-definite the SVD coincides with the eigendecomposition.

Second note that taking $\tilde{W} = W_K \Lambda_K^{1/2} O_K$ attains the minimal value by the Eckhart–Young inequality, Theorem A.4.

Next note that if $\tilde{W}$ attains the minimal value then it must have $\tilde{W}\tilde{W}^\top = W_K \Lambda_K W_K^\top$ by the equality case of Eckhart–Young. Then by matrix square root Lemma A.5 we must indeed have $\tilde{W} = W_K \Lambda_K^{1/2} O_K$ for some orthogonal $O_K$. $\square$

To convert this to the case of a general symmetric target (with possible negative eigenvalues), we use the following decomposition of a matrix into a sum of matrices corresponding to its positive and negative eigenvalues respectively. We will also find the following terminology helpful.

**Definition A.8** (Non-negative eigenspace). Let $M \in \mathbb{R}^{D \times D}$ be a symmetric matrix. Then the *non-negative eigenspace* of $M$ is defined as the span of the eigenvectors of $M$ with non-negative eigenvalues.

**Remark A.9** (Decomposition of matrix into positive and negative eigenspaces). Let $M \in \mathbb{R}^{D \times D}$ be a symmetric matrix. Write $\Lambda_+$ for the diagonal matrix containing the strictly positive eigenvalues of $M$ and $\Lambda_-$ for the strictly negative eigenvalues; let the corresponding eigenvectors be arranged in the matrices $W_+, W_-$ respectively. Define $M_+ := W_+ \Lambda_+ W_+^\top$, $M_- := W_- \Lambda_- W_-^\top$. Then $M_+$ is positive semi-definite, $M_-$ is negative semi-definite, $M = M_+ + M_-$ and $\langle M_+, M_- \rangle_F = 0$ by the orthogonality of eigenvectors of $M$.

**Corollary A.10** (Eckhart–Young for factored estimator of symmetric target). *Let $M \in \mathbb{R}^{D \times D}$ be symmetric with eigenvalues $\lambda_1 \geq \cdots \geq \lambda_D$ such that $\lambda_K > 0$. Then*

$$\underset{\tilde{W} \in \mathbb{R}^{D \times K}}{\arg \min} \|M - \tilde{W}\tilde{W}^\top\|_F^2$$

*is precisely the set of $\tilde{W}$ of the form $\tilde{W} = W_K \Lambda_K^{1/2} O_K$ for some top-$K$ eigenvector-matrix $W_K$ of the GEP $(M, I)$ and some orthogonal $O_K \in \mathcal{O}(K)$, and where $\Lambda_K$ is a diagonal matrix of the top-$K$ eigenvalues.*

*Proof.* Let $\tilde{W} \in \mathbb{R}^{D \times K}$. Write $M = M_+ + M_-$ as in remark A.9, and suppose there are $D_+$ strictly positive and $D_-$ strictly negative eigenvalues of $M$ respectively. Then we can expand out

$$\|M - \tilde{W}\tilde{W}^\top\|^2 = \underbrace{\|M_+ - \tilde{W}\tilde{W}^\top\|^2}_{\geq \sum_{k=K+1}^{D_+} \lambda_k^2} + 2\underbrace{\langle -M_-, \tilde{W}\tilde{W}\rangle_F}_{\geq 0} + 2\underbrace{\langle M_-, M_+\rangle_F}_{0} + \underbrace{\|M_-\|^2}_{\sum_{k=D-D_-+1}^{D} \lambda_k^2} , \tag{14}$$

where the 'underbraced' (in)equalities below the equation follow from: applying corollary A.7 to $M_+$, using that $-M_-$ is positive semi-definite, the orthogonality of $M_-$ to $M_+$, and the definition of $M_-$ in terms of the eigenvalues of $M$, respectively. Combining the terms gives

$$\|M - \tilde{W}\tilde{W}^\top\|^2 \geq \sum_{k=K+1}^{D} \lambda_k^2 , \tag{15}$$

as claimed.

Moreover, from the equality case of corollary A.7, equality holds in the first inequality precisely when $\tilde{W} = W_K \Lambda_K^{1/2} O_K$, and for such a $\tilde{W}$, $\langle M_-, \tilde{W}\tilde{W}\rangle_F = 0$ in fact there is also equality in the second inequality and therefore in the whole expression, as required. $\square$

## A.4 GEP-EY Objective

**Proposition A.11** (GEP-EY-Objective). *Consider the GEP $(A, B)$ with $A$ symmetric and $B$ positive definite; suppose there are at least $K$ strictly positive (generalized) eigenvalues. Then:*

$$\underset{\tilde{U} \in \mathbb{R}^{D \times k}}{\arg\max} \ \operatorname{trace}\left\{ 2\left(\tilde{U}^\top A \tilde{U}\right) - \left(\tilde{U}^\top B \tilde{U}\right)\left(\tilde{U}^\top B \tilde{U}\right) \right\}$$

*is precisely the set of $\tilde{U}$ of the form $\tilde{U} = U_K \Lambda_K^{1/2} O_K$ for some top-$K$ matrix $U_K$ of the GEP and some orthogonal $O_K \in \mathcal{O}(K)$, where $\Lambda_K$ is a diagonal matrix of the top-$K$ eigenvalues.*

*Moreover, the maximum value is precisely $-\sum_{k=1}^{K} \lambda_k^2$.*

*Proof.* First note that there is a bijection between eigenvectors $u$ for the GEP $(A, B)$ and eigenvectors $w = B^{1/2}u$ for the GEP $(M, I)$ where $M := B^{-1/2}AB^{-1/2}$ (e.g. see Chapman et al. (2022)).

Now consider how the Eckhart–Young objective from Corollary A.10 transforms under the corresponding bijection $W = B^{1/2}U$.

We get

$$\begin{aligned}
\|M - \tilde{W}\tilde{W}\|_F^2 &= \|B^{-1/2}AB^{-1/2} - B^{1/2}\tilde{U}\tilde{U}^\top B^{1/2}\|_F^2 \\
&= \|B^{-1/2}AB^{-1/2}\|_F^2 - 2\operatorname{trace}\left(B^{-1/2}AB^{-1/2}B^{1/2}\tilde{U}\tilde{U}^\top B^{1/2}\right) \\
&\quad + \operatorname{trace}\left(B^{1/2}\tilde{U}\tilde{U}^\top B^{1/2}\ B^{1/2}\tilde{U}\tilde{U}^\top B^{1/2}\right) \\
&= \|B^{-1/2}AB^{-1/2}\|_F^2 - \operatorname{trace}\left\{ 2\left(\tilde{U}^\top A\tilde{U}\right) - \left(\tilde{U}^\top B\tilde{U}\right)\left(\tilde{U}^\top B\tilde{U}\right) \right\},
\end{aligned} \tag{16}$$

where the first term is independent of $\tilde{U}$, so we can conclude by Corollary A.10.

The moreover conclusion can follow from computing the objective at any maximiser of the form above. We note that

$$\tilde{U}^\top A \tilde{U} = O_K^\top \Lambda_K^{1/2} U_K^\top A U_K \Lambda_K O_K = O_K^\top \Lambda_K^2 O_K$$
$$\tilde{U}^\top B \tilde{U} = O_K^\top \Lambda_K^{1/2} U_K^\top B U_K \Lambda_K O_K = O_K^\top \Lambda_K O_K$$

plugging into the objective gives

$$\operatorname{trace}\left(2\left(\tilde{U}^\top A \tilde{U}\right) - \left(\tilde{U}^\top B \tilde{U}\right)^2\right) = \operatorname{trace}\left(2 O_K^\top \Lambda_K^2 O_K - O_K^\top \Lambda_K^2 O_K\right) = \sum_{k=1}^{K} \lambda_k^2$$

because the trace of a symmetric matrix is equal to the sum of its eigenvalues. $\square$

# B    TRACTABLE OPTIMIZATION - NO SPURIOUS LOCAL MINIMA

First in appendix B.1 we prove that for general $A, B$ our loss $\mathcal{L}_{\text{GEP-EY}}(U)$ has no spurious local minima. Then in appendix B.3 we apply a result from Ge et al. (2017). This application is somewhat crude, and we expect that a quantitative result with tighter constants could be obtained by adapting the argument of appendix B.1; we leave such analysis to future work.

## B.1    QUALITATIVE RESULTS

First we present our main result and its proof. This makes use of three crucial supporting lemmas, which we will address afterwards. Throughout the subsection we will use an over-bar ($\bar{U}$ etc) to denote quantities corresponding to the 'given original point of interest'. The structure of the proof has many similarities to the arguments of the previous appendix A, but we do not use the tilde convention of remark A.6, instead using different letters to indicate orthonormality: matrices $V, \bar{V}$ have orthonormal columns, while other matrices ($U, \bar{U}, W, \bar{W}$) do not in general. Throughout this section, as elsewhere, the $1/2$ power of a positive-semi-definite matrix denotes its (unique) positive semi-definite square root.

**Proposition B.1** (No spurious local minima). *The (population) objective $\mathcal{L}_{\text{GEP-EY}}$ has no spurious local minima. That is, any matrix $\bar{U}$ that is a local minimum of $\mathcal{L}_{\text{GEP-EY}}$ must in fact be a global minimum of the form described in proposition 3.1.*

*Proof.* Suppose (for contradiction) that $\bar{U}$ is a local optimum of $\mathcal{L}_{\text{GEP-EY}}$, but not a global minimum.

As in appendix A.4, we first reduce to the $B = I$ setting. Write $M = B^{-1/2}AB^{-1/2}$, $W = B^{1/2}U$ and similarly for the 'barred' quantity $\bar{W} = B^{1/2}\bar{U}$.

Following eq. (16), our Eckhart-Young loss transforms as

$$\mathcal{L}_{\text{GEP-EY}}(U) = \|B^{-1/2}AB^{-1/2} - B^{1/2}UU^\top B^{1/2}U\|_F^2 - \|B^{-1/2}AB^{-1/2}\|_F^2$$
$$= \|M - WW^\top\|_F^2 - \|M\|_F^2 =: l(W)$$

and so since $U \mapsto B^{1/2}U$ is a homeomorphism, we see that $\bar{W}$ is a local minimum of $l$ that is not a global minimum.

By lemma B.2, $\bar{W}$ must lie in the space of non-negative eigenvalues of $M$, and therefore also that $l(\bar{W}) = \|M_+ - \bar{W}\bar{W}^\top\|_F^2 - \|M_+\|_F^2$, using the notation of remark A.9.

Next take some QR decomposition $\bar{W} = \bar{V}\bar{R}$. Because $\bar{W}$ is a local minimum of $l$, we must have that $\bar{R}$ is a local minimum of $l' : R \mapsto \|M - \bar{V}RR^\top\bar{V}^\top\|_F$. Therefore $\bar{R}$ must be of the form specified in lemma B.3, and the value of $l$ at $\bar{W}$ is precisely

$$l(\bar{W}) = -\|\bar{V}^\top M_+\bar{V}\|_F^2 \tag{17}$$

Finally we apply lemma B.4 (to the PSD matrix $M_+$) to conclude. This gives an analytic path $V(t)$ with $V(0) = \bar{V}$. We now convert this to a continuous path for $W$. Write $\bar{R} = (\bar{V}^\top M_+\bar{V})^{1/2}\bar{O}$ for an orthogonal matrix $\bar{O}$. Then let $R(t) = (V^\top M_+V)^{1/2}(t)\bar{O}$, and $W(t) = V(t)R(t)$. Then $R(t)$ is continuous since the positive definite square root is continuous on positive semi-definite matrices and by construction, $(RR^\top)(t) = V^\top M_+V$. Note also from the construction in lemma B.4 that $W(t)$ remains in the space spanned by eigenvectors of $M$ with non-negative eigenvalues (we simply rotate towards eigenvector with the largest, and so positive, eigenvalue). Therefore

$$l(W(t)) = -\|V^\top M_+V\|_F^2 < -\|\bar{V}^\top M_+\bar{V}\|_F^2 = l(\bar{W}) \tag{18}$$

contradicting local minimality. $\qquad\square$

The following lemma is stated using the notation of definition A.8 and remark A.9.

**Lemma B.2.** *Let $M \in \mathbb{R}^{D \times D}$ be a symmetric matrix, and write $l(W) = \|M - WW^\top\|_F^2 - \|M\|_F^2$. Then firstly, any $W$ contained within the non-negative eigenspace of $M$ we have*

$$l(W) = \|M_+ - WW^\top\|_F^2 - \|M_+\|_F^2$$

*Secondly, suppose $\bar{W} \in \mathbb{R}^{D \times K}$ is a local minimum of $l$. Then $\text{span}\{\bar{W}\}$ is contained within the non-negative eigenspace of $M$.*

*Proof.* For the first claim, following eq. (14), expand

$$\|M - WW^\top\|^2 = \|M_+ - WW^\top\|^2 - 2\langle M_-, WW^\top\rangle_F + 2\langle M_-, M_+\rangle_F + \|M_-\|^2$$
$$= \|M_+ - WW^\top\|^2 + \|M_-\|^2$$

and use that $\|M\|^2 = \|M_+\|^2 + \|M_-\|^2$.

For the second claim, let $P_-$ be a projection onto the span of eigenvectors of $M$ with strictly negative eigenvalues, and $P_{\geq 0}$. Then write $\bar{W} = P_-\bar{W} + P_{\geq 0}\bar{W} =: \bar{W}_- + \bar{W}_+$. Consider moving along the continuous path $\bar{W}(t) = \gamma(t)\bar{W}_- + \bar{W}_+$ where $\gamma(t) = 1 - t$ for $t \in [0, 1]$. This time expand out further

$$\|M - WW^\top\|^2 = \|M_+\|^2 + \|M_-\|^2 - 2\langle WW^\top, M_+\rangle - 2\langle WW^\top, M_-\rangle_F + \|WW^\top\|^2$$

and then the terms involving $W$ can be written

$$\langle WW^\top, M_+\rangle_F = \langle \bar{W}_+\bar{W}_+^\top, M_+\rangle$$
$$\langle WW^\top, M_-\rangle_F = \gamma^2\langle \bar{W}_-\bar{W}_-^\top, M_-\rangle$$
$$\|WW^\top\|^2 = \|W^\top W\|^2 = \|\bar{W}_+^\top \bar{W}_+ + \gamma^2\bar{W}_-^\top \bar{W}_-\|^2$$
$$= \|\bar{W}_+^\top \bar{W}_+\|^2 + 2\gamma^2\langle \bar{W}_+^\top \bar{W}_+, \bar{W}_-^\top \bar{W}_-\rangle_F + \gamma^4\|\bar{W}_-^\top \bar{W}_-\|^2$$

Since $\langle \bar{W}_-\bar{W}_-^\top, M_-\rangle \leq 0$ we see that the coefficient of $\gamma^2$ is non-negative. If $\bar{W}_- \neq 0$, then the coefficient of $\gamma^4$ is strictly positive and therefore increasing $t$ decreases $\gamma$ and strictly decreases the loss.

We must therefore have $\bar{W}_- = 0$, as required. $\qquad\square$

**Lemma B.3.** *Let $M \in \mathbb{R}^{D\times D}$ be a symmetric positive semi-definite matrix and let $\bar{V} \in \mathbb{R}^{D\times K}$ have orthonormal columns. Consider the loss function*

$$l(R) = \|M - \bar{V}RR^\top\bar{V}\|_F^2 - \|M\|_F^2$$

*over $R \in \mathbb{R}^{K\times K}$. Then the global minima of $l$ are precisely the $R$ satisfying*

$$RR^\top = \bar{V}^\top M\bar{V}.$$

*Moreover, there are no spurious local optima, and the minimum value is precisely*

$$-\|\bar{V}^\top M\bar{V}\|_F^2. \tag{19}$$

*Proof.* First for the global statement. Write $\Gamma = RR^\top$. Then complete the square to give

$$\|M - \bar{V}\Gamma\bar{V}^\top\|_F^2 = \text{trace}(\bar{V}^\top\bar{V})\Gamma^\top(\bar{V}^\top\bar{V})\Gamma - 2\,\text{trace}\,\Gamma(\bar{V}^\top M\bar{V}) + \|M\|_F^2$$
$$= \text{trace}\,\Gamma^2 - 2\,\text{trace}\,\Gamma(\bar{V}^\top M\bar{V}) + \|M\|_F^2$$
$$= \|\Gamma - (\bar{V}^\top M\bar{V})\|_F^2 + \|M\|_F^2 - \|\bar{V}^\top M\bar{V}\|_F^2$$

from which we can read off that the minimum is attained precisely when

$$\Gamma = \bar{V}^\top M\bar{V}$$

and that the optimal value is precisely the value of eq. (19) as claimed (a family of suitable square roots $R$ exist because $\bar{V}^\top M\bar{V}$ inherits positive semi-definite-ness from $M$).

Finally we show there are no spurious local minima. Suppose that $\bar{R}$ is not a global optima. Then the corresponding $\bar{\Gamma}$ is not a global optimum, and $\bar{R} = \bar{\Gamma}^{1/2}\bar{O}$ for some orthogonal matrix $\bar{O}$. But since the objective is just quadratic in $\Gamma$, we can construct a path of positive definite matrices $\Gamma(t)$ with $\Gamma(0) = \bar{\Gamma}$ and $\|M - \bar{V}\Gamma(t)\bar{V}^\top\|_F^2$ strictly decreasing in $t$ (for example following the gradient dynamics). But then, we can simply take $R(t) = \Gamma(t)^{1/2}\bar{O}$, noting that the PSD square root is continuous, to obtain a continuous path of matrices $R(t)$ with $R(0) = \bar{R}$ and $l(R(t)) < l(\bar{R})$ for all $t$. Therefore, $\bar{R}$ is not a local minima either, as required. $\qquad\square$

**Lemma B.4** (Rotating to capture signal, weak). *Let $M \in \mathbb{R}^{D \times D}$ be a symmetric positive definite matrix. Let $\bar{V} \in \mathbb{R}^{D \times K}$ have orthonormal columns, which do not span a top-$K$ subspace for $M$. Then there exists an analytic path $V : [0,1] \mapsto \mathbb{R}^{D \times K}, t \to V(t)$ such that: $V(t)$ has orthonormal columns for all $t$, $V(0) = \bar{V}$, and the signal captured*

$$\zeta(t) = \|V(t)^\top M V(t)\|_F^2$$

*is strictly increasing in $t$.*

*Proof.* Let $\mathcal{V}_K = \text{span}\{\bar{V}\}$. Let the eigenvectors of $M$ be $(v_k^*)_k$ with corresponding eigenvalues $(\lambda_k)_k$ with $\lambda_1 \geq \lambda_2 \geq \cdots \geq \lambda_d \geq 0$. For simplicity we suppose that $v_1^* \notin \mathcal{V}_K$ and that $\lambda_1 > \lambda_2$ (i.e. that there is a strictly separated maximal direction of signal that is not yet captured by the estimated subspace). A very similar construction works more generally but the notation becomes significantly more complicated (we will return to this at the end of the proof).

Now perform a CS-decomposition Stewart & Sun (1990) on the pair of subspaces $(\mathcal{V}_K, \text{span}\{v_1^*\})^4$. This gives a basis $\bar{q}_1, \ldots, \bar{q}_K$ for $\mathcal{V}_K$ where

$$\bar{q}_1 = \cos(\bar{\theta}) \, v_1^* + \sin(\bar{\theta}) \, \bar{p}$$

and $v_1^*, \bar{p}, \bar{q}_2, \ldots, \bar{q}_K$ are mutually orthogonal, and $\bar{\theta} \in [0, \pi/2]$.

Now we construct a path for $t \in [0,1]$ via

$$q_1(t) = \cos(\theta(t)) \, v_1^* + \sin(\theta(t)) \, \bar{p}, \quad \theta(t) := (1-t)\bar{\theta}$$
$$q_k(t) = \bar{q}_k, \quad \text{for } k = 2, \ldots K$$

We can compute

$$q_1^\top M q_1 = \lambda_1 \cos^2 \theta + (\bar{p}^\top M \bar{p}) \sin^2 \theta$$
$$q_1^\top M q_k = \bar{p}^\top M \bar{q}_k \sin \theta, \quad \text{for } k = 2, \ldots K$$

We obtain a corresponding path $V(t)$ as follows. Write $Q(t)$ for the matrix with columns $(q_k(t))_{k=1}^K$, recalling that only the first column depends on $t$. Then $\text{span}\{Q(0)\} = \text{span}\{\bar{q}_k : k \in [K]\} = \mathcal{V}_K$ by construction and so we have $\bar{V} = Q(0) \, \bar{O}$ for some orthogonal matrix $\bar{O} \in \mathbb{R}^{K \times K}$. We therefore define $V(t) = Q(t)\bar{O}$ for $t \in [0,1]$.

To aid with subsequent algebraic manipulations we will write $c^2 = \cos^2 \theta, s^2 = \sin^2 \theta = 1 - c$, $\tau = \bar{p}^\top M \bar{p}$. We can then rewrite $q_1^\top M q_1 = \lambda_1 c^2 + \tau s^2 = \lambda_1 - (\lambda_1 - \tau)s^2$. We also write $\sigma^2 = \sum_{k=2}^K (\bar{p}^\top M \bar{q}_k)^2$. Note that $c^2, s^2$ depend on $t$ through $\theta$, but $\bar{p}, \bar{q}_k$ do not depend on $t$ so neither do $\tau$ or $\sigma^2$. Therefore, up to constant term in $\theta$, we have

$$\zeta(t) = \|V^\top M V\|_F^2 = \|Q^\top M Q\|_F^2$$
$$= \sum_{k,l=1}^K (q_k^\top M q_l)^2$$
$$= (q_1^\top M q_1)^2 + 2\sum_{k=2}^K (q_1^\top M q_k)^2 + \text{cst}$$
$$= (\lambda_1 - (\lambda_1 - \tau)s^2)^2 + \sum_{k=2}^K s^2(\bar{p}^\top M \bar{q}_k)^2 + \text{cst}$$
$$= \lambda_1^2 - 2s^2\lambda_1(\lambda_1 - \tau) + s^4(\lambda_1 - \tau)^2 + 2s^2\sigma^2 + \text{cst}$$
$$= s^4(\lambda_1 - \tau)^2 + 2s^2(\sigma^2 - \lambda_1(\lambda_1 - \tau)) + \lambda_1^2 + \text{cst}$$

Differentiating with respect to $s^2$ therefore gives

$$\frac{\partial \zeta}{\partial(s^2)} = 2s^2(\lambda_1 - \tau)^2 + 2\left(\sigma^2 - \lambda_1(\lambda_1 - \tau)\right)$$

---

[4]This may seem like magic, but can be motivated for example through the characterisation of geodesics on the Stiefel manifold through the CS-decomposition, see the excellent monograph of Edelman et al. (1998)

We see that this is strictly increasing in $s^2$ because $\tau \le \lambda_2 < \lambda_1$. Therefore, it is strictly negative for all $s^2 \in (0, 1)$ if and only if it is non-positive when evaluated at $s^2 = 1$, which gives the condition

$$
\begin{aligned}
0 \ge (\lambda_1 - \tau)^2 + \left(\sigma^2 - \lambda_1(\lambda_1 - \tau)\right) \\
= \lambda_1^2 - 2\tau\lambda_1 + \tau^2 + \sigma^2 - \lambda_1^2 + \tau\lambda_1 \\
= \tau^2 + \sigma^2 - \tau\lambda_1.
\end{aligned}
$$

We now show that this condition in fact follows by Pythagoras' theorem: we have

$$
\sigma^2 + \tau^2 \le \|M\bar{p}\|^2 \le \lambda_2\tau \le \lambda_1\tau,
$$

as required (using $\tau \ge 0$ since $M$ is positve semi-definite). The central inequality holds because $\bar{p}$ is orthogonal to the leading eigenvector $v_1^*$. This can be seen by $v_1^*$ to a full orthonormal basis of eigenvectors for $M$ denoted $(v_k^*)_{k=1}^D$ and with corresponding eigenvectors $(\lambda_k)_{k=1}^D$. Since $\bar{p}$ is orthogonal to $v_1^*$ we can expand $\bar{p} = \sum_{k=2}^K \mu_k v_k^*$ to give

$$
\|M\bar{p}\|^2 = \sum_{k=2}^K \lambda_k^2 \mu_k^2 \le \sum_{k=2}^K \lambda_2 \lambda_k \mu_k^2 = \lambda_2 \bar{p}^\top M \bar{p}
$$

Finally we return to sketch the extension to the general case. Here, let $k$ be the first index $k$ such that $\bar{v}_k \notin \mathcal{V}_k^*$, where $\mathcal{V}_k^*$ denotes the span of eigenvectors of $M$ with eigenvalue greater than or equal to $\lambda_k$ (which may be of dimension larger than $k$ when the $\lambda_k$ eigenvalue is repeated). Then perform the CS decomposition on the pair of subspaces $(\mathcal{V}_K, \mathcal{V}_k^*)$ to get a basis $\bar{q}_l = v_l^* \cos\bar{\theta}_l + \bar{p}_l \sin\bar{\theta}_l$ Construct a path by taking $\theta_k(t) = (1-t)\bar{\theta}_k$ and leaving all other $\theta_l$ fixed. Then $\bar{p}_k$ is orthogonal to $\mathcal{V}_k^*$ so $\bar{p}_k^T M \bar{p}_k < \lambda_k$, which allows the rest of the argument to proceed as before. $\qquad\square$

## B.2 CONJECTURED STRONGER CONSTRUCTION

We now conjecture that a stronger version of the construction of lemma B.4 can produce an analytic path that increases each eigenvalue of $V^\top M V$ individually, rather than just increasing its Frobenius norm. We state this below as conjecture B.5. We have yet to find a complete proof of this result, but have reduced it to a matrix analytic result that appears tractable, and which appears to hold based on numerical simulations.

We present partial progress, because we believe this is interesting and important enough to merit further investigation. We also need the result later when analysing Barlow twins.

**Conjecture B.5** (Rotating to capture signal: each eigenvalue increases). *Let $M \in \mathbb{R}^{D \times D}$ be a symmetric positive definite matrix, with eigenvalues $(\lambda_k^*)_{k=1}^D$. Let $\bar{V} \in \mathbb{R}^{D \times K}$ have orthonormal columns such that $\bar{\Lambda} := \bar{V}^\top M \bar{V}$ is diagonal.*

*Then there exist an analytic path $(V(t))_{t \in [0,1]}$ of matrices in $\mathbb{R}^{D \times K}$ with orthonormal columns such that $V(0) = \bar{V}$, and the matrix $\Lambda(t) := V(t)^\top M V(t)$ is diagonal with entries $\lambda_k(t)$ that are strictly increasing for any $k$ such that $\lambda_k(0) < \lambda_k^*$.*

*Partial proof progress.* Again we exploit the CS-decomposition. Write $\mathcal{V}_K^*$ for the span of eigenvectors of $M$ with eigenvalue greater than or equal to $\lambda_K$ (which may be of dimension larger than $K$ when the $\lambda_K$ eigenvalue is repeated).

Now perform a CS decomposition on the pair of subspaces $(\mathcal{V}_K, \mathcal{V}_K^*)$. This gives a basis $\bar{q}_1, \ldots, \bar{q}_K$ for $\mathcal{V}_K$ of the form

$$
\bar{q}_k = \cos(\bar{\theta}_k)\, v_k^* + \sin(\bar{\theta}_k)\, \bar{p}_k \tag{20}
$$

where $v_1^*, \ldots, v_K^*$ form an orthonormal basis for a $K$-dimensional subspace of $\mathcal{V}_K^*$, and the $\bar{p}_k$ are orthonormal to each other and also to all the $v_k^*$. In fact the $\bar{p}_k$ give a basis for $\mathrm{span}\{\mathcal{V}_K, \mathcal{V}_K^*\} \cap (\mathcal{V}_K^*)^\perp$.

Then as in the proof of lemma B.4, we can consider reducing the angles with some functions $\theta_k(t)$ for $k \in [K]$ to obtain vectors $q_k(t)$. Package the first $K$ vectors $(q_k(t))_{k=1}^K, (v_k^*)_{k=1}^K, (\bar{p}_k)_{k=1}^K$ into the matrices $Q, V^*, \bar{P}$ each in $\mathbb{R}^{D \times K}$. In addition write

$$
C(t) = \mathrm{diag}(\cos\theta_k(t) : k \in [K]), \; S(t) = \mathrm{diag}(\sin\theta_k(t) : k \in [K]) \tag{21}
$$

Then eq. (20) can be rewritten as

$$Q(t) = V^* C(t) + \bar{P} S(t)$$

We then convert back to a path $V(t)$ of the form $V(t) = Q(t)O(t)$ where $O(t)$ is an orthogonal matrix that diagonalises $Q^\top MQ(t)$ (rather than having a constant matrix $O$ as in lemma B.4). The fact that such an orthogonal matrix can be taken to be an analytic function of $t$ follows from Kato (1995)[Subsection 2.6.2][5].

Next note that the eigenvalues of $V^\top MV$ are precisely the eigenvalues of the matrix

$$(Q^\top MQ)(t) = (V^* C(t) + \bar{P} S(t))^\top M (V^* C(t) + \bar{P} S(t))$$
$$= C(t) V^{*\top} MV^* C(t) + S(t) \bar{P}^\top M\bar{P} S(t)$$

And that in addition, each of the eigenvalues of $V^{*\top} MV^*$ is greater than or equal to $\lambda_K$, while the eigenvalues of $\bar{P}^\top M\bar{P}$ are all strictly less than $\lambda_K$ (and indeed less than or equal to the next biggest eigenvalue of $M$). One can therefore write $V^{*\top} MV^* = \lambda_K I_K + \Delta$, and $\bar{P}^\top M\bar{P} = \lambda_K I_K - \Gamma$, for positive semi-definite $\Delta, \Gamma$. Then using $C(t)^2 + S(t)^2 = I_K$, we obtain

$$(Q^\top MQ)(t) = \lambda_K I_K + C(t)\Delta C(t) - S(t)\Gamma S(t).$$

The following claim would give a way to proceed.

**Claim:** Let $\Delta, \Gamma$ be positive semi-definite. Let $C(t), S(t)$ be constructed as in eq. (21) for some functions $\theta_k(t)$ with each $\theta_k$ valued in $[0, \pi/2]$ and decreasing in $t$. Then for each $k$ the $k^{\text{th}}$ eigenvalue $\lambda_k(C(t)\Delta C(t) - S(t)\Gamma S(t))$ is increasing in $t$.

We have verified this numerically for a large number of random positive semi-definite matrices $\Delta, \Gamma$ and decreasing functions $\theta_k$. The result becomes straightforward to prove when either $\Delta$ or $\Gamma$ is zero. However, the proof in general has eluded us.

$\square$

## B.3 QUANTITATIVE RESULTS

To use the results from Ge et al. (2017) we need to introduce their definition of a $(\theta, \gamma, \zeta)$-strict saddle.

**Definition B.6.** We say function $l(\cdot)$ is a $(\theta, \gamma, \zeta)$-**strict saddle** if for any $x$, at least one of the following holds:

1. $\|\nabla l(x)\| \geq \theta$

2. $\lambda_{\min}(\nabla^2 l(x)) \leq -\gamma$

3. $x$ is $\zeta$-close to $\mathcal{X}^*$ - the set of local minima.

.

We can now state restate Lemma 13 from Ge et al. (2017) in our notation; this was used in their analysis of robust PCA, and directly applies to our PCA-type formulation.

**Lemma B.7** (Strict saddle for PCA). *Let $M \in \mathbb{R}^{D \times D}$ be a symmetric PSD matrix, and define the matrix factorization objective over $Z \in \mathbb{R}^{D \times K}$*

$$l(Z) = \|M - ZZ^\top\|^2$$

*Assume that $\lambda_K^* := \lambda_K(M) \geq 15\lambda_{K+1}(M)$. Then*

1. *all local minima satisfy $ZZ^\top = \mathcal{P}_K(M)$ - the best rank-$K$ approximation to $M$*

2. *the objective l(Z) is $(\epsilon, \Omega(\lambda_K^*), \mathcal{O}(\epsilon/\lambda_K^*))$-strict saddle.*

---

[5]Perturbation theory in a finite dimensional space - Perturbation of symmetric operators - Orthonormal families of eigenvectors

However, we do not want to show a strict saddle of $l$ but of $\mathcal{L}_{\text{GEP-EY}} : U \mapsto l(B^{1/2}U)$. Provided that $B$ has strictly positive minimum and bounded maximum eigenvalues this implies that $\mathcal{L}_{\text{GEP-EY}}$ is also strict saddle, as we now make precise.

**Lemma B.8** (Change of variables for strict saddle conditions). *Suppose that $l$ is $(\theta, \gamma, \zeta)$-strict saddle and let $L : U \mapsto l(B^{1/2}U)$ for $B$ with minimal and maximal eigenvalues $\sigma_{min}, \sigma_{max}$ respectively. Then $L$ is $\left(\sigma_{max}^{1/2}\theta, \sigma_{min}\gamma, \sigma_{max}^{1/2}\zeta\right)$-strict saddle.*

*Proof.* Write $g(U) = B^{1/2}U$. Then $L = l \circ g$, so by the chain rule:

$$D_U L = D_{B^{1/2}U} l \circ D_U g \ : \ \delta U \mapsto \langle \nabla l(B^{1/2}U), B^{1/2}\delta U \rangle = \langle B^{1/2}\nabla l(B^{1/2}U), \delta U \rangle$$

Therefore

$$\|\nabla L(U)\| = \|B^{1/2}\nabla l(B^{1/2}U)\| \geq \sigma_{\min}^{1/2}\|l(B^{1/2}U)\|$$

By a further application of the chain rule we have

$$D_U^2 L \ : \ \delta U, \delta U \mapsto D_{B^{1/2}U}^2 l(B^{1/2}\delta U, B^{1/2}\delta U)$$

Suppose $\lambda_{\min}(\nabla^2 l(Z)) \leq -\gamma$ then by the variational characterization of eigenvalues, there exists some $\delta Z$ such that $\langle \delta Z, \nabla^2 l(Z)\delta Z \rangle \leq -\gamma\|\delta Z\|^2$. Then taking $\delta U = B^{-1/2}\delta Z$ gives

$$
\begin{aligned}
\langle \delta U, \nabla^2 L(U)\delta U \rangle &= \langle B^{1/2}\delta U, \nabla^2 l(B^{1/2}U)B^{1/2}\delta U \rangle \\
&= \langle \delta Z, \nabla^2 l(Z)\delta Z \rangle \\
&\leq -\gamma\|\delta Z\|^2 \\
&\leq -\gamma\sigma_{\min}\|\delta U\|^2
\end{aligned}
$$

Thirdly, suppose that $\|B^{1/2}U - Z^*\| \leq \zeta$ for some local optimum $Z^*$ of $l$. Then since $B$ is invertible, $U^* := B^{-1/2}Z^*$ is a local optimum of $L$. In addition:

$$\|U - U^*\| = \|B^{1/2}(U - U^*)\| \leq \sigma_{\max}^{1/2}\|B^{1/2}U - Z^*\| \leq \zeta$$

Finally, consider some arbitrary point $U_0$. Let $Z_0 = B^{1/2}U_0$. Then by the strict saddle property for $l$ one of the following must hold:

1. $\|\nabla l(Z_0)\| \geq \theta \quad \implies \quad \|\nabla L(U_0)\| \geq \sigma_{\min}^{1/2}\theta$

2. $\lambda_{\min}(\nabla^2 l(Z_0)) \leq -\gamma \quad \implies \quad \lambda_{\min}(\nabla^2 L(U_0) \leq -\sigma_{\min}\gamma$

3. $Z_0$ is $\zeta$-close to a local-minimum $Z^*$, which implies that $U_0$ is $(\sigma_{\max}^{1/2}\zeta)$-close to a local minimum $B^{-1/2}Z^*$ of $L$.

$\square$

By combining lemma B.7 with lemma B.8, we can conclude that our objective does indeed satisfy a (quantitative) strict saddle property. This is sufficient to show that certain local search algorithms will converge in polynomial time Ge et al. (2017).

However, this version of the strict saddle property is not quite enough to prove the claim for stochastic gradient descent (SGD). Certain extra conditions are given in Ge et al. (2015) to guarantee polynomial time convergence of noisy SGD. These are: 1. a notion of local strict convexity near any local minima, and 2. boundedness assumptions. The first assumption is easy to show in our setting, but the second clearly fails. That being said, we could approximate the objective by mollifying it to be bounded outside a large ball. Then it should be straightforward to use a supermartingale argument to show that with high probability the sample paths are contained within said ball; and then inherit convergence guarantees from the bounded case.

## C    FURTHER BACKGROUND AND RESULTS FOR LINEAR AND DEEP CCA

This section contains an eclectic assortment of results regarding CCA. We split this into two sub-sections corresponding to linear and deep CCA respectively. To help the reader navigate the sub-sections, we now provide a short summary of their contents.

Firstly, we consider linear CCA. In appendix C.1.1 we show that the GEP formulation of two view (ridge) CCA presented in the main text corresponds to the sequential, constrained formulation of two view (ridge) CCA that is more standard in the literature. Then in appendix C.1.2 we prove versions of eigenvalue interlacing for multiview CCA; these will be useful in appendix D. We also consider CCA with tied weights in appendix C.1.3; we show that when paired data is generated by applying a pair of i.i.d. augmentations to a single original datum the CCA weights can be chosen to be tied; this is useful in appendix D and for the following analysis of the deep case.

Secondly, we consider deep CCA. We give a full proof that our objective recovers a sensible notion of deep multiview CCA (lemma 3.6) in appendix C.2.1. Finally, we consider deep CCA with tied weights in appendix C.2.2 and show that when the data is generated by i.i.d. augmentations tying the weights can only help with capturing correlation; weight tying can therefore be seen as a form of regularisation in this i.i.d. augmentation setting.

### C.1    LINEAR CCA

#### C.1.1    2-VIEW RCCA: EQUIVALENCE OF GEP TO CONSTRAINED FORMULATIONS

The main result in this subsection is lemma C.3 which states that the standard sequential, constrained formulation of ridge CCA (RCCA), and is equivalent to our GEP formulation eq. (6) in the 2 view case. The proof strategy follows a standard argument, from e.g. (Anderson, 2003), but is adapted to our notation for this more general case of ridge CCA.

Note that by taking the parameters $\alpha = 0$ we recover CCA and $\alpha = 1$ recovers PLS and so this conclusion also holds for PLS and CCA, which we view as special cases of RCCA.

First, we introduce more intuitive notation for our general GEP formulation that is closer to that used in previous expositions. Namely, in the block-matrix definitions of eq. (6) we relabel the blocks

$$\Sigma^{(ij)} := A^{(ij)} = \text{Cov}(X^{(i)}, X^{(j)}); \quad \Sigma_\alpha^{(ii)} := B_\alpha^{(ii)} = (1 - \alpha^{(i)})\text{Var}(X^{(i)}) + \alpha^{(i)} I_{D^{(i)}}.$$

to highlight their nature as (regularised) covariance matrices. With this notation the GEP matrices $(A, B)$ can be written in full as

$$A = \begin{pmatrix} 0 & \Sigma^{(12)} \\ \Sigma^{(21)} & 0 \end{pmatrix}, \quad B = \begin{pmatrix} \Sigma_\alpha^{(11)} & 0 \\ 0 & \Sigma_\alpha^{(22)} \end{pmatrix}.$$

We also need the following lemma to help analyse the two-view GEP formulation of RCCA.

**Lemma C.1** (2-view RCCA GEP recovers orthogonality)**.** *Let $U \in \mathbb{R}^{D \times K}$ be a matrix whose columns form a top-$K$ sequence of normalised gevectors for the RCCA GEP, partitioned as $(U^{(1)}, U^{(2)}) \in \mathbb{R}^{D^{(1)} \times K} \times \mathbb{R}^{D^{(2)} \times K}$. Suppose the corresponding top-$K$ gevalues $(\lambda_k)_{k=1}^K$ are all strictly positive. Then in fact $U^{(i)\top} \Sigma_\alpha^{(ii)} U^{(i)} = \frac{1}{2} I_K$ for $i \in [2]$.*

*Proof.* Let $\Lambda = \text{diag}(\lambda_1, \ldots, \lambda_K) \in \mathbb{R}^{K \times K}$ be a diagonal matrix containing the top-$K$ gevalues of the RCCA GEP $(A, B)$. Then, since the columns of $U$ form a top-$K$ sequence of gevectors

$$AU = \begin{pmatrix} \Sigma^{(12)} U^{(2)} \\ \Sigma^{(21)} U^{(1)} \end{pmatrix} = BU\Lambda = \begin{pmatrix} \Sigma_\alpha^{(11)} U^{(1)} \\ \Sigma_\alpha^{(22)} U^{(2)} \end{pmatrix} \Lambda. \tag{22}$$

Write $M^{(i)} = U^{(i)\top} \Sigma_\alpha^{(ii)} U^{(i)}$ for $i \in [2]$. Then each $M^{(i)}$ is symmetric. Plugging this definition into eq. (22) gives

$$M^{(1)}\Lambda = U^{(1)\top} \Sigma_\alpha^{(11)} U^{(1)} \Lambda = U^{(1)\top} \Sigma_\alpha^{(12)} U^{(2)} = \Lambda U^{(2)} \Sigma_\alpha^{(22)} U^{(2)} = \Lambda M^{(2)}$$

But then because $\Lambda$ is diagonal this simply gives the system of equations

$$\lambda_k m_{kl}^{(1)} = \lambda_l m_{kl}^{(2)} \text{ for } k, l \in [K]$$

In particular, since $\lambda_k > 0$, taking $l = k$ yields $m_{kk}^{(2)} = m_{kk}^{(1)}$. Taking $k \neq l$ with $\lambda_k \neq \lambda_l$ the two sets of equations $\lambda_k m_{kl}^{(1)} = \lambda_l m_{kl}^{(2)}$, $\lambda_l m_{kl}^{(1)} = \lambda_k m_{kl}^{(2)}$ have unique solution $m_{kl}^{(2)} = m_{kl}^{(1)} = 0$. While if $k \neq l$ with $\lambda_k = \lambda_l$ then we must have $m_{kl}^{(1)} = m_{kl}^{(2)}$ by dividing through by $\lambda_k = \lambda_l > 0$.

By combining these 3 cases we can conclude that $M^{(1)} = M^{(2)}$. Then by the assumption of $B$-orthonormality we in fact have

$$I_K = U^\top B_\alpha U = \sum_{i \in [2]} U^{(i)\,\top} \Sigma_\alpha^{(ii)} U^{(i)} = M^{(1)} + M^{(2)} = 2M^{(1)}$$

so indeed we conclude $M^{(1)} = M^{(2)} = \frac{1}{2} I_K$, as claimed. $\qquad\square$

**Definition C.2** (2-view Ridge CCA sequential definition). A sequence of vectors $(u_k)_{k=1}^K$ partitioned as $u_k = (u_k^{(i)})_{i=1}^2$ is a top-$K$ sequence of ridge CCA directions if the successive $u_k$ solve the successive maximisations

$$\underset{u = (u^{(1)}, u^{(2)}) \in \mathbb{R}^{D^{(1)}} \times \mathbb{R}^{D^{(2)}}}{\text{maximize}} \quad u^{(1)} A^{(12)} u^{(2)} \quad \text{subject to} \quad u^{(i)\,\top} B_\alpha^{(ii)} u^{(i)} = 1 \text{ for } i \in [2],$$

$$u^{(i)\,\top} B_\alpha^{(ii)} u_l^{(i)} = 0 \text{ for } l \in [k-1] \tag{23}$$

where we define $[0] = \emptyset$ to allow the same formulation to hold for the $k = 1$ case.

**Lemma C.3** (2-view ridge recovers GEP). *A sequence of vectors $(u_k)_{k=1}^K$ is a top-$K$ sequence of normalised ridge CCA directions if and only if $(\frac{1}{\sqrt{2}} u_k)_{k=1}^K$ is a top-$K$ sequence of normalised gevectors for the GEP defined in eq. (6).*

*Proof.* Note first that in each case there is some pair attaining the maximum because we are optimising a continuous objective over a compact set (if $\text{Var}(X^{(i)})$ is not full rank, then can work in its range, and treat kernel separately).

We will prove the claim by induction over $K$.

Suppose that the claim holds for $K - 1$. Note that a sequence of vectors $(u_k)_{k=1}^K$ is a top-$K$ sequence of normalised ridge CCA directions precisely when $(u_k)_{k=1}^{K-1}$ are a top-$K - 1$ sequence of normalised ridge CCA directions and $u_K$ solves the program 23 (taking $k = K$). Similarly, a sequence of vectors $(u_k)_{k=1}^K$ is a top-$K$ sequence of normalised gevectors precisely when $(u_k)_{k=1}^{K-1}$ are a top-$K - 1$ sequence of normalised gevectors and $u_K$ is a normalised gevector with gevalue $\lambda_K$, which is $B$-orthogonal to the previous $(u_k)_{k=1}^{K-1}$. Therefore it is sufficient to show that for any fixed sequence of normalised gevectors (equivalently normalised ridge CCA directions by the induction hypothesis) $(u_k)_{k=1}^{K-1}$: a vector $u_K$ solves the program 23 precisely when it is a normalised gevector with gevalue $\lambda_K$, which is $B$-orthogonal to the previous $(u_k)_{k=1}^{K-1}$.

Thus, suppose we are given arbitrary sequence of normalised gevectors $(u_k)_{k=1}^{K-1}$. We will characterise solutions to the program 23. By adding in appropriate Lagrange multipliers $\mu = (\mu^{(1)}, \mu^{(2)}) \in \mathbb{R} \times \mathbb{R}$ we obtain the Lagrangian

$$L(u_K; \mu) = u_k^{(1)\,\top} A^{(12)} u_k^{(2)} + \sum_{i \in [2]} \left\{ \tfrac{1}{2} \mu_K^{(i)} (1 - u_K^{(i)\,\top} B_\alpha^{(ii)} u_K^{(i)}) - \sum_{l \in [K-1]} \mu_l^{(i)} u_K^{(i)\,\top} B_\alpha^{(ii)} u_l^{(i)} \right\}.$$

This gives the first order conditions

$$0 = \partial_{u_K^{(i)}} L = A^{(ij)} u_K^{(j)} - \mu_K^{(i)} B_\alpha^{(ii)} u_K^{(i)} - \sum_{l \in [K-1]} \mu_l^{(i)} B_\alpha^{(ii)} u_l^{(i)} \tag{24}$$

for $i \in [2]$ and where $j := 3 - i$ denotes the view-index that is not $i$.

By the inductive hypothesis we have $u_l^{(i)\top} B^{(ii)} u_{l'}^{(i)} = \delta_{ll'}$ for $l, l' \in [K-1]$, and also that $A^{(ij)} u_l^{(j)} = \lambda_l B^{(ii)} u_l^{(i)}$. By feasibility, we also have that $u_l^{(i)\top} B^{(ii)} u_K^{(i)} = 0$. So taking the inner product of eq. (24) with $u_l^{(i)}$ gives

$$0 = u_l^{(i)\top} \partial_{u_K^{(i)}} L = \underbrace{u_l^{(i)\top} A^{(ij)} u_K^{(j)}}_{=\lambda_l u_l^{(j)\top} B^{(jj)} u_K^{(j)} = 0} +0 - \mu_l^{(i)} u_l^{(i)\top} B_\alpha^{(ii)} u_l^{(i)} = -\mu_l^{(i)}$$

Therefore the pair of first order conditions eq. (24) reduce to the pair of equations

$$A^{(ij)} u_K^{(j)} = \mu_K^{(i)} B_\alpha^{(ii)} u_K^{(i)}, \quad A^{(ji)} u_K^{(i)} = \mu_K^{(j)} B_\alpha^{(jj)} u_K^{(j)}$$

Next we can show that in fact the remaining Lagrange multipliers for the $K$th pair are equal across views:

$$\mu_K^{(1)} = \mu_K^{(1)} u^{(1)\top} B_\alpha^{(11)} u^{(1)} = u^{(1)\top} A^{(12)} u^{(2)} = u^{(2)} A^{(21)} u^{(1)} = \mu_K^{(2)} u^{(2)\top} B^{(22)} u^{(2)} = \mu_K^{(2)}.$$

So writing $\mu_K = \mu_K^{(1)} = \mu_K^{(2)}$ we in fact have the stronger pair of first order conditions

$$A^{(12)} u_K^{(2)} = \mu_K B_\alpha^{(11)} u_K^{(1)}, \quad A^{(21)} u_K^{(1)} = \mu_K B_\alpha^{(22)} u_K^{(2)}$$

which is precisely saying that $u_K$ is a gevector:

$$\begin{pmatrix} 0 & A^{(12)} \\ A^{(21)} & 0 \end{pmatrix} \begin{pmatrix} u_K^{(1)} \\ u_K^{(2)} \end{pmatrix} = \mu_K \begin{pmatrix} B_\alpha^{(11)} & 0 \\ 0 & B_\alpha^{(22)} \end{pmatrix} \begin{pmatrix} u_K^{(1)} \\ u_K^{(2)} \end{pmatrix}$$

Finally we can tie up the argument. If $\frac{1}{\sqrt{2}} u_K$ is a normalised gevector with gevalue $\mu_K$, then by lemma C.1, $u_K$ is feasible for the program 23 and attains the value $u_K^{(1)\top} A u_K^{(2)} = \mu_K u_K^{(1)\top} B u_K^{(1)} = \mu_K$. Since the maximal value is attained at a gevector, this maximal value is precisely the largest remaining gevalue, namely $\lambda_K$. Therefore $u_K$ is optimal for program 23 precisely when $\frac{1}{\sqrt{2}} u_K$ is a normalised gevector with gevalue $\lambda_K$, as required.

$\qquad\square$

### C.1.2 INTERLACING RESULTS

First we state a standard result from matrix analysis. This is simply Theorem 2.1 from Haemers (1995), but with notation changed to match our context. We therefore omit the (straightforward) proof.

**Lemma C.4.** *Let $V \in \mathbb{R}^{D \times K}$ such that $V^\top V = I_K$ and let $M \in \mathbb{R}^{D \times D}$ be symmetric with an orthonormal set of eigenvectors $v_1, \ldots, v_D$ with eigenvalues $\lambda_1 \geq \cdots \geq \lambda_D$. Define $C = V^\top M V$, and let $C$ have eigenvalues $\mu_1 \geq \cdots \geq \mu_K$ with respective eigenvectors $y_1 \ldots y_K$.*

*Then*

- $\mu_k \leq \lambda_k$ *for $k = 1, \ldots, K$.*

- *if $\mu_k = \lambda_k$ for some $k$ then $C$ has a $\mu_k$-eigenvector $y$ such that $Vy$ is a $\mu_k$-eigenvector of $M$.*

- *if $\mu_k = \lambda_k$ for $k = 1, \ldots, K$ then $Vy_k$ is a $\mu_k$-eigenvector of $M$ for $k = 1, \ldots, K$.*

This immediately gives us a related result for generalized eigenvalues.

**Corollary C.5** (Generalized Eigenvalue Interlacing). *Consider the GEP $(A, B)$ where $A \in \mathbb{R}^{D \times D}$ is symmetric and $B \in \mathbb{R}^{D \times D}$ symmetric positive definite; let these have $B$-orthonormal generalized eigenvectors $u_1, \ldots, u_D$ with eigenvalues $\lambda_1, \ldots, \lambda_D$.*

*Let $U \in \mathbb{R}^{D \times K}$ such that $U^\top B U = I_K$, define $C = U^\top A U$, and let $C$ have eigenvalues $\mu_1 \geq \cdots \geq \mu_K$ with respective eigenvectors $y_1 \ldots y_K$.*

*Then*

- $\mu_k \leq \lambda_k$ *for* $k = 1, \ldots, K$.

- *if* $\mu_k = \lambda_k$ *for some* $k$ *then* $(C, V)$ *has a* $\mu_k$-*generalised-eigenvector* $y$ *such that* $Uy$ *is a* $\mu_k$-*generalised-eigenvector of* $(A, B)$.

- *if* $\mu_k = \lambda_k$ *for* $k = 1, \ldots, K$ *then* $Uy_k$ *is a* $\mu_k$-*generalised-eigenvector of* $(A, B)$ *for* $k = 1, \ldots, K$.

*Proof.* As in previous appendices, we convert from the GEP $(A, B)$ to an eigenvalue problem for $M := B^{-1/2}AB^{-1/2}$ by defining $V = B^{-1/2}U$, and $v_d = B^{1/2}u_d$.

We now check that the conditions and conclusions of lemma C.4 biject with the conditions and conclusions of this present lemma.

Indeed $(u_d)_d$ are $B$-orthonormal gevectors of $(A, B)$ if and only if $(v_d)_d$ are orthonormal evectors of $M$; the matrices $C$ and then coincide and so so does its eigenvectors and eigenvalues.

This proves the result. $\qquad\square$

We can now apply this to the multiview CCA problem, generalising the two-view case.

**Lemma C.6** (Interlacing for MCCA). *Let* $(X^{(i)})_{i=1}^I$ *be random vectors taking values in* $\mathbb{R}^{D^{(i)}}$ *respectively, as in section 2. Take arbitrary full-rank weight matrices* $U^{(i)} \in \mathbb{R}^{D^{(i)} \times K}$ *for* $i \in \{1, \ldots, I\}$ *and define the corresponding transformed variables* $Z^{(i)} = \langle U^{(i)}, X^{(i)} \rangle$. *Then we have the element-wise inequalities*

$$\mathrm{MCCA}_K(Z^{(i)}, \ldots, Z^{(I)}) \leq \mathrm{MCCA}_K(X^{(1)}, \ldots, X^{(I)}) \tag{25}$$

*Moreover simultaneous equality in each component holds if and only if there exist matrices* $Y^{(i)} \in \mathbb{R}^{K \times K}$ *for* $i \in [I]$ *such that the* $(U^{(i)}Y^{(i)})_{i=1}^I$ *are a set of top-$K$ weights for the MCCA problem.*

*Proof.* Let the matrices $A, B$ be those from the MCCA GEP in eq. (6) defined by the input variables $X$. By definition, $\mathrm{MCCA}_K(X^{(1)}, \ldots, X^{(I)})$ is precisely the vector of the top-$K$ such generalised eigenvalues.

Then the corresponding matrices defining the GEP for $Z$ are block matrices $\bar{A}, \bar{B}$ defined by the blocks

$$\begin{aligned} \bar{A}^{(ij)} &= \mathrm{Cov}(Z^{(i)}, Z^{(j)}) = U^{(i)^\top}\mathrm{Cov}(X^{(i)}, X^{(j)})U^{(j)} \\ \bar{B}^{(ii)} &= \mathrm{Var}(Z^{(i)}) = U^{(i)^\top}\mathrm{Var}(X^{(i)})U^{(i)} \end{aligned} \tag{26}$$

Now define the $D \times (KI)$ block diagonal matrix $\tilde{U}$ to have diagonal blocks $U^{(i)}$. Then the definition from eq. (26) is equivalent to the block-matrix equations $\bar{A} = \bar{U}^\top A\bar{U}$, $\bar{B} = \bar{U}^\top B\bar{U}$, both in $\mathbb{R}^{(KI) \times (KI)}$. Finally, we define a normalised version $\hat{U} = \bar{U}\bar{B}^{-1/2}$ (possible because $B$ positive definite and $\bar{U}$ of full rank).

We can now apply the eigenvalue interlacing result of corollary C.5 to the GEP $(A, B)$ and $B$-orthonormal matrix $\hat{U} \in \mathbb{R}^{D \times IK}$. Let the matrix $\bar{B}^{-1/2}\bar{A}\bar{B}^{-1/2} = \hat{U}^\top A\hat{U}$ have top-$K$ eigenvalues $\rho_1 \geq \cdots \geq \rho_K$ with respective eigenvectors $y_1, \ldots, y_K$. Then the $(\rho_k)_{k=1}^K$ are precisely the first $K$ successive multiview correlations between the $Z^{(i)}$. As before, the first $K$ successive multiview correlations $\rho_k^*$ between the $X^{(i)}$ are precisely the first $K$ generalised eigenvalues of the GEP $(A, B)$. We therefore we have the element-wise inequalities $\rho_k \leq \rho_k^*$ for each $k = 1, \ldots, K$.

Moreover, equality for each of the top-$K$ multiview correlations implies that $\hat{U}y_k$ is a generalised-eigenvector of the original GEP $(A, B)$ for $k = 1, \ldots K$ (still by corollary C.5). Letting $Y^{(i)} = \begin{pmatrix} y_1^{(i)} & \ldots & y_K^{(i)} \end{pmatrix}$ then gives the equality case statement.

$\qquad\square$

### C.1.3 CCA WITH TIED WEIGHTS

It is intuitive that under certain symmetry between $(X^{(1)}, X^{(2)})$ that the CCA weights might be tied, i.e. $u_k^{(1)} = u_k^{(2)}$ for all $k$ with $\rho_k > 0$. One natural sort of symmetry to consider is exchangeability, that is that $(X^{(1)}, X^{(2)}) \stackrel{d}{=} (X^{(2)}, X^{(1)})$. However, exchangeability is not sufficient to guarantee tied weights, as the following example shows.

**Example C.7.** *Let $X^{(1)}, X^{(2)}$ be a pair of $\mathbb{R}^2$ valued random vectors with*

$$\mathrm{Var}\left(X^{(i)}\right) = I_2 \text{ for } i \in [2], \quad \mathrm{Cov}\left(X^{(1)}, X^{(2)}\right) = \begin{pmatrix} 0 & \rho \\ \rho & 0 \end{pmatrix}$$

*Then one one possible choice of canonical directions is*

$$(u_1^{(1)}, u_1^{(2)}), (u_2^{(1)}, u_2^{(2)}) = (e_1, e_2), (e_2, e_1)$$

*where $e_1 = \begin{pmatrix} 1 \\ 0 \end{pmatrix}, e_2 = \begin{pmatrix} 0 \\ 1 \end{pmatrix}$ are standard basis vectors.*

*In fact, the space of canonical directions is degenerate (due to the repeated eigenvalue of $\rho$) but we can use standard results to characterise possible choices of canonical directions. Take any set of canonical directions $u_k^{(i)}$ for $k \in [2], i \in [2]$. Let $U^{(i)}$ be a matrix whose columns are the first and second canonical directions for the $i^{th}$ view, for $i \in [2]$. Then these are of the form*

$$U^{(1)} = \begin{pmatrix} 1 & 0 \\ 0 & 1 \end{pmatrix} O, \quad U^{(2)} = \begin{pmatrix} 0 & 1 \\ 1 & 0 \end{pmatrix} O$$

*where $O \in \mathbb{R}^{2 \times 2}$ is orthogonal.*

**Lemma C.8.** *Let $(X^{(1)}, X^{(2)})$ be an exchangeable pair of random vectors, each of full rank. Suppose that $(u^{(1)}, u^{(2)})$ are a pair of canonical directions with $u^{(1)} \neq u^{(2)}$ and canonical correlation $\rho > 0$. Then $\mathrm{Cov}(X^{(1)}, X^{(2)})$ has a strictly negative eigenvalue.*

*Proof.* Then by the GEP formulation of CCA, we must have

$$\Sigma^{(12)} u^{(2)} = \rho \Sigma^{(11)} u^{(1)}, \quad \Sigma^{(21)} u^{(1)} = \rho \Sigma^{(22)} u^{(2)}$$

In the exchangeable setting we have $\Sigma^{(11)} = \Sigma^{(22)}, \Sigma^{(12)} = \Sigma^{(21)}$. Write $\Delta = u^{(2)} - u^{(1)}$. Then we can combine the two previous equations to see

$$\Sigma^{(12)}(u^{(2)} - u^{(1)}) = \rho \Sigma^{(11)}(u^{(1)} - u^{(2)})$$

and so

$$\Delta^\top \Sigma^{(12)} \Delta = -\rho \Delta^\top \Sigma^{(11)} \Delta \leq 0$$

Therefore, when $\rho > 0$ and $\Sigma^{(11)}$ is full rank then this is strict inequality and so the cross-covariance matrix must have a negative eigenvalue. $\qquad \square$

Conveniently, under the data generating process commonly used in SSL, this cannot happen!

**Lemma C.9** (Generated by augmentations)**.** *Consider a pair of random vectors $(X^{(1)}, X^{(2)})$ generated via*

$$\begin{aligned} X^{(0)} &\sim P_X \\ g^{(i)} &\sim \mathcal{G} \text{ independently, for } i = 1, 2 \\ X^{(i)} &= g^{(i)}(X^{(0)}) \text{ for } i = 1, 2 \end{aligned} \qquad (27)$$

*then any canonical directions $u_k^{(1)}, u_k^{(2)}$ with $\rho_k > 0$ must satisfy $u_k^{(1)} = u_k^{(2)}$.*

*Moreover, for any $K \leq D^{(1)}$, there exist a full set of CCA weights $(U, U)$ with $U \in \mathbb{R}^{D^{(1)} \times K}$.*

*Proof.* Write $\bar{g}(x) = \mathbb{E}_{g \sim \mathcal{G}}(g(x))$. Then by the tower law

$$
\begin{aligned}
\mathrm{Cov}(X^{(1)}, X^{(2)}) &= \mathbb{E}(X^{(1)}X^{(2)\top}) - \mathbb{E}(X^{(1)})\mathbb{E}(X^{(2)})^\top \\
&= \mathbb{E}\left(\bar{g}(X^{(0)})\bar{g}(X^{(0)})\right) - \mathbb{E}\left(\bar{g}(X^{(0)})\right)\mathbb{E}\left(\bar{g}(X^{(0)})\right)^\top \\
&= \mathrm{Var}\left(\bar{g}(X^{(0)})\right) \succeq 0
\end{aligned}
$$

so is a symmetric positive semi-definite matrix.

Then lemma C.8 immediately implies the first conclusion, that $u_k^{(1)}, u_k^{(2)}$ with $\rho_k > 0$ must satisfy $u_k^{(1)} = u_k^{(2)}$.

For the final conclusion, recall that constructing CCA directions $(u_k^{(1)}, u_k^{(2)})_k$ is equivalent to a singular value decomposition of $T = \mathrm{Var}(X^{(1)})^{-1/2}\mathrm{Cov}(X^{(1)}, X^{(2)})\mathrm{Var}(X^{(2)})^{-1/2}$. Under the generative model we have $\mathrm{Var}(X^{(1)}) = \mathrm{Var}(X^{(2)})$ so in fact the target matrix $T$ is symmetric positive semi-definite. Therefore, if we take an eigen-decomposition of $T$, this will also give a singular value decomposition, and so mapping back through $\mathrm{Var}(X^{(1)})^{-1/2}$ will give a full set of CCA weights $(U, U)$ with $U \in \mathbb{R}^{D^{(1)} \times D^{(1)}}$, as claimed. $\qquad\square$

## C.2   DEEP CCA

### C.2.1   ECKHART-YOUNG LOSS RECOVERS DEEP CCA

*Proof.* Write $f^{(i)}(X^{(i)}; \theta^{(i)}) = U^{(i)\top}g^{(i)}(X^{(i)}; \phi^{(i)})$ where the $U^{(i)}$ are matrices parameterising the final layer and $g^{(i)}$ defines the representations in the penultimate layer.

Because $\hat{\theta}$ is a local minimum of $\mathcal{L}_{\mathrm{EY}}(\theta)$ we must have $\hat{U}$ a local minimum of the map $l : U \mapsto \mathcal{L}_{\mathrm{EY}}((U, \hat{\phi}))$. Writing $\hat{Y} = g(X; \hat{\phi})$ for the corresponding penultimate-layer representations we get

$$
\begin{aligned}
l(U) := \mathcal{L}_{\mathrm{EY}}((U, \hat{\phi})) &= -2\,\mathrm{trace}\left(\sum_{i \neq j}\mathrm{Cov}(U^{(i)\top}\hat{Y}^{(i)}, U^{(j)\top}\hat{Y}^{(j)})\right) + \left\|\sum_i \mathrm{Var}(U^{(i)\top}\hat{Y}^{(i)})\right\|_F^2 \\
&= -2\,\mathrm{trace}\left(U^\top A(\hat{Y})U\right) + \|U^\top B(\hat{Y})U\|_F^2
\end{aligned}
$$

where $A(\hat{Y}), B(\hat{Y})$ are as in eq. (6) with $X$ replaced by $\hat{Y}$. This is precisely our Eckhart-Young loss for linear CCA on the $\hat{Y}$. So by proposition 3.2, $\hat{U}$ must also be a global minimum of $l(U)$ and then by proposition 3.1 the optimal value is precisely $-\|\mathrm{MCCA}_K(\hat{Y})\|_2^2$.

This in turn is equal to $-\|\mathrm{MCCA}_K(\hat{Z})\|_2^2$ by a simple sandwiching argument. Indeed, by proposition 3.1 $\min_V \mathcal{L}_{\mathrm{EY}}((V^{(i)\top}X^{(i)})_i) = -\|\mathrm{MCCA}_K(\hat{Z})\|_2^2$. Then we can chain inequalities

$$
\begin{aligned}
-\|\mathrm{MCCA}_K(\hat{Y})\|_2^2 = \mathcal{L}_{\mathrm{EY}}(\hat{Z}) &\geq \min_V \mathcal{L}_{\mathrm{EY}}((V^{(i)\top}X^{(i)})_i) \\
&\geq \min_U \mathcal{L}_{\mathrm{EY}}((U^{(i)\top}\hat{Y}^{(i)})_i) = -\|\mathrm{MCCA}_K(\hat{Y})\|_2^2
\end{aligned}
$$

to conclude. $\qquad\square$

### C.2.2   DEEP CCA WITH TIED WEIGHTS

Finally we are ready to analyse the Deep CCA with tied weights under this augmented-pairs-of-data assumption.

**Proposition C.10** (Deep CCA tied weights)**.** *Consider a pair of random vectors generated as in eq.* (27)*. Let $(\hat{f}^{(1)}, \hat{f}^{(2)})$ be a pair of functions optimising (a function space version of) our Eckhart-Young loss for population Deep CCA*

$$
(\hat{f}^{(1)}, \hat{f}^{(2)}) \in \underset{f^{(1)}, f^{(2)} \in \mathcal{F}}{\arg\min}\ \mathcal{L}_{EY}\left(f^{(1)}(X^{(1)}), f^{(2)}(X^{(2)})\right) \tag{28}
$$

Assume that $\mathcal{F}$ is a class of functions $f : \mathbb{R}^{D^{(1)}} \mapsto \mathbb{R}^K$ closed under left-composition with linear maps (e.g. $\mathcal{F}$ defined by varying parameters of a neural network with a final linear layer): i.e. $f \in \mathcal{F}, \ O \in \mathbb{R}^{K \times K} \implies O \circ f \in \mathcal{F}$.

Then in fact the Siamese network pairs $(\hat{f}^{(1)}, \hat{f}^{(1)})$ and $(\hat{f}^{(2)}, \hat{f}^{(2)})$ must both also attain that same minimal value. Moreover, there is a constant vector $c \in \mathbb{R}^K$ such that

$$\mathbb{E}_{g \sim \mathcal{G}} \big[ \hat{f}^{(1)}(g(X^{(0)})) \mid X^{(0)} \big] = c + \mathbb{E}_{g \sim \mathcal{G}} \big[ \hat{f}^{(2)}(g(X^{(0)})) \mid X^{(0)} \big] \quad a.s. \tag{29}$$

*Proof.* For the rest of this proof (and only for this proof) define the matrix-valued functions $C : \mathcal{F}^2 \to \mathbb{R}^{K \times K}$ and $V : \mathcal{F} \to \mathbb{R}^{K \times K}$ by

$$C(f^{(1)}, f^{(2)}) = \mathrm{Cov} \left( f^{(1)}(X^{(1)}), f^{(2)}(X^{(2)}) \right), \quad V(f) = \mathrm{Var} \left( f(X^{(1)}) \right)$$

The Eckhart-Young loss can be written in terms of these functions as

$$\mathcal{L}_{\mathrm{EY}}(f^{(1)}, f^{(2)}) = -4 \sum_{k=1}^{K} C_{kk}(f^{(1)}, f^{(2)}) + \|V(f^{(1)}) + V(f^{(2)})\|_F^2$$

Firstly, we investigate how the between-view covariance term transforms under 'tying' the networks for each view, for a general choice of $f^{(1)}, f^{(2)}$ (not necessarily optimisers). We will decompose the covariance terms much like in the proof of lemma C.9. Write

$$\overline{f^{(i)}g}(x) = \mathbb{E}_{g \sim \mathcal{G}} \left( f^{(i)} \circ g(x) \right)$$

Then by Cauchy-Schwarz we have

$$\begin{aligned} C_{kk}(f^{(1)}, f^{(2)}) &= \mathrm{Cov} \left( f^{(1)} \circ g^{(1)}(X^{(0)}), f^{(2)} \circ g^{(2)}(X^{(0)}) \right) \\ &= \mathrm{Cov} \left( \overline{f_k^{(1)}g}(X^{(0)}), \overline{f_k^{(2)}g}(X^{(0)}) \right) \\ &\leq \left\{ \mathrm{Var} \left( \overline{f_k^{(1)}g}(X^{(0)}) \right) \mathrm{Var} \left( \overline{f_k^{(2)}g}(X^{(0)}) \right) \right\}^{1/2} \\ &= \left\{ C_{kk}(f^{(1)}, f^{(1)}) \, C_{kk}(f^{(2)}, f^{(2)}) \right\}^{1/2} \end{aligned} \tag{30}$$

and so a further application of Cauchy-Schwarz, this time on $\mathbb{R}^K$, followed by AM-GM inequality gives

$$\begin{aligned} \sum_k C_{kk}(f^{(1)}, f^{(2)}) &\leq \sum_k C_{kk}(f^{(1)}, f^{(1)})^{1/2} \, C_{kk}(f^{(2)}, f^{(2)})^{1/2} \\ &\leq \left( \sum_k C_{kk}(f^{(1)}, f^{(1)}) \right)^{1/2} \left( \sum_k C_{kk}(f^{(2)}, f^{(2)}) \right)^{1/2} \\ &\leq \frac{1}{2} \left( \sum_k C_{kk}(f^{(1)}, f^{(1)}) + \sum_k C_{kk}(f^{(2)}, f^{(2)}) \right) \end{aligned} \tag{31}$$

Secondly, we investigate the within-view variance terms at the optimal $\hat{f}^{(1)}, \hat{f}^{(2)}$ from the proposition statement. This is where we need to use closure under linear maps. Note that for any matrices $U^{(1)}, U^{(2)} \in \mathbb{R}^{K \times K}$ we have $U^{(1)\top} \hat{f}^{(1)}, U^{(2)\top} \hat{f}^{(2)} \in \mathcal{F}$ and we recover the original $\hat{f}^{(1)}, \hat{f}^{(2)}$ by taking $U^{(1)} = U^{(2)} = I_K$. Because the $\hat{f}^{(i)}$ are optimal in eq. (28) we must have

$$(I_K, I_K) \in \underset{U^{(1)}, U^{(2)} \in \mathbb{R}^{K \times K}}{\arg\min} \mathcal{L}_{\mathrm{EY}}(U^{(1)\top} \hat{f}^{(1)}, U^{(2)\top} \hat{f}^{(2)}).$$

This loss term can be expanded out as

$$-2 \operatorname{trace} U^\top \begin{pmatrix} 0 & C(\hat{f}^{(1)}, \hat{f}^{(2)}) \\ C(\hat{f}^{(2)}, \hat{f}^{(1)}) & 0 \end{pmatrix} U + \left\| U^\top \begin{pmatrix} V(\hat{f}^{(1)}) & 0 \\ 0 & V(\hat{f}^{(2)}) \end{pmatrix} U \right\|_F^2$$

to show that it is precisely the Eckhart-Young loss for linear CCA on the representations $\hat{f}^{(i)}(X^{(i)})$. Therefore, by lemma C.1 we have at the optimum that

$$(I_K^\top V(\hat{f}^{(1)})I_K) = (I_K^\top V(\hat{f}^{(2)})I_K), \quad \text{i.e. } V(\hat{f}^{(1)}) = V(\hat{f}^{(2)}). \tag{32}$$

Finally, we combine these two results, eq. (31) and eq. (32), to conclude:

$$\begin{aligned}
\mathcal{L}_{\text{EY}}(\hat{f}^{(1)}, \hat{f}^{(2)}) &= -4\sum_{k=1}^{K} C_{kk}(\hat{f}^{(1)}, \hat{f}^{(2)}) + \|V(\hat{f}^{(1)}) + V(\hat{f}^{(2)})\|_F^2 \\
&\geq -4 \times \frac{1}{2}\left(\sum_k C_{kk}(\hat{f}^{(1)}, \hat{f}^{(1)}) + \sum_k C_{kk}(\hat{f}^{(2)}, \hat{f}^{(2)})\right) \\
&\qquad + \frac{1}{2}\left(\|2V(\hat{f}^{(1)})\|_F^2 + \|2V(\hat{f}^{(2)})\|_F^2\right) \\
&= \frac{1}{2}\left(\mathcal{L}_{\text{EY}}(\hat{f}^{(1)}, \hat{f}^{(1)}) + \mathcal{L}_{\text{EY}}(\hat{f}^{(2)}, \hat{f}^{(2)})\right)
\end{aligned}$$

Since we $\hat{f}^{(1)}, \hat{f}^{(2)}$ were constructed to minimize $\mathcal{L}_{\text{EY}}$ this immediately implies that $\mathcal{L}_{\text{EY}}(\hat{f}^{(1)}, \hat{f}^{(1)}) = \mathcal{L}_{\text{EY}}(\hat{f}^{(2)}, \hat{f}^{(2)}) = \mathcal{L}_{\text{EY}}(\hat{f}^{(1)}, \hat{f}^{(2)})$ attain the same minimal value.

Moreover, by chasing back through the inequalities, equality implies that there is equality in eq. (30) for each $k \in [K]$. These equalities directly imply eq. (29). $\qquad\square$

**Remark C.11** (Extension to other formulations of Deep CCA). We note that this proof technique yields very similar results for other formulations of Deep CCA. Firstly, it can be applied to VICReg (interpreted as a Deep CCA method, see appendix D.3.3); this is because the VICReg objective can be viewed as a between-view-correlation reward term of the form $\sum_k C_{kk}(f^{(1)}, f^{(2)})$, while the within-view covariance matrices are equal at any global optimum by lemma D.6.

The technique could also be applied to variants of Deep CCA where the within-view covariance matrices are constrained to be identity, but different correlation objectives are used, such as $\sum_k C_{kk}(f^{(1)}, f^{(2)})$ for $p > 1$[6]

---

[6]Such formulations do indeed recover classical CCA in the linear setting by Wells et al. (2024)[Section 5.1].

## D RELATIONSHIP TO VICREG AND BARLOW TWINS

### D.1 INTRODUCTION AND LOSS FUNCTIONS

To compare our formulation to the VICReg and Barlow Twins methods, we synthesise notation from the main text with that from the original works. Consider pairs of random variables $X^{(1)}, X^{(2)}$ which we think of as pairs of augmented input data (e.g. distorted images). Now consider pairs of *embeddings* $Z^{(1)} = f(X^{(1)}), Z^{(2)} = f(X^{(2)})$. Define the covariance matrices

$$C^{(11)} = \text{Cov}(Z^{(1)}), \quad C^{(22)} = \text{Cov}(Z^{(2)}), \quad C^{(12)} = \text{Cov}(Z^{(1)}, Z^{(2)}). \tag{33}$$

Throughout the rest of this section, as in the main text, we use $K$ to denote the dimension of the embeddings (and so also the dimension of the relevant covariance matrices).

Our SSL-EY loss can be conveniently written in this notation as

$$\mathcal{L}_{SSL-EY} = \text{trace}\left(-2(C^{(12)} + C^{(21)}) + (C^{(11)} + C^{(22)})^2\right)$$

$$= -4\sum_{k=1}^{K} C_{kk}^{(12)} + \sum_{i,j=1}^{K} (C_{kl}^{(11)} + C_{kl}^{(22)})^2.$$

It may be interesting to compare this to the formulations of VICReg and Barlow twins below.

The main aim of this appendix is to show that these techniques are equivalent to CCA in the linear case. We present a complete argument for VICReg in D.3 and a partial picture for Barlow twins in D.4. To facilitate this analysis, we first introduce certain notions of decomposition of a pair of weights into the subspace they capture and their component non-orthogonality in subsection D.2.

We next state the loss functions for VICReg and Barlow Twins with this synthesized notation. We warn the reader that throughout this section we use $\mathcal{L}_{VR}, \mathcal{L}_{BT}$ to denote the VICReg and Barlow Twins loss functions, but these may take different arguments depending on what parameterisation we are using; we hope this simplified/overloaded notation will improve clarity.

#### D.1.1 VICREG LOSS

The VICReg loss is often written (Balestriero & LeCun, 2022) as

$$\mathcal{L}_{VR} = \gamma\mathbb{E}\|Z^{(1)} - Z^{(2)}\|^2 + \sum_{i \in \{1,2\}} \left[\alpha\sum_{k=1}^{K}\left(1 - \sqrt{\text{Var}(Z_i^{(i)})}\right)_+ + \beta\sum_{\substack{k,l=1 \\ k \neq l}}^{K} \text{Cov}(Z_i^{(i)}, Z_j^{(i)})^2\right]$$

where $(\cdot)_+ := \max(\cdot, 0)$. All of these quantities can be written in the notation of (33)

$$\mathbb{E}\|Z^{(1)} - Z^{(2)}\|^2 = \text{trace}\left(C^{(11)} + C^{(22)} - 2C^{(12)}\right)$$

$$\text{Var}(Z_k^{(i)}) = C_{kk}^{(ii)}$$

$$\text{Cov}(Z_k^{(i)}, Z_l^{(i)}) = C_{kl}^{(ii)}.$$

So in our unifying notation the VICReg loss becomes

$$\mathcal{L}_{VR} = -2\gamma\sum_{k=1}^{K} C_{kk}^{(12)} + \sum_{i \in \{1,2\}} \left[\beta\|C^{(ii)}\|_F^2 + \sum_{k=1}^{K}\left(\alpha\left(1 - \sqrt{C_{kk}^{(ii)}}\right)_+ - \beta C_{kk}^{(ii)^2} + \gamma C_{kk}^{(ii)}\right)\right]$$

$$= -2\gamma\,\text{trace}(C^{(12)}) + \sum_{i \in \{1,2\}} l_{VR}(C^{(ii)}) \tag{34}$$

where we define $l_{VR} : \mathbb{R}^{K \times K} \to \mathbb{R}$ by the expression in the first line. This extra notation will be helpful in appendix D.3.

We now make some observations. Firstly, like in Deep CCA, this objective only depends on the covariances between $Z^{(1)}, Z^{(2)}$ from (33). Secondly, the first term can be thought of as reward, and the second as penalty. Thirdly, this reward term only depends on the covariance between $Z^{(1)}, Z^{(2)}$, whereas the penalty term only depends on the variance matrices of $Z^{(1)}, Z^{(2)}$ respectively. These observations provide key motivation behind the argument in appendix D.3.

## D.1.2 Barlow Twins loss

The Barlow Twins loss is usually written in the form

$$\mathcal{L}_{\text{BT}}(C) = \sum_{k=1}^{K}(1 - C_{kk})^2 + \beta \sum_{k \neq l} C_{kl}^2$$

where $C = \text{Corr}(Z^{(1)}, Z^{(2)})$ is the cross correlation matrix. Note that this objective is independent of the scale of the columns of $Z^{(1)}, Z^{(2)}$. In particular, for any optimal solution, we may pick an equivalent optimal solution such that each entry of $Z^{(1)}, Z^{(2)}$ has unit variance. We can therefore write a constrained form of the Barlow Twins objective using the covariance matrices of the last two subsections. Namely

$$\mathcal{L}_{\text{BT}} = \sum_{k=1}^{K}(1 - C_{kk}^{(12)})^2 + \beta \sum_{k \neq l} {C_{kl}^{(12)}}^2 + \mathbb{1}_{\{C_{kk}^{(11)} = C_{kk}^{(22)} = 1 \, \forall i = 1, \ldots, K\}} \tag{35}$$

where we use $\mathbb{1}_{\{\}}$ as in the convex optimization literature to give the formal value of $\infty$ when the constraint is not satisfied, and $0$ when the constraint is satisfied.

## D.2 Subspace-orthogonality decomposition

The analysis in the rest of this appendix considers both tied-weight (Siamese) and untied-weight settings. In this subsection, we first consider the broader, untied weight setting, then apply this result to the tied-weight setting.

### D.2.1 Untied weights

Since we shall only work in the linear setting, each method defines linear transformations corresponding to a pair of weight matrices $B^{(1)}, B^{(2)}$, where the embeddings are given by

$$Z^{(i)} = {B^{(i)}}^\top X^{(i)} \quad \text{for } i \in \{1, 2\}$$

We now state three different ways one can re-parameterise the weight matrices $B^{(i)}$ for more convenient analysis; these are all essentially the same, but differ in their treatment of low-rank weight matrices. Our VICReg analysis needs formulation **2**, our Barlow twins analysis needs formulation **3**, while we also state formulation **1** for the sake of completeness.

**Lemma D.1** (CCA basis for subspace). *Suppose that for each $i$ the components of $X^{(i)}$ are linearly independent. Let $B^{(1)}, B^{(2)}$, be an arbitrary set of weights. Define $R^{(i)} = \text{rank}(B^{(1)})$ for $i = 1, 2$. Without loss of generality (WLOG), suppose that $R^{(1)} \leq R^{(2)} =: R$. Then the following three formulations hold:*

**1. Both $T^{(i)}$ of full rank but possibly different heights:** *There exist $U^{(i)} \in \mathbb{R}^{D \times R^{(i)}}, T^{(i)} \in \mathbb{R}^{R^{(i)} \times K}$ for $i = 1, 2$ such that $B^{(i)} = U^{(i)} T^{(i)}$, each $T^{(i)}$ is of full rank $R^{(i)}$, and*

$${U^{(i)}}^\top \text{Var}\left(X^{(i)}\right) U^{(i)} = I_{R^{(i)}} \quad \text{for } i \in \{1, 2\}, \quad {U^{(1)}}^\top \text{Cov}\left(X^{(1)}, X^{(2)}\right) U^{(2)} = \Lambda \tag{36}$$

*where $\Lambda \in \mathbb{R}^{R^{(1)} \times R^{(2)}}$ is a diagonal matrix of canonical correlations for the subspace of transformed variables.*

**2. At least one full rank $T^{(i)}$ and same height:** *There exist matrices $U^{(i)} \in \mathbb{R}^{D \times R}, T^{(i)} \in \mathbb{R}^{R \times K}$ for $i = 1, 2$ such that $T^{(2)}$ is of full rank $R$, $B^{(i)} = U^{(i)} T^{(i)}$ for each $i$, and*

$${U^{(i)}}^\top \text{Var}\left(X^{(i)}\right) U^{(i)} = I_R \quad \text{for } i \in \{1, 2\}, \quad {U^{(1)}}^\top \text{Cov}\left(X^{(1)}, X^{(2)}\right) U^{(2)} = \Lambda \tag{37}$$

*where $\Lambda \in \mathbb{R}^{R \times R}$ is a diagonal matrix (of canonical correlations for some augmented subspace of transformed variables).*

**3. Square $T^{(i)}$ not necessarily full rank:** *There exist matrices $U^{(i)} \in \mathbb{R}^{D \times K}, T^{(i)} \in \mathbb{R}^{K \times K}$ for $i = 1, 2$ such that $B^{(i)} = U^{(i)} T^{(i)}$ for each $i$, and*

$$U^{(i)^\top} \mathrm{Var}\left(X^{(i)}\right) U^{(i)} = I_R \quad \text{for } i \in \{1, 2\}, \quad U^{(1)^\top} \mathrm{Cov}\left(X^{(1)}, X^{(2)}\right) U^{(2)} = \Lambda \qquad (38)$$

*where $\Lambda \in \mathbb{R}^{K \times K}$ is a diagonal matrix (of canonical correlations for some augmented subspace of transformed variables).*

*Proof.* The key technical care here is to deal with possible linear dependence amongst the columns of the $B^{(i)}$ matrices (i.e. when $R^{(i)} < K$).

For each $i$, take a subset $I^{(i)}$ of the indices $[K]$ such that the columns $B^{(i)}_{I^{(i)}}$ are linearly independent.= By linear independence of the components of $X^{(i)}$, the random variables $Z^{(i)}_{I^{(i)}}$ are also linearly independent. We can also write

$$B^{(i)} = B^{(i)}_{I^{(i)}} M^{(i)} \qquad (39)$$

where $M^{(i)} \in \mathbb{R}^{R^{(i)} \times R}$ expresses the columns of $B^{(i)}$ in this column basis.

We now construct 'augmented' weight matrices $\bar{B}^{(i)}$ depending on which case we want to prove.

- For Case 1 (both $T^{(i)}$ of full rank but possibly different heights), do not perform augmentation; define $\bar{B}^{(i)} = B^{(i)}_{I^{(i)}}$

- For Case 2 (at least one full rank $T^{(i)}$ and same height): for each $i$, if $R^{(i)} < R$ then augment via the concatenation

$$\bar{B}^{(i)} = \left(B^{(i)}_{I^{(i)}} \quad \tilde{B}^{(i)}_+\right) \in \mathbb{R}^{D^{(i)} \times R}$$

where $\tilde{B}^{(i)}_+ \in \mathbb{R}^{D^{(i)} \times (R - R^{(i)})}$ are additional columns such that the resulting $R$ transformed variables are linearly independent.

- For Case 3 (square $T^{(i)}$ not necessarily full rank): for each $i$, if $R^{(i)} < K$ then augment via the concatenation

$$\bar{B}^{(i)} = \left(B^{(i)}_{I^{(i)}} \quad \tilde{B}^{(i)}_+\right) \in \mathbb{R}^{D^{(i)} \times K}$$

where $\tilde{B}^{(i)}_+ \in \mathbb{R}^{D^{(i)} \times (K - R^{(i)})}$ are additional columns such that the resulting $K$ transformed variables are linearly independent.

We return to considering the three cases in parallel. Define $\bar{Z}^{(i)} = \bar{B}^{(i)^\top} X^{(i)}$ for $i = 1, 2$. In each case, write $S^{(i)}$ for the dimension of $\bar{Z}^{(i)}$; so $S^{(i)}$ is $R^{(i)}$, $R$, $K$ in the three cases respectively. Perform (linear) CCA on the pair of random vectors $(\bar{Z}^{(1)}, \bar{Z}^{(2)})$. This gives weight matrices $V^{(i)} \in \mathbb{R}^{S^{(i)} \times S^{(i)}}$, and the diagonal matrix of correlations $\Lambda \in \mathbb{R}^{S^{(1)} \times S^{(2)}}$ such that the transformed representations $V^{(i)^\top} \bar{Z}^{(i)}$ have identity within-view-covariance matrices, and have maximal correlation between views, i.e.

$$V^{(i)^\top} \mathrm{Var}\left(\bar{Z}^{(i)}\right) V^{(i)} = I_{S^{(i)}} \text{ for } i \in \{1, 2\}, \quad V^{(1)^\top} \mathrm{Cov}\left(\bar{Z}^{(1)}, \bar{Z}^{(2)}\right) V^{(2)} = \Lambda \qquad (40)$$

The $V^{(i)}$ are (therefore) of full rank, so we can define

$$U^{(i)} := \bar{B}^{(i)} V^{(i)}$$
$$\bar{T}^{(i)} := V^{(i)^{-1}}$$

which gives us

$$\bar{B}^{(i)} = U^{(i)} \bar{T}^{(i)} \qquad (41)$$

And therefore also that $B_{I^{(i)}}^{(i)} = \bar{B}_{:R^{(i)}}^{(i)} = U^{(i)}\bar{T}_{:R^{(i)}}^{(i)}$. Finally, define $T^{(i)} := \bar{T}_{:R^{(i)}}^{(i)} M^{(i)} \in \mathbb{R}^{R \times K}$. Then by substituting in eq. (39) we immediately recover that $B^{(i)} = U^{(i)} T^{(i)}$.

Moreover we have that

$$
\begin{aligned}
U^{(i)^\top} \mathrm{Cov}\left(X^{(i)}, X^{(j)}\right) U^{(j)} &= V^{(i)^\top} \bar{B}^{(i)^\top} \mathrm{Cov}\left(X^{(i)}, X^{(j)}\right) \bar{B}^{(j)} V^{(j)} \\
&= V^{(i)^\top} \mathrm{Cov}\left(\bar{Z}^{(i)}, \bar{Z}^{(j)}\right) V^{(j)}
\end{aligned}
$$

so applying eq. (40) yields the claims of eq. (36), eq. (45) and eq. (38) respectively. $\qquad\square$

**Remark D.2** (Degeneracy). In Case 1, the matrices $U^{(i)}$ are effectively determined by the $B^{(i)}$. Indeed the transformed variables $U_r^{(i)^\top} X = V_r^{(i)^\top} Z_{:R^{(i)}}^{(i)}$ are precisely the canonical variates from applying CCA to the pair of subspaces $\mathrm{span}\{Z^{(1)}\}, \mathrm{span}\{Z^{(2)}\}$. Therefore, by the linear independence of the original variables, any degeneracy corresponds to degeneracy in the CCA solution.

In Case 2 and Case 3, we no longer have uniqueness of $U, \Lambda$, because the choice of augmentation for $Z^{(1)}$ was arbitrary.

### D.2.2 APPLICATION TO SSL LOSS FUNCTIONS

The power of lemma D.1 is that the covariance matrices $C^{(ij)}$ can be written as the following simple functions of $\Lambda$ and the $T^{(i)}$.

$$
C^{(ij)} = T^{(i)^\top} U^{(i)^\top} \mathrm{Cov}\left(X^{(i)}, X^{(j)}\right) U^{(j)} T^{(j)} = \begin{cases} T^{(i)^\top} T^{(i)} & \text{if } i = j \\ T^{(i)^\top} \Lambda T^{(j)} & \text{if } i \neq j \end{cases} \tag{42}
$$

Both our loss functions (34), (35) are functions of the $C^{(ij)}$ so can also be written as (fairly) simple functions of $T, \Lambda$. As in the proof of lemma D.1, let the corresponding $\Lambda$ have dimensions $S^{(1)} \times S^{(2)}$ where $S^{(i)}$ takes value $R^{(i)}, R, K$ in the three cases respectively. To simplify these expressions we introduce the following notation for semi-inner-product-like[7] bi-linear forms with respect to the matrix $\Lambda \in \mathbb{R}^{S^{(1)} \times S^{(2)}}$, which generalise the Euclidean inner product for pairs of vectors $m^{(i)} \in \mathbb{R}^{S^{(i)}}$ and Frobenius inner product for pairs of matrices $M^{(i)} \in \mathbb{R}^{S^{(i)} \times J}$ respectively:

$$
\langle m^{(1)}, m^{(2)} \rangle_\Lambda := m^{(1)^\top} \Lambda m^{(2)}
$$

$$
\langle M^{(1)}, M^{(2)} \rangle_\Lambda := \mathrm{trace}(M^{(1)^\top} \Lambda M^{(2)})
$$

With this notation we obtain the expressions

$$
\bar{\mathcal{L}}_{\mathrm{VR}}(T^{(1)}, T^{(2)}; \Lambda) = -2\gamma \langle T^{(1)}, T^{(2)} \rangle_\Lambda + \sum_{i \in \{1,2\}} l_{\mathrm{VR}}(T^{(i)^\top} T^{(i)}) \tag{43}
$$

$$
\bar{\mathcal{L}}_{\mathrm{BT}}(T^{(1)}, T^{(2)}; \Lambda) = \sum_{k=1}^K \left(1 - \langle T_{\cdot k}^{(1)}, T_{\cdot k}^{(2)} \rangle_\Lambda\right)^2 + \beta \sum_{k \neq l} \langle T_{\cdot k}^{(1)}, T_{\cdot l}^{(2)} \rangle_\Lambda^2 + \mathbb{1}_{\{\|T_{\cdot k}^{(i)}\|^2 = 1 \ k \in [K], i \in [2]\}} \tag{44}
$$

### D.2.3 TIED WEIGHTS

**Lemma D.3** (CCA basis for subspace, tied-weights). *Suppose $X^{(1)}, X^{(2)}$ are generated by the data-generating mechanism from eq. (27). Suppose we have tied (but otherwise arbitrary) weights $B^{(1)} = B^{(2)} = B$ of rank $R \leq K$. Then the following two formulations hold:*

*1. $T$ **of full rank but not necessarily square:** There exist matrices $U \in \mathbb{R}^{D \times R}, T \in \mathbb{R}^{R \times K}$ such that $T$ is of full rank, $B = UT$ and*

$$
U^\top \mathrm{Var}\left(X^{(i)}\right) U = I_R \quad \text{for } i \in \{1, 2\}, \quad U^\top \mathrm{Cov}\left(X^{(1)}, X^{(2)}\right) U = \Lambda \tag{45}
$$

---

[7]An inner product is typically defined as satisfying a positivity assumption, which is not be satisfied if $\Lambda$ is not of full rank.

*where $\Lambda \in \mathbb{R}^{R \times R}$ is a diagonal matrix of canonical correlations.*

**2. $T$ square but not necessarily full rank:** *There exist matrices $U \in \mathbb{R}^{D \times K}, T \in \mathbb{R}^{K \times K}$ such that $B = UT$ and*

$$U^\top \mathrm{Var}\left(X^{(i)}\right) U = I_K \quad \text{for } i \in \{1, 2\}, \quad U^\top \mathrm{Cov}\left(X^{(1)}, X^{(2)}\right) U = \Lambda \tag{46}$$

*where $\Lambda \in \mathbb{R}^{K \times K}$ is a diagonal matrix (of canonical correlations for some augmented subspace of transformed variables).*

*Proof.* We only give a sketched argument here, because the construction is almost identical to the proof of the untied case, lemma D.1 (just drop the superscripts and apply the symmetry).

Take a subset $I \subset [K]$ such that columns $B_I$ are linearly independent. Let the matrix $M$ be such that $B = B_I M$. If in Case 2 ($T$ square but not necessarily full rank) and $R < K$ then augment $B_I$ with an extra column to form full-rank $\bar{B} \in \mathbb{R}^{D^{(1)} \times K}$, otherwise just set $\bar{B} = B_I$.

The key observation is that the random variables $\bar{Z}^{(i)} := \bar{B}^\top X^{(i)}$ also follow a data-generating mechanism of eq. (27), but now with a different set of augmentations - simply defined via $\tilde{g}(X^{(0)}) = B^\top g(X^{(0)})$. Therefore, by lemma C.9, we can pick a symmetric pair of CCA weights $(V, V)$ for $(\bar{Z}^{(1)}, \bar{Z}^{(2)})$.

We can now wrap up loose ends following the proof of lemma D.1. Define $U := \bar{B}V, \bar{T} := V^{-1}, T := \bar{T}_{:R} M$. The argument then concludes by analogy to proof of lemma D.1 $\qquad \square$

In light of this result, we introduce short-hand for the versions of $\bar{\mathcal{L}}_{\mathrm{VR}}, \bar{\mathcal{L}}_{\mathrm{BT}}$ arising from tying the $T$-weights in eq. (43) and eq. (44). Simply write

$$\bar{\mathcal{L}}_{\mathrm{VR}}(T; \Lambda) = \bar{\mathcal{L}}_{\mathrm{VR}}(T, T; \Lambda); \quad \bar{\mathcal{L}}_{\mathrm{BT}}(T; \Lambda) = \bar{\mathcal{L}}_{\mathrm{BT}}(T, T; \Lambda) \tag{47}$$

### D.3 VICREG ANALYSIS

We are now ready to prove that VICReg recovers CCA in the linear setting; we consider both a general case with untied VICReg weights and a special case where the data is generated by i.i.d. augmentations as in eq. (27) and the VICReg weights are tied. In each case, we prove that the subspace of random variables generated by the VICReg representations correspond to a CCA subspace, though this subspace might have dimension strictly less than $K$.

The tied weight case with i.i.d. augmented data becomes straightforward with the decomposition into $T, \Lambda$ from the previous appendix D.2.3. The key is to note that when $T$ is of full rank the reward

$$\mathrm{trace}\, T^\top \Lambda T = \sum_{r=1}^{R} \lambda_r \|T_{r\cdot}\|^2$$

is strictly increasing in each $\lambda_r$. Therefore, the loss is minimized by maximizing each entry of $\Lambda$, and so, by eigenvalue interlacing, we must recover the CCA solution. We give full details in appendix D.3.1.

The untied weight case is more challenging, but reduces to the same computation. We apply Case 2 of lemma D.1 then use a symmetry argument to show that the resulting $T^{(1)}, T^{(2)}$ can be tied, and have the same rank. We give full details in appendix D.3.2.

Then in appendix D.3.3 we address the final two bullet point claims from the main text.

Finally in appendix D.3.4 we present a simple computation to show that one will expect VICReg to collapse (even in this linear case) for a wide range of tuning parameters.

#### D.3.1 TIED WEIGHTS

**Proposition D.4** (VICReg with linear, tied weights recovers CCA under i.i.d. augmentation set-up)**.** *Let $X^{(1)}, X^{(2)}$ be random vectors in $\mathbb{R}^{D^{(1)}}$ generated as in eq. (27), with strictly positive top-$K$*

*canonical correlations. Consider applying VICReg in the linear case with tied weights. Let $\hat{B}$ be a globally optimal weight matrix:*

$$\hat{B} \in \underset{B \in \mathbb{R}^{D^{(1)} \times K}}{\arg\min} \mathcal{L}_{VR}(B^\top X^{(1)}, B^\top X^{(2)}) \tag{48}$$

*Then there is some $R \leq K$, some $T \in \mathbb{R}^{R \times K}$ and (tied pair of) top-$R$ optimal CCA weights $(\hat{U}, \hat{U})$ such that $\hat{B} = \hat{U}\hat{T}$.*

*Proof.* By lemma D.3 there exist $\hat{U} \in \mathbb{R}^{D^{(1)} \times K}, \hat{T} \in \mathbb{R}^{R \times K}$ such that $\hat{B} = \hat{U}\hat{T}$, $\hat{T}$ is of full rank and that eq. (46) holds for the 'hatted' matrices $\hat{U}, \hat{T}, \hat{\Lambda}$.

Now let $(\tilde{U}, \tilde{U})$ be a tied-pair of top-$R$ CCA matrices; construct the corresponding VICReg weights $\tilde{B} := \tilde{U}\hat{T}$. This gives the inequality

$$\mathcal{L}_{\mathrm{VR}}(\hat{B}^\top X^{(1)}, \hat{B}^\top X^{(2)}) = \bar{\mathcal{L}}_{\mathrm{VR}}(\hat{T}, \hat{T}; \hat{\Lambda}) = -2\gamma \sum_{r=1}^{R} \hat{\lambda}_r \|\hat{T}_{r\cdot}\|^2 + 2l_{\mathrm{VR}}(\hat{T}^\top \hat{T})$$

$$\geq -2\gamma \sum_{r=1}^{R} \tilde{\lambda}_r \|\hat{T}_{r\cdot}\|^2 + 2l_{\mathrm{VR}}(\hat{T}^\top \hat{T}) = \bar{\mathcal{L}}_{\mathrm{VR}}(\hat{T}, \hat{T}; \tilde{\Lambda}) = \mathcal{L}_{\mathrm{VR}}(\tilde{B}^\top X^{(1)}, \tilde{B}^\top X^{(2)}) \tag{49}$$

Where inequality eq. (51) follows from CCA interlacing lemma C.6. Moreover, because $\hat{T}$ is full rank, there is equality if and only if $\hat{\lambda}_r = \hat{\lambda}_r$ for all $r \in [R]$; by the equality case of CCA interlacing, only happens when $(\hat{U}, \hat{U})$ define a top-$R$ CCA subspace for $(X^{(1)}, X^{(2)})$. □

### D.3.2 UNTIED WEIGHTS

**Proposition D.5** (VICReg-CCA equivalence). *Let $X^{(1)}, X^{(2)}$ be random vectors in $\mathbb{R}^{D^{(1)}}, \mathbb{R}^{D^{(2)}}$ respectively. We consider VICReg, and CCA in the linear case; i.e. where $Z^{(1)}, Z^{(2)}$ are linear functions of $X^{(1)}, X^{(2)}$. Then the set of optimal subspaces for VICReg corresponds to the set of optimal subspaces for CCA.*

*In particular, for any optimal VICReg weights*

$$\hat{B}^{(1)}, \hat{B}^{(2)} \in \underset{B^{(1)}, B^{(2)} \in \mathbb{R}^{D^{(1)} \times K}, \mathbb{R}^{D^{(2)} \times K}}{\arg\min} \mathcal{L}_{VR}(B^{(1)T} X^{(1)}, B^{(2)T} X^{(2)}) \tag{50}$$

*there there is some $R \leq K$, some $\hat{T} \in \mathbb{R}^{R \times K}$ and top-$R$ optimal CCA weights*

$$\hat{U}^{(1)}, \hat{U}^{(2)} \in \underset{U^{(1)}, U^{(2)} \in \mathbb{R}^{D^{(1)} \times R}, \mathbb{R}^{D^{(2)} \times R}}{\arg\min} \mathcal{L}_{CCA}(U^{(1)T} X^{(1)}, U^{(2)T} X^{(2)})$$

*such that $\hat{B}^{(1)} = \hat{U}^{(1)}\hat{T}, \hat{B}^{(2)} = \hat{U}^{(2)}\hat{T}$.*

*Proof.* Take $\hat{U}^{(i)}, \hat{T}^{(i)}, \hat{\Lambda}$ as in Case 2 of lemma D.1.

By considering alternative sets of weights of the form $B^{(i)} = \hat{U}^{(i)} T^{(i)}$ for arbitrary $T^{(i)} \in \mathbb{R}^{R \times K}$ condition eq. (50) implies

$$\hat{T}^{(1)}, \hat{T}^{(2)} \in \underset{T^{(1)}, T^{(2)} \in \mathbb{R}^{R \times K}}{\arg\min} \bar{\mathcal{L}}_{\mathrm{VR}}(T^{(1)}, T^{(2)}; \hat{\Lambda})$$

Then by applying lemma D.6 (below) to the form of $\bar{\mathcal{L}}_{\mathrm{VR}}$ in (43) shows that $\bar{\mathcal{L}}_{\mathrm{VR}}(\hat{T}^{(1)}, \hat{T}^{(2)}; \hat{\Lambda}) = \bar{\mathcal{L}}_{\mathrm{VR}}(\hat{T}^{(2)}, \hat{T}^{(2)}; \hat{\Lambda})$.

We next construct a corresponding set of VICReg weights spanning an optimal CCA subspace. Let $\tilde{U}^{(i)}$ be a pair of top-$R$ CCA weight matrices, with corresponding $R \times R$ diagonal matrix of

canonical correlations $\tilde{\Lambda}$. Then construct the VICReg weights by $\tilde{B}^{(i)} = \tilde{U}^{(i)}\hat{T}^{(2)}$. This gives the chain of inequalities

$$
\begin{aligned}
\mathcal{L}_{\text{VR}}(\hat{B}^{(1)\top}X^{(1)}, \hat{B}^{(2)\top}X^{(2)}) &= \bar{\mathcal{L}}_{\text{VR}}(\hat{T}^{(2)}, \hat{T}^{(2)}, \hat{\Lambda}) \\
&= -2\gamma \sum_{r=1}^{R} \hat{\lambda}_r \|\hat{T}^{(2)}_{r\cdot}\|^2 + 2l_{\text{VR}}(\hat{T}^{(2)\top}\hat{T}^{(2)}) \\
&\geq -2\gamma \sum_{r=1}^{R} \tilde{\lambda}_r \|\hat{T}^{(2)}_{r\cdot}\|^2 + 2l_{\text{VR}}(\hat{T}^{(2)\top}\hat{T}^{(2)}) \quad\quad (51) \\
&= \bar{\mathcal{L}}_{\text{VR}}(\hat{T}^{(2)}, \hat{T}^{(2)}, \tilde{\Lambda}) \\
&= \mathcal{L}_{\text{VR}}(\tilde{B}^{(1)\top}X^{(1)}, \tilde{B}^{(2)\top}X^{(2)})
\end{aligned}
$$

Where inequality eq. (51) follows from CCA interlacing lemma C.6; moreover, there is equality if and only if $\hat{\lambda}_r = \hat{\lambda}_r$ for all $r \in [R]$, which by the equality case of CCA interlacing, only happens when $\hat{U}^{(i)}$ define a top-$R$ CCA subspace for $(X^{(1)}, X^{(2)})$.

Finally, since the top-$K$ canonical correlations are strictly positive, these equalities imply that $\hat{\lambda}_r > 0$ for all $r \in [R]$, so the 'moreover' claim of lemma D.6 shows us that in fact we have $T^{(1)}_{r\cdot} = T^{(2)}_{r\cdot}$ for all $r$ and therefore that $\hat{T}^{(1)} = \hat{T}^{(2)} = \hat{T}$, as required. $\square$

**Lemma D.6.** *Consider minimizing a loss function of the form*

$$
\mathcal{L}(T^{(1)}, T^{(2)}) = -2\langle T^{(1)}, T^{(2)}\rangle_\Lambda + f(T^{(1)}) + f(T^{(2)})
$$

*over $T^{(1)}, T^{(2)} \in \mathbb{R}^{R\times K}$ where $\Lambda \in \mathbb{R}^{R\times R}$ is diagonal with entries and $f : \mathbb{R}^{R\times K} \to \mathbb{R}$ is some arbitrary function. Let $\hat{T}^{(1)}, \hat{T}^{(2)}$ be a pair of matrices minimizing this loss function. Then we have*

$$
\mathcal{L}(\hat{T}^{(1)}, \hat{T}^{(1)}) = \mathcal{L}(\hat{T}^{(2)}, \hat{T}^{(2)}) = \mathcal{L}(\hat{T}^{(1)}, \hat{T}^{(2)})
$$

*Moreover any such pair of minimisers must satisfy $\hat{T}^{(1)}_{r\cdot} = \hat{T}^{(2)}_{r\cdot}$ for all indices $r$ where $\lambda_r > 0$.*

*Proof.* We show that for any pair $T^{(1)}, T^{(2)}$, $\mathcal{L}(T^{(1)}, T^{(2)}) \geq \min\left(\mathcal{L}(T^{(1)}, T^{(1)}), \mathcal{L}(T^{(2)}, T^{(2)})\right)$. By expanding out the matrix inner product

$$
\begin{aligned}
\mathcal{L}(T^{(1)}, T^{(2)}) &= -2\langle T^{(1)}, T^{(2)}\rangle_\Lambda + f(T^{(1)}) + f(T^{(2)}) \\
&= \|T^{(1)} - T^{(2)}\|^2_\Lambda - \|T^{(1)}\|^2_\Lambda - \|T^{(2)}\|^2_\Lambda + f(T^{(1)}) + f(T^{(2)}) \\
&= \|T^{(1)} - T^{(2)}\|^2_\Lambda + \frac{1}{2}\left(\mathcal{L}(T^{(1)}, T^{(1)}) + \mathcal{L}(T^{(2)}, T^{(2)})\right) \\
&\geq \frac{1}{2}\left(\mathcal{L}(T^{(1)}, T^{(1)}) + \mathcal{L}(T^{(2)}, T^{(2)})\right) \\
&\geq \min\left(\mathcal{L}(T^{(1)}, T^{(1)}), \mathcal{L}(T^{(2)}, T^{(2)})\right)
\end{aligned}
$$

where the final line used that for any $a, b \in \mathbb{R}$, $\frac{1}{2}(a + b) \geq \min(a, b)$. Equality in this final line implies the losses are all the same. Equality in the penultimate line shows that the $r^{\text{th}}$ rows coincide when $\lambda_r > 0$. $\square$

### D.3.3 INTERPRETATION AS DEEP CCA

For each $K \in \mathbb{N}$, define $\Phi_K : [0, 1]^K \to \mathbb{R}$ by

$$
\Phi_K(\lambda) = \min_{T\in\mathbb{R}^{K\times K}} \bar{\mathcal{L}}_{\text{VR}}(T; \text{diag}(\lambda)). \quad\quad (52)
$$

**Lemma D.7** (Minimum is attained). *The minimum in eq. (52) is always attained; in fact, for each given $\lambda$ there exists $\hat{T}$ with columns $\|\hat{T}\|_k \leq 1 \, \forall k \in [K]$ such that $\Phi_K(\lambda) = \bar{\mathcal{L}}_{VR}(\hat{T}; \text{diag}(\lambda))$.*

*Proof.* Define the function $\psi : \mathbb{R}^{K \times K} \to \mathbb{R}^{K \times K}$ mapping an arbitrary $T \in \mathbb{R}^{K \times K}$ to a shrunken copy whose columns all have norm less than or equal to 1 and defined by

$$(\psi(T))_k = \frac{1}{\max(\|T_k\|_2, 1)} T_k$$

Then $\psi(T)$ is contained within the set

$$\mathcal{T} := \left\{ \tilde{T} \in \mathbb{R}^{K \times K} \,\middle|\, \|\tilde{T}_k\|_2 \leq 1 \text{ for } k \in [K] \right\}$$

which is a compact subset of $\mathbb{R}^{K \times K}$ (w.r.t. the natural topology e.g. generated by the Frobenius inner product). And for any $\lambda \in [0,1]^K$, comparing term by term, and writing $\tilde{T} = \psi(T)$ we have

$$\begin{aligned}
\bar{\mathcal{L}}_{\text{VR}}(T; \lambda) &= \gamma \sum_k \|T_k\|_{I-\Lambda}^2 + \beta \sum_{k,l:k \neq l} \langle T_k, T_l \rangle_2^2 + \alpha \left(1 - \|T_k\|_2\right)_+ \\
&\geq \gamma \sum_k \|\tilde{T}_k\|_{I-\Lambda}^2 + \beta \sum_{k,l:k \neq l} \langle \tilde{T}_k, \tilde{T}_l \rangle_2^2 + \alpha \left(1 - \|\tilde{T}_k\|_2\right)_+ \quad (53) \\
&= \bar{\mathcal{L}}_{\text{VR}}(\psi(T); \lambda).
\end{aligned}$$

For any given $\lambda$, because $\bar{\mathcal{L}}_{\text{VR}}(\cdot; \lambda)$ is a continuous function on the compact set $\mathcal{T}$, it attains its minimum on $\mathcal{T}$ at some $\hat{T} \in \mathcal{T}$. But then for any $T \in \mathbb{R}^{K \times K}$, $\bar{\mathcal{L}}_{\text{VR}}(T; \lambda) \geq \bar{\mathcal{L}}_{\text{VR}}(\psi(T); \lambda) \geq \bar{\mathcal{L}}_{\text{VR}}(\hat{T}; \lambda)$ by eq. (53); So $\hat{T}$ is also a minimiser of $\bar{\mathcal{L}}_{\text{VR}}(\cdot; \lambda)$ over the whole domain $\mathbb{R}^{K \times K}$, as claimed. $\qquad \square$

**Proposition D.8** ($\Phi_K$ relates VICReg to CCA). *We have*

1. *$\Phi_K$ is element-wise decreasing in $\lambda$.*

2. *For $X^{(1)}, X^{(2)}$ generated by i.i.d. augmentations (eq. (27)), we have*

$$\min_{B \in \mathbb{R}^{D^{(1)} \times K}} \mathcal{L}_{VR}(B^\top X^{(1)}, B^\top X^{(2)}) = \Phi_K\left(\text{CCA}_K(X^{(1)}, X^{(2)})\right)$$

3. *For general random vectors $X^{(1)} \in \mathbb{R}^{D^{(1)}}, X^{(2)} \in \mathbb{R}^{D^{(2)}}$, we have*

$$\min_{B^{(1)}, B^{(2)} \in \mathbb{R}^{D^{(1)} \times K}, \mathbb{R}^{D^{(2)} \times K}} \mathcal{L}_{VR}(B^{(1)T} X^{(1)}, B^{(2)T} X^{(2)}) = \Phi_K\left(\text{CCA}_K(X^{(1)}, X^{(2)})\right)$$

*Proof.* Let $\lambda \in [0,1]^K$ be fixed, and let $\hat{T}$ be a corresponding minimiser from lemma D.7. Then

1. Take any $k \in [K], \lambda'_k \in (\lambda_k, 1]$ and fill the remaining entries of the vector $\lambda'$ by $\lambda'_l = \lambda_l$ for $l \in [K] \setminus \{k\}$. Then,

$$\Phi_K(\lambda') \leq \bar{\mathcal{L}}_{\text{VR}}(\hat{T}; \lambda') = -2\gamma \sum_{l=1}^L \lambda'_l \|\hat{T}_{l\cdot}\|^2 + 2l_{\text{VR}}(\hat{T}^\top \hat{T})$$

$$\leq -2\gamma \sum_{l=1}^L \lambda_l \|\hat{T}_{l\cdot}\|^2 + 2l_{\text{VR}}(\hat{T}^\top \hat{T}) = \bar{\mathcal{L}}_{\text{VR}}(\hat{T}; \lambda) = \Phi_K(\lambda)$$

   as required.

2. This follows directly from the proof of proposition D.4.

3. This follows directly from the proof of proposition D.5.

$\qquad \square$

**Interpretation:** these results can help us understand the deep case, provided that there is a final linear layer in the representations. We consider the untied case for now, and leave the tied case to the reader. At any global optimum $\hat{\theta}$ with corresponding representations $\hat{Z}^{(i)}$, because there is a final linear layer

$$\mathcal{L}_{\text{VR}}(\hat{Z}^{(1)}, \hat{Z}^{(2)}) = \min_{B^{(1)}, B^{(2)} \in \mathbb{R}^{K \times K}} \mathcal{L}_{\text{VR}}(B^{(1)\top} \hat{Z}^{(1)}, B^{(2)\top} \hat{Z}^{(2)}) = \Phi_K\left(\text{CCA}_K(\hat{Z}^{(1)}, \hat{Z}^{(2)})\right)$$

This is very similar to our result for Deep CCA lemma 3.6 but with $\Phi_K(\cdot)$ in place of $\|\cdot\|_2^2$. Note however, that $\Phi_K$ need not be strictly decreasing in each argument; indeed it will be constant in arguments where the corresponding row $\hat{T}_{l\cdot}$ is zero. So (deep) VICReg may also learn low-rank representations. In the next subsection we will show that this phenomenon is in some sense 'generic'.

### D.3.4 COLLAPSE EXAMPLE

In the previous subsections, we proved that VICReg recovers an optimal CCA subspace, but its dimension $R$ was allowed to be smaller than the target dimension $K$. We now show that it is possible to have $R < K$ for a wide range of choices of the VICReg penalty parameters $\alpha, \beta, \gamma$. We consider a very simple case of learning representations of dimension $K = 2$ that collapse to give representations of rank $R = 1$. By lemma D.6 it is sufficient to consider tied $T^{(1)} = T^{(2)} = T$.

We proceed in two steps. First we require a technical lemma, lemma D.9 to show that the columns of an optimiser 'cannot be too small'. Then we use a quantity from this lemma in proposition D.10 to construct a broad range of parameter values for which there is collapse.

**Lemma D.9.** *Let $\Lambda = \text{diag}(\lambda_1, \lambda_2)$ with $1 > \lambda_1 > \lambda_2 \geq 0$. Consider minimisers*

$$\hat{T} \in \arg\min_{T \in \mathbb{R}^{2 \times 2}} \bar{\mathcal{L}}_{VR}(T; \Lambda)$$

*of the VICReg loss with parameters $\alpha > 0$. Then there exist a constant $\mu = \mu(\alpha, \gamma; \Lambda) > 0$ such that $\|\hat{T}_k\|_2 \geq \mu$ for any minimiser $\hat{T}$.*

*Proof.* **Main idea:** First construct a good $T^*$ of restricted diagonal form. Second, show that any $T$ with a very small column has higher loss than this good choice of $T^*$.

**First** consider optimising the $\bar{\mathcal{L}}_{\text{VR}}$ over $T$ restricted to be of the form

$$T = \begin{pmatrix} m_1 & 0 \\ 0 & m_2 \end{pmatrix} \tag{54}$$

Then for $m_1, m_2 \leq 1$ because the off-diagonal terms of $C^{(ii)}$ are zero the orthogonality penalty is zero and so we have

$$\bar{\mathcal{L}}_{\text{VR}}(T; \Lambda) = 2\gamma\{m_1^2(1 - \lambda_1) + m_2^2(1 - \lambda_2)\} + 2\left(\alpha(1 - m_1) + \alpha(1 - m_2) + 0\right)$$

So write

$$f_k(m_k) := m_k^2 \gamma(1 - \lambda_k) - m_k \alpha \tag{55}$$

to simplify

$$\frac{1}{2}\bar{\mathcal{L}}_{\text{VR}}(T; \Lambda) = 2\alpha + f_1(m_1) + f_2(m_2).$$

Then by completing the square we have

$$f_k(m_k) = \gamma(1 - \lambda_k)\left(m_k - \frac{\alpha}{2\gamma(1 - \lambda_k)}\right)^2 - \frac{\alpha^2}{2\gamma(1 - \lambda_k)}$$

and therefore $f_k$ has the unique minimiser

$$m_k^* := \arg\min_{m_k \in [0,1]} f_k(m_k) = \min\left\{1, \frac{\alpha}{2\gamma(1 - \lambda_k)}\right\} \in \left(0, \frac{\alpha}{\gamma(1 - \lambda_k)}\right)$$

and moreover, at this minimiser certainly

$$f_k(m_k^*) = m_k^* \left( m_k^* - \frac{\alpha}{\gamma(1 - \lambda_k)} \right) \gamma(1 - \lambda_k) < 0$$

Therefore take $T^* = \begin{pmatrix} m_1^* & 0 \\ 0 & m_2^* \end{pmatrix}$ to attain the minimum value $\frac{1}{2}\bar{\mathcal{L}}_{\text{VR}}(T^*; \Lambda) = 2\alpha + f_1(m_1^*) + f_2(m_2^*)$ over diagonal $T$.

**Second**, consider an arbitrary $T$ of form eq. (69). Since the off-diagonal penalty ($\beta$-term) is always non-negative

$$\frac{1}{2}\bar{\mathcal{L}}_{\text{VR}}(T; \Lambda) \geq \sum_{k=1}^{2} \gamma m_k^2 \left( \cos^2 \theta_k (1 - \lambda_1) + \sin^2 \theta_k (1 - \lambda_2) \right) + \sum_{k=1}^{2} \alpha(1 - m_k) + 0$$
$$\geq 2\alpha + f_1(m_1) + f_1(m_2)$$
$$\geq 2\alpha + f_1(m_1^*) + f_1(m_2)$$

Now we can construct

$$\mu(\alpha, \gamma; \Lambda) := \min \left\{ m_1^*, \frac{-f_2(m_2^*)}{\alpha} \right\} \tag{56}$$

then for $m_2 \in (0, \mu)$ we have

$$f_1(m_2) > f_1(\mu) \geq -\alpha\mu \geq f_2(m_2^*)$$

where the first inequality follows because $f_1$ is strictly decreasing on $(0, m_1^*)$, the second follows immediately from the definition of $f_1$ in eq. (55), and the third by construction of $\mu$ in eq. (56).

Finally conclude that

$$\frac{1}{2}\bar{\mathcal{L}}_{\text{VR}}(T; \Lambda) \geq 2\alpha + f_1(m_1^*) + f_1(m_2) > 2\alpha + f_1(m_1^*) + f_2(m_2^*) = \frac{1}{2}\bar{\mathcal{L}}_{\text{VR}}(T^*; \Lambda)$$

so $T$ cannot be a global minimum when $m_2 \in (0, \mu)$. $\qquad\square$

**Proposition D.10.** *Let $\Lambda = \text{diag}(\lambda_1, \lambda_2)$ with $1 > \lambda_1 > \lambda_2 \geq 0$. Consider a minimiser*

$$\hat{T} \in \underset{T \in \mathbb{R}^{2 \times 2}}{\arg\min} \bar{\mathcal{L}}_{VR}(T; \Lambda)$$

*of the VICReg loss with parameters $\alpha, \beta, \gamma > 0$ satisfying*

$$2\beta < \gamma(\lambda_1 - \lambda_2)\mu^2 \tag{57}$$

*where $\mu = \mu(\alpha, \gamma; \Lambda)$ gives a lower bound for the column norms $\|\hat{T}_k\|$ as in eq. (56) above.*

*Then the bottom row of $\hat{T}$ is zero: $\hat{T}_{2k}^{(1)} = 0$ for $k = 1, 2$.*

*Proof.* Any $T \in \mathbb{R}^{2 \times 2}$ can be re-parameterised to the form

$$T = \begin{pmatrix} m_1 \cos \theta_1 & m_2 \cos \theta_2 \\ m_1 \sin \theta_1 & m_2 \sin \theta_2 \end{pmatrix} \tag{58}$$

Then we show that for any such $T$,

$$\bar{\mathcal{L}}_{\text{VR}}(T; \Lambda) \geq \bar{\mathcal{L}}_{\text{VR}}(T'; \Lambda) \text{ where } T' = \begin{pmatrix} m_1 & m_2 \\ 0 & 0 \end{pmatrix}$$

with equality if and only if $\theta_1 = \theta_2 = 0$.

First note that for $T$ of form eq. (69) we have It will be convenient to rewrite the original VICReg loss eq. (34) to separate back out the terms depending on each penalty parameter

$$\mathcal{L}_{\text{VR}} = \gamma \sum_{k=1}^{K} (C_{kk}^{(11)} + C_{kk}^{(22)} - 2C_{kk}^{(12)}) + \sum_{i \in \{1,2\}} \left( \alpha \sum_k (1 - \sqrt{C_{kk}^{(ii)}}) + \beta \sum_{k \neq l} C_{kl}^{(ii)2} \right)$$

which gives

$$\bar{\mathcal{L}}_{\text{VR}}(T;\Lambda) = 2\gamma\left(\langle T,T\rangle_I - \langle T,T\rangle_\Lambda\right) + 2\left(\alpha\sum_k(1-\|T_{\cdot k}\|)_+ + \beta\sum_{k\neq l}\langle T_{\cdot k},T_{\cdot l}\rangle^2\right)$$

$$= 2\left(\gamma\langle T,T\rangle_{I-\Lambda} + \beta\sum_{k\neq l}\langle T_{\cdot k},T_{\cdot l}\rangle^2 + \alpha\sum_k(1-\|T_{\cdot k}\|)_+\right)$$

Indeed, we can simply expand the difference; write $\bar{\lambda}(\theta) = \lambda_1\cos^2\theta + \lambda_2\sin^2\theta$.

$$\frac{1}{2}\left(\bar{\mathcal{L}}_{\text{VR}}(T;\Lambda) - \bar{\mathcal{L}}_{\text{VR}}(T';\Lambda)\right) = \gamma\left(\langle T,T\rangle_{I-\Lambda} - \langle T',T'\rangle_{I-\Lambda}\right) + \beta\sum_{k\neq l}\left(\langle T_{\cdot k},T_{\cdot l}\rangle^2 - \langle T'_{\cdot k},T'_{\cdot l}\rangle^2\right)$$

$$= \gamma\left(\sum_{k=1}^2 m_k^2\left\{(1-\bar{\lambda}(\theta_k)) - (1-\lambda_1)\right\}\right)$$

$$+ 2\beta\left(m_1 m_2\left\{(\cos\theta_1\cos\theta_2 + \sin\theta_2\sin\theta_1)^2 - 1\right\}\right)$$

where the $\alpha$ terms vanish because $\|T_{\cdot k}\| = m_k$ is preserved. Note that the first term is positive, but that the second term is negative. We will show that in the regime of interest, the magnitude of the negative term is small and so the net contribution is positive. Indeed, we further process the terms separately

$$(1-\bar{\lambda}(\theta_k)) - (1-\lambda_1) = \lambda_1 - \bar{\lambda}(\theta_k)$$
$$= (\lambda_1 - \lambda_2)\sin^2\theta_k \geq 0$$

while by standard trigonometric identities

$$1 - (\cos\theta_1\cos\theta_2 + \sin\theta_2\sin\theta_1)^2 = 1 - \cos^2(\theta_1 - \theta_2)$$
$$= \sin^2(\theta_1 - \theta_2)$$
$$= (\sin\theta_1\cos\theta_2 - \sin\theta_2\cos\theta_1)^2$$
$$\leq (\sin\theta_1 + \sin\theta_2)^2$$
$$\leq 2(\sin^2\theta_1 + \sin^2\theta_2)$$

We can now put these inequalities into the previous step and use the fact that $\mu \leq m_1, m_2 \leq 1$ to get

$$\frac{1}{2}\left(\bar{\mathcal{L}}_{\text{VR}}(T;\Lambda) - \bar{\mathcal{L}}_{\text{VR}}(T';\Lambda)\right) \geq 2\gamma\mu^2(\lambda_1-\lambda_2)(\sin^2\theta_1 + \sin^2\theta_2) - 2\beta \times 2(\sin^2\theta_1 + \sin^2\theta_2)$$

$$= 2(\sin^2\theta_1 + \sin^2\theta_2)\left(\gamma\mu^2(\lambda_1-\lambda_2) - 2\beta\right)$$

so indeed, this difference is strictly positive provided $\theta_1, \theta_2$ are not both zero and eq. (57) holds, as required. $\qquad\square$

## D.4   BARLOW TWINS ANALYSIS

### D.4.1   TIED WEIGHTS

We present a single result whose proof is complete apart from an application of conjecture B.5. A key tool for the partial proof is lemma D.12, which may give the reader better geometrical intuition for the Barlow twins loss.

**Conjecture D.11** (Barlow twins tied weights). *Let $X^{(1)}, X^{(2)}$ be random vectors in $\mathbb{R}^{D^{(1)}}$ generated as in eq. (27), with strictly positive top-$K$ canonical correlations. Consider applying Barlow twins in the linear case with tied weights, with weight matrix $B$ such that $Z^{(i)} = B^\top X^{(i)}$ for $i = 1, 2$. Let $\hat{B}$ be a locally optimal weight matrix of rank $R \leq K$ such that $\text{CCA}_R(\hat{B}^\top X^{(1)}, \hat{B}^\top X^{(2)}) > 0$ in each component. Then $\hat{B}$ defines a CCA subspace of rank $R$.*

Note that in this tied-weight setting, $T^{(1)} = T^{(2)} = T$, the Barlow twins loss can be written as

$$\bar{\mathcal{L}}_{\text{BT}}(T; \Lambda) = \sum_{k=1}^{K} \left(1 - \|T_{\cdot k}\|_{\Lambda}^2\right)^2 + \beta \sum_{k \neq l} \langle T_{\cdot k}, T_{\cdot l}\rangle_{\Lambda}^2 + \mathbb{1}_{\{\|T_{\cdot k}\|^2 = 1 \, \forall i = 1, \ldots, K\}} \tag{59}$$

**Lemma D.12.** *Let $1 \leq K \leq S$ be integers and let $\Lambda = \text{diag}(\lambda_1, \ldots, \lambda_S)$ be an $S \times S$ diagonal matrix with elements $\lambda_r \in [0,1] \, \forall r \in [S]$. Let $T \in \mathbb{R}^{S \times K}$ be a stationary point of $l : \mathbb{R}^{S \times K} \to \mathbb{R}, \tilde{T} \mapsto \mathcal{L}_{\text{BT}}(\tilde{T}; \Lambda)$. Then for each $r \in [S]$ we have*

$$\lambda_r \frac{\partial \mathcal{L}_{\text{BT}}(T; \Lambda)}{\partial \lambda_r} = \sum_k -L_k T_{rk}^2 \tag{60}$$

*where*

$$L_k = (1 - C_{kk})C_{kk} - \beta \sum_{l:l \neq k} C_{kl}^2 \ \text{and, as before, } \ C_{kl} = T_k^\top \Lambda T_l \,. \tag{61}$$

*Moreover, if in addition $T$ is a local optimum of $l$ and $1 > \lambda_1 \geq \cdots \geq \lambda_K > 0$, then $L_k > 0$ for all $k \in [K]$.*

*Proof of lemma D.12.* To clean up notation for this proof we will write $T_k = T_{\cdot k}$ (for columns of $T$) and $C_{kk} := C_{kk}^{(12)}$ (drop the superscripts). First, compute

$$\frac{\partial \bar{\mathcal{L}}_{\text{BT}}(T; \Lambda)}{\partial \lambda_r} = \sum_k (C_{kk} - 1)T_{rk}^2 + \beta \sum_{k \neq l} C_{kl} T_{rk} T_{rl} \tag{62}$$

Our proof idea is to use the first order conditions from the Lagrangian formulation of (59) to show that the right hand side of this expression is less than zero.

The Lagrangian corresponding to the constrained program (59) is

$$\bar{\mathcal{L}}_{\text{BT}}(\tilde{T}, \tilde{L}; \Lambda) = \sum_{k=1}^{K} \left(1 - \|\tilde{T}_{\cdot k}\|_{\Lambda}^2\right)^2 + \beta \sum_{k \neq l} \langle \tilde{T}_{\cdot k}, \tilde{T}_{\cdot l}\rangle_{\Lambda}^2 + 2 \sum_{k=1}^{K} \tilde{L}_k(\|\tilde{T}_{\cdot k}\|^2 - 1)$$

where $\tilde{L} \in \mathbb{R}^K$ is the Lagrange multiplier.

Now let $T$ be any stationary point of $\mathcal{L}_{\text{BT}}(\tilde{T}; \Lambda)$[8]. Then this is a stationary point of (59) so there must be some Lagrange multiplier $L$ for which it satisfies the first order conditions

$$0 = \frac{\partial \bar{\mathcal{L}}_{\text{BT}}(T, L; \Lambda)}{\partial T_{\cdot k}} = 4(C_{kk} - 1)\Lambda T_k + 4\beta \sum_{l:l \neq k} C_{kl} \Lambda T_l + 4 L_k T_k$$

Rearranging gives

$$L_k T_k = (1 - C_{kk})\Lambda T_k - \beta \sum_{l:l \neq k} C_{kl} \Lambda T_l \tag{63}$$

We now take inner products of this vector equation with judicious choices of direction.

$$e_r^\top (63): \qquad\qquad L_k T_{rk} = (1 - C_{kk})\lambda_r T_{rk} - \beta \sum_{l:l \neq k} C_{kl} \lambda_r T_{rl} \tag{64}$$

$$T_k^\top (63): \qquad\qquad L_k = (1 - C_{kk})C_{kk} - \beta \sum_{l:l \neq k} C_{kl}^2 \tag{65}$$

Observe that (64) looks a lot like (62) but involves the $L_k$ nuisance parameter, while (65) and might help us control on this nuisance parameter.

---

[8]Note that some optimiser must exist because the objective is continuous and the constraint set is compact.

Plugging (64) into (62) gives

$$\lambda_r \frac{\partial \mathcal{L}_{\mathrm{BT}}}{\partial \lambda_r} = \lambda_r \sum_k \left\{ (C_{kk} - 1)T_{rk}^2 + \beta \sum_{l:l \neq k} C_{kl} T_{rk} T_{rl} \right\} = \sum_k -L_k T_{rk}^2$$

**Argument for the moreover statement - want to show $\hat{L}_k > 0$ for all $k \in [K]$:**

As in the statement of the lemma, assume

$$1 > \lambda_1 \geq \cdots \geq \lambda_K > 0. \tag{66}$$

Fix an arbitrary $k \in [K]$. Write $\mathcal{C} = \mathrm{span}\{e_1, \ldots, e_K\} \cap \mathrm{span}\{\Lambda T_{(-k)}\}^\perp$; this is non-empty because it is the orthogonal complement of a $(K - 1)$-dimensional subspace in a $K$-dimensional space.

If $T_k \in \mathcal{C}$, then $T_k \in \mathrm{span}\{\Lambda T_{(-k)}\}^\perp$ then $C_{lk} = 0$ for all $l \neq k$ and so eq. (65) implies that $L_k = (1 - C_{kk})C_{kk} \geq 0$. Because $T_k \in \mathcal{C} \subset \mathrm{span}\{e_1, \ldots, e_K\}$ we have $\sum_{s=1}^K T_{sk}^2 = 1$. Then condition eq. (66) implies that $C_{kk} = \sum_{s=1}^K \lambda_s T_{sk}^2 \in [\lambda_K, \lambda_1] \subset (0, 1)$ so in fact $L_k > 0$.

Otherwise, we can take a unit vector $p_k \in \mathcal{C} \cap \mathrm{span}\{T_k\}^\perp$. Then $p_k$ is a unit vector in $\mathrm{span}\{e_1, \ldots, e_K\}$ but orthogonal both to $T_k$ and $\mathrm{span}\{\Lambda T_{(-k)}\}$.

Now consider rotating $T_k$ towards $p_k$ by an angle $\theta$, i.e. parameterise a path

$$T_k(\theta) = \cos\theta \, T_k + \sin\theta \, p_k$$

then we can write

$$C_{kk}(\theta) = \cos^2\theta \, T_k^\top \Lambda T_k + 2\sin\theta\cos\theta \, T_k^\top \Lambda p_k + \sin^2\theta \, p_k^\top \Lambda p_k$$
$$C_{kl}(\theta) = \cos\theta \, T_k^\top \Lambda T_l$$

then differentiating these quantities gives

$$\dot{C}_{kk}(\theta) = -2\cos\theta\sin\theta \, C_{kk}(0) + 2\left(-\sin^2\theta + \cos^2\theta\right) T_k^\top \Lambda p_k + 2\cos\theta\sin\theta \, p_k^\top \Lambda p_k$$
$$\dot{C}_{kl}(\theta) = -\sin\theta \, T_k^\top \Lambda T_l$$

In particular, evaluating at $\theta = 0$ gives $\dot{C}_{kl} = 0$ and $\dot{C}_{kk} = 2 \, T_k^\top \Lambda p_k$. Since $T$ is stationary, the derivative of $\mathcal{L}$ along this path must be zero. Plugging in these expressions gives

$$0 = \frac{1}{2}\partial_\theta(\mathcal{L})|_{\theta=0} = -(1 - C_{kk})\dot{C}_{kk} + \sum_{l:l \neq k} C_{kl}\dot{C}_{kl}$$
$$= -2(1 - C_{kk})T_k^\top \Lambda p_k + 0$$

so because $C_{kk} \leq \lambda_1 < 1$ we must in fact have $T_k^\top \Lambda p_k = 0$. This observation is necessary to simplify the following Hessian-like calculations.

For further convenience, from now we work in terms of $s^2 := \sin^2\theta$ rather than $\theta$ itself and also introduce $\delta := T_k^\top \Lambda T_k - p_k^\top \Lambda p_k$ so we can write

$$C_{kk}(\theta) = C_{kk}(0) - \sin^2\theta \left(T_k^\top \Lambda T_k - p_k^\top \Lambda p_k\right) = C_{kk}(0) - s^2\delta$$

which gives

$$\mathcal{L}(\theta) - \mathcal{L}(0) = (1 - C_{kk}(0) + s^2\delta)^2 - (1 - C_{kk}(0))^2 + 2\beta \sum_{l:l \neq k} \cos^2\theta C_{kl}(0)^2 - 2\beta \sum_{l:l \neq k} C_{kl}(0)^2$$
$$= 2s^2\delta(1 - C_{kk}(0)) + s^4\delta^2 - 2s^2\beta \sum_{l:l \neq k} C_{kl}(0)^2$$

In particular, for $T$ to be a local optimum we must have

$$0 \leq \frac{1}{2}\partial_{s^2}\mathcal{L}(s^2) - \mathcal{L}(0)|_{s^2=0}$$

$$= \left\{ \delta(1 - C_{kk}(0)) - \beta \sum_{l:l\neq k} C_{kl}(0)^2 + s^2\delta^2 \right\}|_{s^2=0}$$

$$= C_{kk}(0)(1 - C_{kk}(0)) - \beta \sum_{l:l\neq k} C_{kl}(0)^2 - p_k^\top \Lambda p_k \, (1 - C_{kk}(0))$$

$$= L_k - p_k^\top \Lambda p_k \, (1 - C_{kk}(0))$$

and so indeed, we in fact have

$$L_k \geq p_k^\top \Lambda p_k \, (1 - C_{kk}(0)) \geq 0$$

moreover, when $1 > \lambda_1 \geq \lambda_K > 0$ this inequality becomes strict, as required. $\square$

*Partial proof of conjecture D.11.* First apply Case 1 ($T$ of full rank but not necessarily square) of lemma D.3 to get $\hat{U} \in \mathbb{R}^{D^{(1)} \times R}, \hat{T} \in \mathbb{R}^{R \times K}$ such that $\hat{B} = \hat{U}\hat{T}$ and $\hat{T}$ of full row-rank. Then since $\hat{B}$ is a local minimum, $\hat{T}$ is a stationary point of $l : T \mapsto \bar{\mathcal{L}}_{\mathrm{BT}}(T, \hat{\Lambda})$. So applying lemma D.12 for each $r \in [R]$ we have

$$\hat{\lambda}_r \frac{\partial \mathcal{L}_{\mathrm{BT}}(\hat{T}; \hat{\Lambda})}{\partial \hat{\lambda}_r} = \sum_k -\hat{L}_k \hat{T}_{rk}^2. \tag{67}$$

Now apply Case 2 ($T$ square but not necessarily full rank) of lemma D.3 to get $\hat{U}' \in \mathbb{R}^{D \times D}, \hat{T}' \in \mathbb{R}^{D \times K}$ such that $\hat{B} = \hat{U}'\hat{T}'$. Now $\hat{T}'$ will not be of full row-rank, but $\hat{U}'$ must give a full basis of CCA directions for $(X^{(1)}, X^{(2)})$. This implies that the corresponding $\hat{\Lambda}' \in \mathbb{R}^{D \times D}$ is a diagonal matrix whose diagonal entries are the full-vector of population canonical correlations $\mathrm{CCA}(X^{(1)}, X^{(2)}) \in \mathbb{R}^D$. In particular this means that $1 > \hat{\lambda}'_1 \geq \cdots \geq \hat{\lambda}'_K > 0$.

We now apply lemma D.12 a second time, now with $\hat{T}', \hat{\Lambda}'$. Note that, $\hat{C}_{kl} = \mathrm{Cov}(\hat{B}_k^\top X^{(1)}, \hat{B}_l^\top X^{(2)})$ and is a function of $B$ so must is independent of the choice of decomposition of $\hat{B}$. Then eq. (61) shows that the Lagrange multipliers are identical to those from before, so are precisely $(\hat{L}_k)_k$. Therefore, by applying the final 'moreover' conclusion of lemma D.12 we deduce that $\hat{L}_k > 0$ for all $k \in [K]$.

Applying this into, eq. (67) for each $r \in [R]$ and using the fact $\hat{T}_{r\cdot} \neq 0$ because of $\hat{T}$s full row-rank implies that $\hat{\lambda}_r > 0$. Therefore we also have $\frac{\partial \mathcal{L}_{\mathrm{BT}}(\hat{T}; \hat{\Lambda})}{\partial \hat{\lambda}_r} > 0 \, \forall r \in [R]$.

Suppose, for contradiction, that $\hat{\lambda}_r < \tilde{\lambda}_r$ the true $r^{\mathrm{th}}$ canonical correlation. Then by conjecture B.5, we may construct a (continuous) path $\hat{U}(t)$ for $t \in [0, 1]$ with $\hat{U}(0) = \hat{U}, \hat{U}(t)^\top \mathrm{Var}(X^{(1)})\hat{U}(t) = I_K$ for all $t$, and $\hat{\Lambda}(t) := \hat{U}(t)^\top \mathrm{Cov}(X^{(1)}, X^{(2)})\hat{U}(t)$ such that for all $t > 0$

$$\hat{\lambda}_r(t) > \hat{\lambda}_r(0) \quad \text{and} \quad \hat{\lambda}_s(t) > \hat{\lambda}_s(0) \, \forall s \in [R] \setminus \{r\}$$

Correspondingly define the (continuous) path $\hat{B}(t) = \hat{U}(t)\hat{T}$.

But then

$$\mathcal{L}_{\mathrm{BT}}(\hat{B}(t)) - \mathcal{L}_{\mathrm{BT}}(\hat{B}(0)) = \bar{\mathcal{L}}_{\mathrm{BT}}(\hat{T}, \hat{\Lambda}(t)) - \bar{\mathcal{L}}_{\mathrm{BT}}(\hat{T}, \hat{\Lambda}(0))$$

$$= \sum_s \frac{\partial \mathcal{L}_{\mathrm{BT}}}{\partial \lambda_s}(\hat{\lambda}_s(t) - \hat{\lambda}_s(0)) + o\left(\hat{\lambda}_s(t) - \hat{\lambda}_s(0)\right)$$

and so is strictly negative for sufficiently small $t$. This contradicts local optimality of $\hat{B}$.

Therefore the top-$R$ entries of $\hat{\Lambda}$ must be the top-$R$ canonical correlations, and so by lemma C.6 we must have that $\hat{U}$ defines a CCA subspace, and so $\hat{B}$ must also define a CCA subspace, as claimed. $\square$

### D.4.2 UN-TIED WEIGHTS

For VICReg we saw that the computation for the untied weight case reduced to that of the tied weight case provided that we could prove that minimisers of the corresponding matrix loss $\bar{\mathcal{L}}_{\text{VR}}(\cdot, \cdot; \Lambda)$ were symmetric. It is natural to expect that similarly any minimisers of the Barlow twins matrix loss $\bar{\mathcal{L}}_{\text{BT}}(\cdot, \cdot; \Lambda)$ are symmetric.

We have observed this to be the case in toy simulations (for a wide variety of values of $\Lambda$ and hyper-parameter $\beta$), but are not yet able to give a rigorous proof.

If one can prove this observation, then one can also prove notions of the remaining the bullet points from the main text analogously to the VICReg versions: Barlow twins also recovers CCA subspaces in the general case of untied weights, and the optimal loss is decreasing in correlation signal, so can be interpreted as an algorithm for Deep CCA.

### D.4.3 COLLAPSE EXAMPLE

Finally we give a short computation, analogous to that of appendix D.3.4, to show that collapse is a generic phenomenon, even in a very simple setting.

**Proposition D.13.** *Let* $\Lambda = \text{diag}(\lambda_1, \lambda_2)$ *with* $1 \geq \lambda_1 > \lambda_2 \geq 0$. *Consider minimisers*

$$\hat{T} \in \underset{T \in \mathbb{R}^{2 \times 2}}{\arg\min} \, \bar{\mathcal{L}}_{BT}(T; \Lambda)$$

*of the tied Barlow twins matrix loss with parameter* $\beta > 0$ *satisfying*

$$\beta < \frac{2(1 - \lambda_1)(\lambda_1 - \lambda_2)}{(3\lambda_1 - \lambda_2)(\lambda_1 + \lambda_2)} =: C(\lambda_1, \lambda_2) \tag{68}$$

*Then* $\hat{T} = \begin{pmatrix} \pm 1 & \pm 1 \\ 0 & 0 \end{pmatrix}$ *are the 4 global minimisers.*

*Proof.* Any $T \in \mathbb{R}^{2 \times 2}$ can be parameterised as

$$T = \begin{pmatrix} \cos\theta_1 & \cos\theta_2 \\ \sin\theta_1 & \sin\theta_2 \end{pmatrix} \tag{69}$$

We use the same convenient notation $\bar{\lambda}(\theta) = \lambda_1 \cos^2\theta + \lambda_2 \sin^2\theta$ as in appendix D.3.4. Then the Barlow twins loss becomes

$$\bar{\mathcal{L}}_{\text{BT}}(\theta_1, \theta_2) = \sum_{k=1}^{2} \left(1 - \bar{\lambda}(\theta_k)\right)^2 + 2\beta \left(\lambda_1 \cos\theta_1 \cos\theta_2 + \lambda_2 \sin\theta_1 \sin\theta_2\right)^2 \tag{70}$$

We now compare each of these terms to the corresponding loss when $\theta_1 = \theta_2 = 0$. For convenience, introduce the quantity $\Delta := \lambda_1 - \lambda_2$. First we bound the reward term:

$$\left(1 - \bar{\lambda}(\theta)\right)^2 - (1 - \lambda_1)^2 = \left((1 - \lambda_1) + \Delta \sin^2\theta\right)^2 - (1 - \lambda_1)^2$$
$$= 2\Delta(1 - \lambda_1)\sin^2\theta + (\Delta)^2 \sin^4\theta$$
$$\geq 2\Delta(1 - \lambda_1)\sin^2\theta$$

Giving us

$$\sum_{k=1}^{2} \left(1 - \bar{\lambda}(\theta_k)\right)^2 \geq 2\Delta(1 - \lambda_1)\left(\sin^2\theta_1 + \sin^2\theta_2\right)$$

Next we bound the penalty term. To save space we use the shorthand $c_k = \cos\theta_k, s_k = \sin\theta_k$ for $k = 1, 2$, and introduce the quantity $\gamma := \lambda_2/\lambda_1 \leq 1$.

$$\lambda_1^2 - (\lambda_1 c_1 c_2 + \lambda_2 s_1 s_2)^2 = \lambda_1^2 \left\{ \prod_{k=1}^{2} (c_k^2 + s_k^2)^2 - \left( c_1^2 c_2^2 + 2\gamma c_1 c_2 s_1 s_2 + \gamma^2 s_1^2 s_2^2 \right) \right\}$$

$$= \lambda_1^2 \left\{ c_1^2 s_2^2 + c_2^2 s_1^2 - 2\gamma c_1 c_2 s_1 s_2 + (1 - \gamma^2) s_1^2 s_2^2 \right\}$$

$$= \lambda_1^2 \left\{ (1 - \gamma) \left( c_1^2 s_2^2 + c_2^2 s_2^2 \right) + \gamma (c_1 s_2 - s_1 c_2)^2 + (1 - \gamma^2) s_1^2 s_2^2 \right\}$$

$$\leq \lambda_1^2 \left\{ (1 - \gamma) \left( s_1^2 + s_2^2 \right) + \gamma (s_1 + s_2)^2 + (1 - \gamma^2) \frac{1}{2} \left( s_1^2 + s_2^2 \right) \right\}$$

$$= \lambda_1^2 \left( s_1^2 + s_2^2 \right) \left( (1 - \gamma) + 2\gamma + \frac{1}{2}(1 - \gamma^2) \right)$$

$$= \frac{1}{2} \lambda_1^2 \left( s_1^2 + s_2^2 \right) \left( 3 + 2\gamma - \gamma^2 \right)$$

Finally put these two inequalities into eq. (70) to get

$$\bar{\mathcal{L}}_{\mathrm{BT}}(\theta_1, \theta_2) - \bar{\mathcal{L}}_{\mathrm{BT}}(0, 0) \geq \left( s_1^2 + s_2^2 \right) \left\{ 2\Delta(1 - \lambda_1) - \beta \lambda_1^2 \left( 3 + 2\gamma - \gamma^2 \right) \right\}$$

and so this will be strictly positive whenever either $\sin\theta_k \neq 0$, provided that

$$\beta < \frac{2\Delta(1 - \lambda_1)}{\lambda_1^2 (3 + 2\gamma - \gamma^2)} = \frac{2(\lambda_1 - \lambda_2)(1 - \lambda_1)}{(3\lambda_1 - \lambda_2)(\lambda_1 + \lambda_2)} =: C(\lambda_1, \lambda_2)$$

as claimed. □

# E   FAST UPDATES FOR (MULTIVIEW) STOCHASTIC CCA (AND PLS)

## E.1   BACK-PROPAGATION FOR EMPIRICAL COVARIANCES

To help us analyse the full details of back-propagation in the linear case, we first prove a lemma regarding the gradients of the empirical covariance operator.

**Lemma E.1** (Back-prop for empirical covariance). *Let $e \in \mathbb{R}^M, f \in \mathbb{R}^M$. Then $\widehat{\mathrm{Cov}}(e, f)$ and*

$$\frac{\partial \widehat{\mathrm{Cov}}(e, f)}{\partial e}$$

*can both be computed in $\mathcal{O}(M)$ time.*

*Proof.* Let $1_M \in \mathbb{R}^M$ be a vector of ones and $\mathcal{P}^{\perp}_{1_M} = I_M - \frac{1}{M} 1_M^{\top} 1_M$ be the projection away from this vector, then we can write $\bar{e} = \mathcal{P}^{\perp}_{1_M} e, \bar{f} = \mathcal{P}^{\perp}_{1_M} f$. Moreover, exploiting the identity-plus-low-rank structure of $\mathcal{P}^{\perp}_{1_M}$ allows us to compute these quantities in $\mathcal{O}(M)$ time.

Then by definition

$$\widehat{\mathrm{Cov}}(e, f) = \frac{1}{M-1} \bar{e}^{\top} \bar{f}$$

which is again computable in $\mathcal{O}(M)$ time.

For the backward pass, first note that

$$\frac{\partial \bar{e}}{\partial e} \; : \; \delta e \mapsto \mathcal{P}^{\perp}_{1_M} \delta e$$

So the derivative with respect to $e$ is

$$\frac{\partial \widehat{\mathrm{Cov}}(e, f)}{\partial e} = \frac{1}{M-1} \frac{\partial \bar{e}^{\top} \bar{f}}{\partial e} = \frac{1}{M-1} \left( \frac{\partial \bar{e}}{\partial e} \bar{f} \right) = \frac{1}{M-1} \mathcal{P}^{\perp}_{1_M} \bar{f} = \frac{1}{M-1} \bar{f}$$

because $\bar{f}$ is independent of $e$, and already mean-centred. So all that remains is element-wise division, which again costs $\mathcal{O}(M)$ time. $\qquad\square$

FORWARD PASS

1. **Compute the transformed variables Z:**
$$\mathbf{Z}^{(i)} = U^{(i)} \mathbf{X}^{(i)}, \tag{71}$$
   with a complexity of $\mathcal{O}(MKD)$.

2. **Compute** $\mathrm{trace}\, \hat{C}(\theta)[\mathbf{Z}]$**:** the diagonal elements of $\hat{C}$ are simply
$$\hat{C}_{kk} = \sum_{i \neq j} \widehat{\mathrm{Cov}}(\mathbf{Z}^{(i)}_k, \mathbf{Z}^{(j)}_k)$$
   which each summand can be computed in $\mathcal{O}(M)$ time, so summing over $i, j, k$ gives total complexity of $\mathcal{O}(I^2 K M)$.

3. **Compute** $\hat{V}(\theta)[\mathbf{Z}]$**:** For $\hat{V}_{\alpha}[\mathbf{Z}]$:
$$\hat{V}_{\alpha}(\theta)[\mathbf{Z}] = \sum_i \alpha_i U^{(i)\top} U^{(i)} + (1 - \alpha_i) \widehat{\mathrm{Var}}(\mathbf{Z}^{(i)}),$$
   each $U^{(i)\top} U^{(i)}$ can be computed with a complexity of $\mathcal{O}(D_i K^2)$ and the total cost of evaluating all of these is $\mathcal{O}(K^2 D)$. Each summand in the second term costs $\mathcal{O}(MK^2)$ by lemma E.1 so evaluating the full second term costs $\mathcal{O}(IMK^2)$.

4. **Evaluate** $\hat{\mathcal{L}}_{\mathrm{EY}}[\mathbf{Z}, \mathbf{Z}']$**:**
$$\hat{\mathcal{L}}_{\mathrm{EY}}[\mathbf{Z}, \mathbf{Z}'] = -2 \,\mathrm{trace}\, \hat{C}[\mathbf{Z}] + \langle \hat{V}_{\alpha}[\mathbf{Z}], \hat{V}_{\alpha}[\mathbf{Z}'] \rangle_F. \tag{72}$$
   The dominant complexity here is the $\mathcal{O}(K^2)$ cost of computing the Frobenius inner product.

BACKWARD PASS

1. **Gradient with respect to $\mathbf{Z}^{(i)}$:** Using the chain rule, the gradient will flow back from the final computed value, $\hat{\mathcal{L}}_{\text{EY}}[\mathbf{Z}, \mathbf{Z}']$, through the operations that produced it.

2. **Gradient of** $\operatorname{trace} \hat{C}(\theta)[\mathbf{Z}]$ **with respect to $\mathbf{Z}_k^{(i)}$:** Is precisely

$$\frac{\partial \hat{C}_{kk}}{\partial \mathbf{Z}_k^{(i)}} = \frac{2}{M-1} \sum_{j \neq i} \bar{\mathbf{Z}}_k^{(j)},$$

where $\bar{\mathbf{Z}}_k^{(j)} = \mathcal{P}_{1_M}^{\perp} \bar{\mathbf{Z}}_k^{(j)}$, from lemma E.1 and so can be computed in $\mathcal{O}(IM)$ time.

3. **Gradients of** $\langle \hat{V}_\alpha[\mathbf{Z}], \hat{V}_\alpha[\mathbf{Z}'] \rangle_F$ **with respect to $\mathbf{Z}_k^{(i)}$:** By applying lemma E.1, the gradient of the empirical variance term is

$$\frac{\partial \widehat{\operatorname{Var}}(\mathbf{Z}^{(i)})_{l,l'}}{\partial \mathbf{Z}_k^{(i)}} = \begin{cases} \frac{2}{M-1} \mathbf{Z}_k^{(i)} & \text{if } l = l' = k \\ \frac{1}{M-1} \mathbf{Z}_l^{(i)} & \text{if } l \neq l' = k \\ 0 & \text{otherwise.} \end{cases}$$

and so

$$\frac{\partial \langle \hat{V}_\alpha[\mathbf{Z}], \hat{V}_\alpha[\mathbf{Z}'] \rangle_F}{\partial \mathbf{Z}_k^{(i)}} = \frac{(1-\alpha_i)}{M-1} \left( 2\hat{V}_\alpha[\mathbf{Z}']_{kk} \mathbf{Z}_k^{(i)} + \sum_l (\hat{V}_\alpha[\mathbf{Z}']_{lk} \mathbf{Z}_l^{(i)} + \hat{V}_\alpha[\mathbf{Z}']_{kl} \mathbf{Z}_k^{(i)}) \right)$$

$$= \frac{2(1-\alpha_i)}{M-1} \sum_{l=1}^{K} \hat{V}_\alpha[\mathbf{Z}']_{lk} \mathbf{Z}_l^{(i)}$$

this can be computed in $\mathcal{O}(MK)$ time.

4. **Gradients of $\hat{\mathcal{L}}_{\text{EY}}[\mathbf{Z}, \mathbf{Z}']$ with respect to $\mathbf{Z}_k^{(i)}$:** can therefore be computed for a given $\mathbf{Z}_k^{(i)}$ in $\mathcal{O}\left(M(K+I)\right)$ time and so, adding up over all $i, k$ gives total $\mathcal{O}\left(IM(K+I)\right)$ time.

5. **Gradients of** $\langle \hat{V}_\alpha[\mathbf{Z}], \hat{V}_\alpha[\mathbf{Z}'] \rangle_F$ **with respect to $U_k^{(i)}$:** is similarly

$$\frac{2\alpha_i}{M-1} \sum_{l=1}^{K} (\hat{V}_\alpha[\mathbf{Z}]_{lk} + \hat{V}_\alpha[\mathbf{Z}']_{lk}) U_l^{(i)}$$

so can be computed in $\mathcal{O}\left(D_i K\right)$ time.

6. **Finally compute gradients with respect to $U_k^{(i)}$:** simply have $Z_k^{(i)} = U_k^{(i)\top} \mathbf{X}^{(i)}$ so the final gradients are

$$\frac{\partial \hat{\mathcal{L}}_{\text{EY}}}{\partial U_k^{(i)}} = \left( \frac{\partial \hat{\mathcal{L}}_{\text{EY}}}{\partial \mathbf{Z}_k^{(i)}} \right)^{\top} \mathbf{X}^{(i)} + \frac{\partial \langle \hat{V}_\alpha[\mathbf{Z}], \hat{V}_\alpha[\mathbf{Z}'] \rangle_F}{\partial U_k^{(i)}} \tag{73}$$

so the dominant cost is the $\mathcal{O}\left(MD_i\right)$ multiplication.

Since $D \gg K, M$, the dominant cost each final gradient is $\mathcal{O}\left(MD_i\right)$. Summing up over $i, k$ gives total cost $\mathcal{O}\left(KM \sum D_i\right) = \mathcal{O}(KMD)$, as claimed.

## F   ADDITIONAL STOCHASTIC CCA EXPERIMENTS

### F.1   SUPPLEMENTARY EXPERIMENTS: SPLIT CIFAR

In this supplementary section, we provide additional experimental results on the Split CIFAR dataset, where the left and right halves of CIFAR-10 images are utilized as separate views for canonical correlation analysis. These experiments aim to bolster the findings reported in the main paper on the MediaMill dataset.

**Observations:** As shown in Figure 5, our method CCA-EY demonstrates similar advantages on the Split CIFAR dataset as observed in the main experiments on MediaMill. Specifically, CCA-EY outperforms both $\gamma$-EigenGame and SGHA in terms of PCC across all tested mini-batch sizes. Moreover, Figure 5a shows that CCA-EY converges faster than the baselines when using a mini-batch size of 20. It is important to note that these trends echo the findings in our primary experiments, further confirming the robustness and efficacy of CCA-EY across different datasets and configurations.

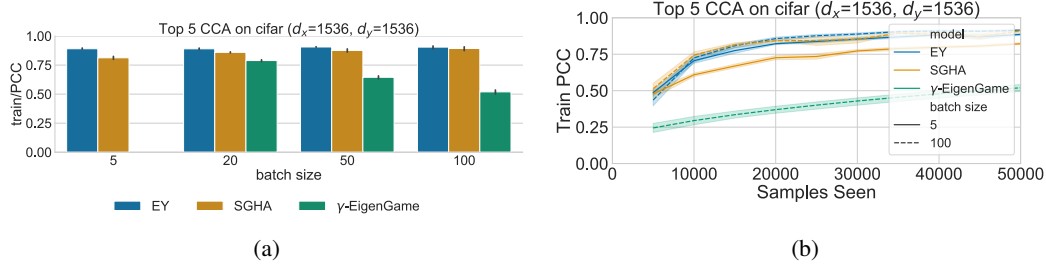

|           | (a)           |           | (b)           |

Figure 5: Experiments onSplit CIFAR with Stochastic CCA: (a) Proportion of Correlation Captured (PCC) across varying mini-batch sizes (left), and (b) Convergence behavior with respect to samples seen for mini-batch size 20 (right). Both subfigures compare CCA-EY against prior methods ($\gamma$-EigenGame and SGHA). Shaded regions signify $\pm$ one standard deviation around the mean.

## G   ADDITIONAL DCCA EXPERIMENTS

In this section, we delve into the performance of DCCA-EY against other DCCA methods. The experimental setup is borrowed from Wang et al. (2015b), utilizing the XRMB dataset. We use mini-batch sizes ranging from 20 to 100 and train the models for 50 epochs. Our metric here is the Total Correlation Captured (TCC), given by TCC $= \sum_{i=1}^{k} \rho_i$.

**Observations:** As depicted in Figure 6, DCCA-STOL shows limitations in scalability, struggling to optimize a 50-dimensional representation when the mini-batch size is less than 50. This is particularly evident in the performance curve for XRMB (Figure 6a). On the other hand, DCCA-NOI performs similarly to DCCA-EY but only for larger mini-batch sizes and with slower speed to convergence.

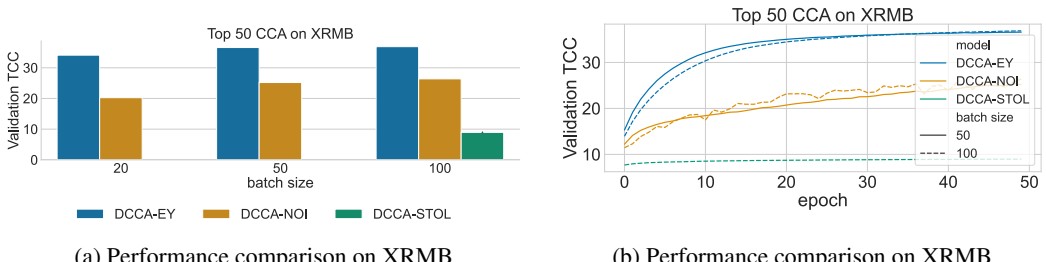

(a) Performance comparison on XRMB          (b) Performance comparison on XRMB

Figure 6: Validation TCC by DCCA-EY vs prior work on the XRMB dataset. Subfigure (a) shows the validation correlation for different batch sizes among various models. Subfigure (b) depicts the validation correlation against the number of epochs for a fixed batch size of 50.

# H  ADDITIONAL SSL EXPERIMENTS

## H.1  JOINT EMBEDDING FOR SSL AND THE ROLE OF THE PROJECTOR

Many recent SSL methods, including Barlow Twins and VICReg use an encoder-projector setup, as illustrated in Figure 7. Input data is mapped through some *encoder g* to obtain representations; these representations are then mapped through a projector[9] $h$ to a (typically) higher-dimensional *embedding*. Crucially, it is the representations that are used for down-stream tasks but the embeddings that are used to train the model. Typically, the encoder is a neural network with domain appropriate architecture, but the projector is a (relatively shallow) multi-layer perceptron.

The idea of joint embedding methods is that similar inputs should have similar embeddings. To train them, one obtains pairs $X, X'$ of similar input data through domain-specific augmentation methods; the encoder and projector then learn to optimise some objective characterizing how close $Z, Z'$ are.

Encoder-projector architectures, have had impressive empirical success, but despite recent work Ma et al. (2023); Jing et al. (2021), there is relatively little understanding of why they work so well. Our more principled objective opens the door for a better understanding of this phenomenon, which may lead to improved future architectures.

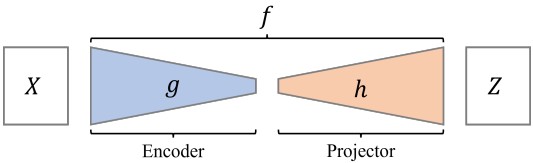

Figure 7: A schematic diagram of the architecture used by Joint Embedding methods which include VICReg, and Barlow Twins

In Figure 8, we demonstrate that our model's performance plateaus at a much smaller projector dimension. This serves as empirical evidence supporting our algorithm as a robust choice for a range of scenarios.

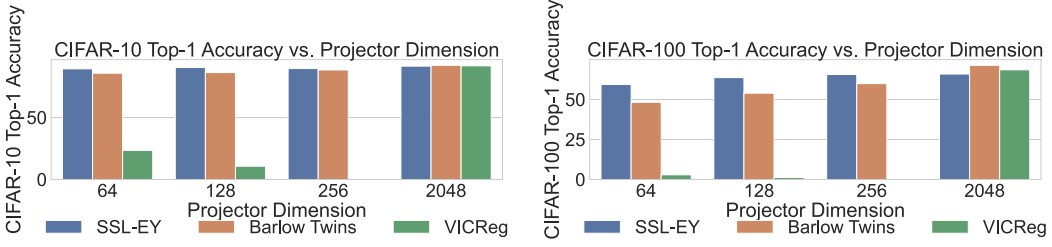

Figure 8: Performance saturation in our model occurs at a much smaller projector size compared to VICReg and Barlow Twins, demonstrating its robustness.

## H.2  UNDERSTANDING LONG-TERM CONVERGENCE

A key insight from our learning curves in Figure 9 and Figure 10 is that the performance variation observed at 1000 epochs is largely a function of noise in early optimization stages. All models seem to follow similar convergence trends, underscoring that the performance differences are not indicative of intrinsic model superiority.

---

[9]Sometimes alternatively called an *expander*.

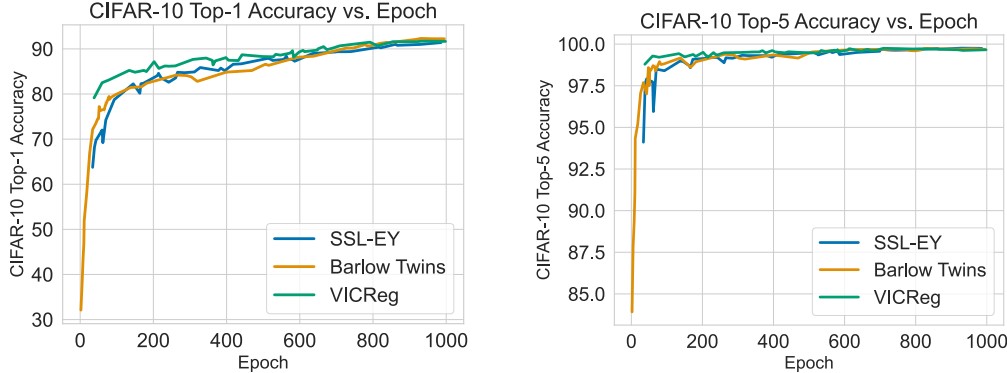

Figure 9: Learning curves for CIFAR-10 showing that the performance of models after 1000 epochs is influenced by noise in early optimization, with all models converging similarly.

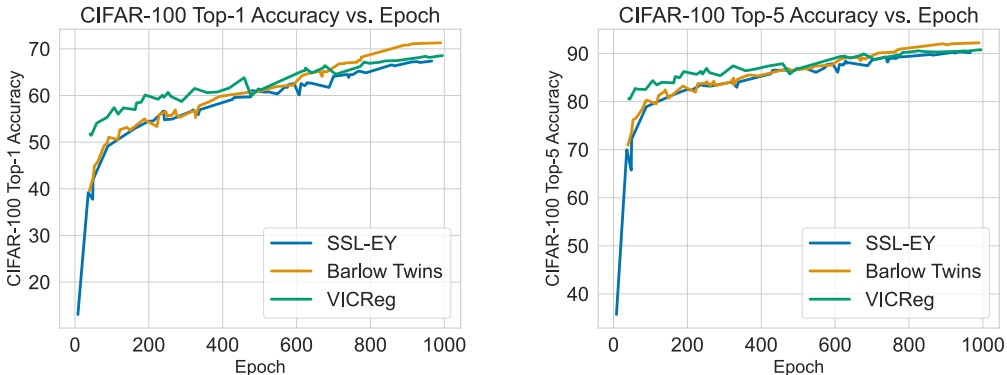

Figure 10: Learning curves for CIFAR-100 emphasizing the role of early optimization noise in the performance after 1000 epochs, highlighting the similar convergence of all models.

# I REPRODUCIBILITY

In this section, we give further detail to allow readers to reproduce the results in the paper.

## I.1 CODE

We make code for the Stochastic CCA and Stochastic Deep CCA experiments available in the attached zip file. We will make this available as a public Github repository.

## I.2 COMPUTER RESOURCES

Each of the four types of experiment required slightly different resources due to the relative scale of the problems.

| Experiment | CPU/GPU Resources |
|---|---|
| Stochastic CCA | NVIDIA GeForce RTX 2080 Ti |
| Deep CCA | NVIDIA GeForce RTX 2080 Ti |
| Deep MCCA | NVIDIA GeForce RTX 2080 Ti |
| Stochastic PLS | NVIDIA GeForce GTX 1650 Ti |
| SSL | 4-8 NVIDIA GeForce RTX 2080 Ti, Quadro RTX 8000 |
| | Quadro RTX 6000, or NVIDIA GeForce GTX 1080 Ti GPU devices |

Table 2: Computer resources for each experiment type

## I.3 FURTHER EXPERIMENT DETAILS

In this section, we give further details regarding the descriptions of the metrics and parameter search.

### I.3.1 STOCHASTIC CCA

**Parameters:** For each method, we searched over a hyperparameter grid using Biewald (2020).

| Parameter | Values |
|---|---|
| minibatch size | 5,20,50,100 |
| components | 5 |
| epochs | 1 |
| seed | 1, 2, 3, 4, 5 |
| lr | 0.01, 0.001, 0.0001 |
| $\gamma^{10}$ | 0.01,0.1,1,10 |

### I.3.2 DEEP CCA

**Further details:** As in Wang et al. (2015b), we used multilayer perceptrons with two hidden layers with size 800 and an output layer of 50 with ReLU activations. We train for 20 epochs.

**Parameters:** For each method, we searched over a hyperparameter grid using Biewald (2020).

| Parameter | Values |
|---|---|
| minibatch size | 100, 50, 20 |
| lr | 1e-3, 1e-4, 1e-5 |
| $\rho^{11}$ | 0.6, 0.8, 0.9 |
| epochs | 50 |

### I.3.3 DEEP MCCA

**Parameters:** For each method, we searched over a hyperparameter grid using Biewald (2020).

| Parameter | Values |
|---|---|
| minibatch size | 5,10,20,50,100,200 |
| components | 50 |
| epochs | 100 |
| lr | 0.01, 0.001, 0.0001, 0.00001 |

### I.3.4   UK BIOBANK PLS

**Partial Least Squares**

Following section 2, we can combine equations 4 and 6 with $\alpha_i = 1 \forall i$ in order to write Partial Least Squares as a Generalized Eigenvalue Problem:

$$A = \begin{pmatrix} 0 & \text{Cov}(X^{(1)}, X^{(2)}) \\ \text{Cov}(X^{(2)}, X^{(1)}) & 0 \end{pmatrix}, \quad B = \begin{pmatrix} I_{D_1} & 0 \\ 0 & I_{D_2} \end{pmatrix}, \quad u = \begin{pmatrix} u^{(1)} \\ u^{(2)} \end{pmatrix}. \tag{74}$$

Note that since $B$ is an Identity matrix by construction, we do not need to make stochastic approximations of $B$ during optimization.

**Further details:** The UK BioBank data consisted of real-valued continuous brain volumes and ordinal, integer genetic variants. We used pre-processed (using FreeSurfer (Fischl, 2012)) grey-matter volumes for 66 cortical (Desikan-Killiany atlas) and 16 subcortical brain regions and 582,565 autosomal genetic variants. The effects of age, age squared, intracranial volume, sex, and the first 20 genetic principal components for population structure were removed from the brain features using linear regression to account for any confounding effects. Each brain ROI was normalized by removing the mean and dividing the standard deviation. We processed the genetics data using PLINK (Purcell et al., 2007) keeping genetic variants with a minor allele frequency of at least 1% and a maximum missingness rate of 2%. We used mean imputation to fill in missing values and centered each variant.

To generate measures of genetic disease risk, we calculated polygenic risk scores using PRSice (Euesden et al., 2014). We calculated scores, with a p-value threshold of 0.05, using GWAS summary statistics for the following diseases; Alzheimer's (Lambert et al., 2013), Schizophrenia (Trubetskoy et al., 2022), Bipolar (Mullins et al., 2021), ADHD (Demontis et al., 2023), ALS (van Rheenen et al., 2021), Parkinson's (Nalls et al., 2019), and Epilepsy (International League Against Epilepsy Consortium on Complex Epilepsies, 2018), using the referenced GWAS studies.

The GEP-EY PLS analysis was trained for 100 epochs using a learning rate of 0.0001 with a minibatch size of 500.

### I.3.5   SELF-SUPERVISED LEARNING

In this section, we provide a comprehensive overview of the experimental settings and configurations used in our self-supervised experiments.

As stated before, we use the standard setup from solo-learn's pretraining scripts. For the backbone network, we use ResNet-18. The projector network consists of hidden dimensions and output dimensions both set to 2048. We employ the LARS optimizer with a learning rate of 0.3 for the backbone and 0.1 for the classifier. The batch size is set to 256, and weight decay is set to $1 \times 10^{-4}$. Additional optimizer parameters include clip_lr set to True, $\eta$ set to 0.02, and exclude_bias_n_norm set to True. The learning rate scheduler used is a warmup cosine scheduler. The models are trained for 1000 epochs. The model's calculations are performed with a numerical precision of 16 bits.

**VICReg and Barlow Twins:** Both models employ similar data augmentations, specified in Tables 3 and 4. In table 3 we show the shared augmentations while in table 4 we show the differences. Note that Barlow Twins uses two different augmentations with 50% probability each.

| Augmentation | Parameters |
|---|---|
| ColorJitter | brightness = 0.4, contrast = 0.4, saturation = 0.2, hue = 0.1, prob = 0.8 |
| Grayscale | prob = 0.2 |
| HorizontalFlip | prob = 0.5 |
| CropSize | 32 |

Table 3: Shared augmentations for VICReg and Barlow Twins

| Augmentation | VICReg | Barlow Twins (crop 1) | Barlow Twins (crop 2) |
|---|---|---|---|
| RandomResizedCrop | Yes | Yes | Yes |
| crop min scale | 0.2 | 0.08 | 0.08 |
| crop max scale | 1.0 | 1.0 | 1.0 |
| Solarization | Yes | No | Yes |
| | prob = 0.1 | prob = 0.0 | prob = 0.2 |
| NumCrops | 2 | 1 | 1 |

Table 4: Different augmentations for VICReg and Barlow Twins

## I.4   PyTorch Pseudo-Code: Unifying the Algorithms under the Generalized Eigenproblem (GEP) Framework

In this work, we introduce three distinct algorithms: DMCCA-EY, PLS-EY, and SSL-EY. Despite their apparent differences, they are all specialized instances of a generalized eigenproblem (GEP). All these algorithms maximize the objective function outlined in Proposition 3.1, making them special cases of our main contribution.

Algorithm 2 gives a general loss for DCCA and DMCCA. Algorithm 3 shows how we can adapt the loss function for stochastic PLS problems. Algorithm 4 gives a generic SSL loss.

**Algorithm 2:** DMCCA-EY Loss Function in Python

```python
def DMCCA_EY(views, views_prime):
    z = encode(views) # Encode the views
    z_prime = encode(views_prime) # Encode the prime views
    A, B, B_prime = torch.zeros(z[0].shape[1], z[0].shape[1]),
     torch.zeros(z[0].shape[1], z[0].shape[1]),
     torch.zeros(z[0].shape[1], z[0].shape[1]) # Initialize
     matrices
    for zi, zj in all_pairs(z):
        A += get_cross_covariance(zi, zj) # Compute
         cross-covariance
        B += get_auto_covariance(zi) # Compute auto-covariance
    for zi in z_prime:
        B_prime += get_auto_covariance(zi) # Compute
         auto-covariance for prime views
    A, B, B_prime = A / len(z), B / len(z), B_prime / len(z_prime)
     # Normalize matrices
    return -torch.trace(2 * A - B @ B_prime) # Calculate loss
```

**Algorithm 3:** PLS-EY Loss Function in Python

```python
def PLS_EY(views):
    z, weights = encode_and_weights(views) # Encode the views and
     get weights
    A, B = torch.zeros(z[0].shape[1], z[0].shape[1]),
     torch.zeros(weights[0].shape[1], weights[0].shape[1]) #
     Initialize matrices
    for zi, zj in all_pairs(z):
        A += cross_covariance_PLS(zi, zj) # Compute
         cross-covariance for PLS
    for wi in weights:
        B += auto_covariance_weights_PLS(wi) # Compute
         auto-covariance for PLS
    A, B = A / len(z), B / len(weights) # Normalize matrices
    return -torch.trace(2 * A - B @ B) # Calculate loss
```

**Algorithm 4:** SSL-EY Loss Function in Python

```python
def SSL_EY(views, views_prime):
    z = encode(views) # Encode the views
    z_prime = encode(views_prime) # Encode the prime views
    A, B, B_prime = torch.zeros(z[0].shape[1], z[0].shape[1]),
     torch.zeros(z[0].shape[1], z[0].shape[1]),
     torch.zeros(z[0].shape[1], z[0].shape[1]) # Initialize
     matrices
    for zi, zj in all_pairs(z):
        A += get_cross_covariance(zi, zj) # Compute
         cross-covariance
        B += get_auto_covariance(zi) # Compute auto-covariance
    for zi in z_prime:
        B_prime += get_auto_covariance(zi) # Compute
         auto-covariance for prime views
    A, B, B_prime = A / len(z), B / len(z), B_prime / len(z_prime)
     # Normalize matrices
```

I.4.1 SOLO-LEARN ADAPTATION

The version of SSL-EY in algorithm 5 is designed to integrate seamlessly into solo-learn, offering support for distributed training.

**Algorithm 5:** Solo-Learn Loss function for distributed SSL-EY in Python

```python
# Define the SSL-EY loss function
# Input:  Projected features from two views
def SSL_EY(z1, z2):
    # Get the minibatch size and feature dimension
    N, D = z1.size()
    # Compute the covariance matrix from the concatenated
     features
    C = torch.cov(torch.hstack((z1, z2)).T)
    # Average the covariance matrix across all processes if
     distributed training is enabled
    if dist.is_available() and dist.is_initialized():
        dist.all_reduce(C)
        world_size = dist.get_world_size()
        C /= world_size
    # Extract symmetric and anti-symmetric blocks of C
    A = C[:D, D:] + C[D:, :D]
    B = C[:D, :D] + C[D:, D:]
    # Return the SSL-EY loss value
    return -torch.trace(2 * A - B @ B)
```

