# OpenReview forum: "Unconstrained Stochastic CCA: Unifying Multiview and Self-Supervised Learning"
_ICLR.cc/2024/Conference — ICLR 2024 poster_

### Official Review · Reviewer_wHCq · 2023-11-01

**Soundness:** 3 good
**Presentation:** 3 good
**Contribution:** 3 good
**Rating:** 8
**Confidence:** 2

**Summary:**

UPDATE:
The authors have done a good job engaging with the reviewers during the discussion period. I have updated my score accordingly to an 8.

Classical algorithms for solving linear problems arising from CCA, PLS, and GEP are computationally infeasible (slow and memory-intensive) for huge datasets. The authors formulate top-subspace GEPs via a new unconstrained objective. The main result is then a family of algorithms that solves stochastic PLS, stochastic CCA and deep CCA by applying SGD to this objective. Empirical results show faster and better convergence than previous methods. For the first time, a very large biomedical dataset is tackled with PLS.

**Strengths:**

- This paper does require a relatively strong background and interest in linear algebra (GEP), optimisation, classical self-supervised/unsupervised techniques (PCA, CCA, PLS), and I have to admit that I found this paper challenging to read. However, when I did check isolated individual details carefully, I was not able to spot any immediately obvious issues. (Soundness +)
- While the material is dense, the paper is clearly laid out and well-written. I particularly like section 2.2 and section 3.2/3.3, which first present general frameworks and then discuss special cases, which is a helpful tutorial-like description. (Presentation +)
- The paper is of general interest to the community, particularly those interested in (self/un)supervised methods. Unifying theories under a single framework is an attractive story for a paper.

Overall I find this paper to be good quality, however I have low confidence in my evaluation. I am receptive to increasing my evaluation and confidence if the authors can provide a good rebuttal and help clarify my queries.

**Weaknesses:**

Clarity:
- I do not see any precise discussion of Barlow Twins or VICReg in the main paper, however Proposition 1 informally states a result related to these formulations. I wonder what is the value in this --- a reader who knows about these formulations might dismiss the informal result (looking at the Appendix instead), and a reader who does not know about these formulations is left completely in the dark. Perhaps section 3.4 could be removed and replaced with a sentence referring to appendix D. (Presentation -)
- I checked the appendix I.3.4 but was not able to properly understand basic properties of dataset UK Biobank. Are the features in this data categorical, integer, real-valued, mixed, ... ? What are the modelling and "convenience" considerations when applying your method? In section 5.4, it is mentioned that "this can reveal novel phenotypes of interest and uncover genetic mechanisms of disease and brain morphometry." What is the precise task that is being performed here, with respect to the PLS formulation (e.g. with reference to the symbols in equations 4, 5, 6, 7)? (Presentation -, and perhaps Soundness -)

Minor and typos:
- It is not clear what is meant by "which should be true in general" in footnote 3. Perhaps if the observations are drawn from a continuous probability distribution, with probability 1 they are linearly independent?
- Check the references. E.g. "stochastic optimization for pca and pls" should be "stochastic optimization for PCA and PLS". Perhaps the authors could also seek advice from the area chair (if the paper is accepted) about the most appropriate way to cite the multiple papers with huge author lists (this might be common in other fields where these papers are published, but in ML it looks really bizarre).

**Questions:**

- I am having trouble with Lemma 3.2. When we assume that there is a final linear layer in each neural net, does this parameter $\theta$ only relate to this final layer, or to the whole neural net? How then is it intuitively possible that $\theta$ could be a local optimum?
- Ridge-regularized CCA. Does the ridge term also change the aforementioned "notions of uniqueness" to a stronger notion?
- "unlike Gemp et al. (2022) we do not simplify the problem by first performing PCA on the data before applying the CCA methods, which explains the decrease in performance of γ-EigenGame compared to their original work. " What then is the motivation for not first simplifying the problem using PCA preprocessing?

---

> ### Author Response · Authors · 2023-11-17
>
> Thank you for such a thorough and insightful review.
> We appreciate your positive feedback with regard to our paper's soundness, clear layout, and relevance to the community.
>
> Thank you in particular for your constructive criticism.
> We found this extremely helpful and propose a number of improvements to the text in response.
> In the following comment, we address each of your main comments in detail, outlining improvements where relevant.
>
> We sincerely hope these resolve any concerns you may have. If so, we kindly ask that you consider raising your score. Regardless of the outcome, we thank you again for such helpful feedback.
>
>
> ### Minor and typos
> - Yes, assuming the observations are from a continuous probability distribution is sufficient to give linear independence with probability 1. We have modified the footnote accordingly.
> - References: we have fixed the capitalisation and will seek advice from the area chair regarding the long author lists.

---

> ### Author Response · Authors · 2023-11-17
>
> ### Lemma 3.2 - equivalence to previous Deep CCA formulations
> Thank you for raising Lemma 3.2.
> On reflection, we agree that the proof was unclear - and have taken the opportunity to rewrite it with more careful notation.
> Unfortunately, this took us over the 9 page limit, so we only give a sketched summary in the main text and give full details appendix C.1 (referenced in the main-text sketched summary).
>
> To directly answer your first question: the parameter $\theta$ relates to the whole neural network.
> To better explain the local-optimum argument we introduce notation: decompose $\theta = (U, \phi)$ into a set of final-layer weights $U$, and a set of weights for the previous layers $\phi$. Then if $\hat{\theta} = (\hat{U}, \hat{\phi})$ is a local optimum of $\mathcal{L}(\theta)$, then $\hat{U}$ is a local optimum of the function $l(U) = \mathcal{L}((U,\hat{\phi}))$.
> Though in practice we would not reach a local optimum in finite time with SGD, one may expect the final estimates to be near a local optimum, and therefore for the conclusions of the lemma to apply at least approximately.
>
> Does that answer your question? (and are you satisfied with our proposed change? or is there more we can do?)
>
> ### Discussion on Barlow Twins and VICReg
> Thank you for raising section 3.4.
> On reflection, we agree that (what was) Proposition 1 could be better integrated with the main text, and that it may be confusing to readers who do not know about Barlow twins and VICReg.
>
> We have therefore taken the opportunity to rewrite the section.
> Our proposals include:
> - Focusing on VICReg - and writing the VICReg loss explicitly. (there was not enough space to state both losses, and we have more complete theoretical analysis for VICReg than Barlow twins).
> - Replacing the 'informal' proposition with a sequence of bullet point conclusions.
> - Brief sentences to better motivate why these results are of interest to the SSL community.
>
> Do you agree that these changes improve the presentation? (and is there more we can do to improve this section?)
>
> ### Uniqueness for CCA
> Your question about uniqueness is a good one - and a detail that we had hoped to 'brush under the carpet' (we wanted to focus on computation, and not get bogged down in regularisation).
> We will give a very short answer here due to space constraints, but hope to expand upon this later in the discussion.
>
> For GEPs with $B$ strictly positive definite there is non-uniqueness up to choice of basis within each generalised eigenspace.
> In high dimensional *sample* CCA there is extreme degeneracy - all canonical correlations are 1 (as discussed in the main text) and there is additional degeneracy in the canonical directions due to the low-rank structure of the sample covariances (the full-batch $\hat{B}$ has non-trivial kernel).
> This degeneracy is overcome by sample ridge CCA by enforcing that $\hat{B}_\alpha$ is strictly positive definite (note however, that the generalised eigenvalues no longer correspond to canonical correlations, and need not be in [0,1], complicating interpretation).
> This phenomenon is analogous to the phenomenon that ridge regression will find unique solutions even in underdetermined systems where OLS gives a degenerate family of solutions (which we suspect motivated your question).
>
> Does this clarify matters?
>
> ### Clarification on the UK Biobank Task
> We have also made some small improvements in response to your question about the Partial Least Squares task - thank you for raising this.
> - *features:* the UK BioBank consists of real-valued continuous brain volumes and ordinal, integer genetic variants. We have added this information to appendix I.3.4.
> - *modelling and "convenience" considerations:* could you please clarify what you are referring to here - we do not believe we used the word "convenience" in this UK Biobank context.
> - *precise task:* it is now standard in Imaging-Genetics research to perform (full sample) CCA or PLS on multi-view datasets, in order to interpret the estimated directions or to use transformed variables for downstream tasks. We use stochastic PLS in this scenario because the dataset is extremely high dimensional and has far more features than samples ($D \gg N$) so un-regularised CCA would break down - moreover, it is more convenient to run a single PLS than to perform tuning parameter selection for ridge CCA.
>   With regard to equations 4, 5, 6, 7: we (hoped to) define PLS through our 'Unified GEP formulation' (equation 7) with $\alpha_i = 1 \, \forall i$, and here there are two views ($I=2$).
>   In this case equation 7 specialises to give the GEP defined by
> $$A = \begin{pmatrix} 0 &\operatorname{Cov}(X^{(1)}, X^{(2)}) \\\\ \operatorname{Cov}(X^{(2)}, X^{(2)}) & 0 \end{pmatrix}, \quad
> 	B = \begin{pmatrix}I_{D_1} & 0 \\\\ 0 & I_{D_2} \end{pmatrix}, \quad
> 	u =\begin{pmatrix}	u^{(1)} \\\\ u^{(2)} \end{pmatrix}.$$
> 	We have also added this equation to appendix I.3.4, with a short discussion.
>
> Does that answer your question?

---

> > ### Comment · Reviewer_wHCq · 2023-11-21
> >
> > - Thanks very much for your clarification around Lemma 3.2 and the local optimum. That's very helpful.
> > - Thanks for your pointers on uniqueness of CCA under ridge regularisation. I understand the complication around the generalised eigenvalues not being in $[0,1]$. I think it might not be necessary to expand upon the paper if you do not feel it is necessary, it was more just a question for myself. If it does fit then great.
> > - Thanks for annotating the UK Biobank dataset and notations. My question about modelling and convenience was about separating two sometimes conflicting requirements in getting an algorithm to run on a dataset: (1) The modelling requirement --- does the statistical (or even predictive) model match a realistic assumption that is likely to hold in practice? And (2) The convenience / tractability consideration --- Will the algorithm run in a reasonable amount of time without linear algebra errors / ill conditioning / other practical considerations. For example in ridge regression, we often add a non-zero ridge term or a larger ridge term than would be reasonable under consideration (1) just so the linear algebra library doesn't complain (2).
> > - I will not have time to go through your updates to the Barlow Twins section, although I do appreciate your efforts.
> > - Thanks for your general response about the PCA preprocessing step. This makes sense to me, and I hope it serves to alleviate Reviewer 73xD's concern.
> >
> > After seeing the updated manuscript, I think this paper should be placed between a 6 and an 8. I will leave my score as is for now and give Reviewer 73xD an opportunity to upgrade their score to a 6. If they don't update their score, I will update my score to an 8, unless any other significant issues are raised during the discussion.

---

> ### Author Response · Authors · 2023-11-22
>
> Thank you once again for your careful review and helpful feedback.
>
> We admit that we had not planned to expand upon the CCA-uniqueness discussion in the paper; but we agree that it would be good to include more details. We will consider how to add appropriate discussion to our Appendix C on CCA background for the camera-ready version.
>
> Thank you also for clarifying your comment regarding the UK Biobank dataset. To attempt to answer your questions:
> 1. **Modelling:** This raises some difficult issues, of a more philosophical nature. Our subjective opinion is that there is not a single statistical model associated with CCA, ridge CCA or PLS, and indeed the methods can be applied without reference to a probabilistic model. That being said, CCA, ridge CCA and PLS can (all) be interpreted as linear Gaussian models - and indeed there are a number of slightly different probabilistic formulations in the literature already, e.g. [1,2,3,4] (incidentally, we hope to unify these in future work).
> These Gaussian models may not be perfectly 'realistic' but they are often reasonable approximations to reality.
> Finally, to redirect this discussion, we would like to reiterate that the main reason we worked on the CCA, ridge-CCA, PLS family is because these methods are already very popular in practice - both due to their simplicity and empirically-established usefulness.
>
> 2. **Tractability:** Yes, our algorithm will run in a reasonable amount of time without linear algebra errors (there are no inverses, simply matrix multiplications) - even for unregularised high-dimensional full-sample CCA where the solutions are degenerate.
>     (Interestingly, our algorithm appears to enjoy a regularising effect from SGD in this degenerate case, and often learns solutions with high out-of-sample correlations; but we had to leave further exploration of this phenomenon for later work.)
>
> To summarise our motivation for the experiment: we wanted to illustrate the computational benefits of our methods on real, high-dimensional UK Biobank data; we had a choice of using any method on the ridge-regularised spectrum between CCA and PLS. Un-regularised full-sample CCA is degenerate, and, though a general ridge CCA would be a reasonable in principle, it could require careful tuning parameter selection (and associated discussion). By focusing on PLS we could avoid tuning parameter selection entirely. An added advantage of PLS is that it is particularly popular in similar biomedical contexts, easing comparison with previous work. We admit that it would also be interesting to compare our PLS analysis to appropriately tuned ridge CCA, and hope to investigate this further.
>
> We realise this response does not closely relate to your original comment about the UK Biobank task, and contains certain subjective opinions, but we hope that it helps clarify the points raised in your most immediate comment. Please let us know if there is anything we can improve upon or clarify further. Thank you again for your help.
>
> [1] M G Gustafsson, "A probabilistic derivation of the partial least-squares algorithm", J Chem Inf Comput Sci.  2001 Mar-Apr;41(2):288-94.
>
> [2] De Bie & De Moor, "On the Regularization of Canonical Correlation Analysis", ICA 2003, https://www.researchgate.net/publication/229057909_On_the_Regularization_of_Canonical_Correlation_Analysis
>
> [3] Bach & Jordan, "A Probabilistic Interpretation of Canonical Correlation Analysis", UC Berkeley technical report, 2005
>
> [4] Said el Bouhaddani, et al. "Probabilistic partial least squares model: Identifiability, estimation and application." arXiv preprint arXiv:2010.00554 (2020).

---

### Official Review · Reviewer_73xD · 2023-11-01

**Soundness:** 3 good
**Presentation:** 3 good
**Contribution:** 2 fair
**Rating:** 5
**Confidence:** 2

**Summary:**

The paper proposes a new unconstrained loss function that introduces solutions to GEPs, paving the way for solving Canonical Correlation Analysis in large scale setting in an efficient manner.

**Strengths:**

The paper conducts extensive experiments across various benchmarks, and proves the effectiveness of the proposed method by the superior empirical results, compared to baselines such as SGHA and $\gamma$-EigenGame.

**Weaknesses:**

I have concern for the novelty of the method compared to Gemp et al. (2022). “unlike Gemp et al. (2022) we do not simplify the problem by first performing PCA on the data before applying the CCA methods “  The author needs to show insights as for the innovation of the proposed method.

**Questions:**

Justification of the novelty of the proposed method.

---

> ### Author Response · Authors · 2023-11-17
>
> Thank you for agreeing that the results of our extensive experiments prove the effectiveness of our proposed method.
> We appreciate this opportunity to address the concerns raised in your review, particularly regarding the novelty of our approach compared to prior work such as [1].
>
> We have addressed your concern about our decision not to perform PCA preprocessing in our common response. We hope that you find this helpful.
>
> In the remainder of this response, we will reiterate the novel aspects of our proposed method, with particular emphasis on comparisons to Gemp et al. (2022) [1], as you requested.
> Comparing directly to [1] we will argue that our work is of much broader scope and that even in the narrow scope where our work is comparable to [1], our method is very different (and in fact has a number of advantages, which we hope to convince you of).
> We will also argue that our work is novel within the literature more broadly.
> By elaborating on these points, we hope to persuade you of the novelty of our contribution.
>
> We sincerely hope that this response successfully addresses your concerns; if so we kindly ask that you consider raising your score. Regardless of the outcome, we thank you again for the time you have taken to review our work.
>
> ## Novelty of our Proposed Methods
>
> Firstly, we address novelty more broadly.
> Our Eckhart-Young inspired characterisation for GEPs is novel - it does not appear elsewhere in the literature (to our knowledge).
> Our presentation of a single framework and class of algorithms to tackle stochastic CCA, PLS and Deep CCA is also novel, as is the application of Deep CCA to SSL.
> Even our stochastic GEP algorithm is very different to all previous proposals - it is the first such algorithm formulated with an explicit loss function that permits unbiased gradient updates.
>
> The work of Gemp et al. (2022) is narrower in that it only proposes algorithms for stochastic GEPs.
> Their algorithms do not have an explicit loss function, and instead use update steps which are inspired by their previous work on an 'Eigengame' for PCA (a very different motivation).
> Crucially, this means that their method cannot be extended to Deep CCA, let alone SSL; these deep extensions were a key focus of our work, and comprised the majority of our experiments (3 out of 5).
>
> Even if we only compare proposals for stochastic GEPs, the $\gamma$-eigengame method of Gemp et al. [1] is very different to our method, and is in fact problematic in a number of ways.
> We reiterate that our method is derived by applying SGD to a population loss function that characterises GEPs, has unbiased update steps and only involves a single learning rate parameter for SGD.
> By contrast $\gamma$-eigengame (Algorithm 2 in [1]):
> - **Does not have an explicit loss function.** Instead motivates update directions by a heuristic argument.
> - **Has biased update steps,** because it involves normalisation by sample quantities.
> - **Has flimsier theoretical support:** they use an error-propagation argument to justify their claim that their algorithm converges with probability 1. However, this proof appears incomplete (red flags include "Assume $\epsilon_i'$ is $\mathcal{O}(\epsilon_i)$ to ease the exposition"). By contrast, we can directly apply results from the literature on SGD to deduce convergence with probability 1.
> - **Most importantly, their method is much more complicated.** It requires "introducing an auxiliary variable, denoted $[B\hat{v}]_j$ to track the running averages of $B\hat{v}_j$" and an additional step to "manually clip the result to be greater than or equal to $\rho$, the minimum singular value of $B$". Their method therefore requires knowledge of a lower bound $\rho$ to the singular values of $B$, which is hard to justify in practice. Also, they require an additional learning rate parameter / step size sequence $\gamma_t$ for updates of $[B\hat{v}]_j$. The performance of their method depends delicately on the choice of this parameter, and tuning it induces a significant computational overhead.
> Finally, we highlight the practical advantages of our method over $\gamma$-eigengame, as illustrated in our experiments.
>
> If anything, our proposal is most closely related to the update rule of SGHA [2]; yet because this cannot be written as the gradient of a population loss function it cannot be easily extended to Deep CCA, and again has flimsier theoretical support.
>
> [1] Gemp, Ian, Charlie Chen, and Brian McWilliams. "The generalized eigenvalue problem as a Nash equilibrium." arXiv preprint arXiv:2206.04993 (2022).
>
> [2] Chen, Zhehui, et al. "On constrained nonconvex stochastic optimization: A case study for generalized eigenvalue decomposition." The 22nd International Conference on Artificial Intelligence and Statistics. PMLR, 2019.

---

> > ### Comment · Reviewer_wHCq · 2023-11-22
> >
> > As we are getting close to the end of the discussion period, I just wanted to check whether Reviewer 73xD has had a chance to look over the rebuttals and see whether their opinion of the paper has changed?

---

### Official Review · Reviewer_NeR6 · 2023-11-01

**Soundness:** 4 excellent
**Presentation:** 3 good
**Contribution:** 4 excellent
**Rating:** 8
**Confidence:** 3

**Summary:**

This paper proposes a new formulation for general CCA problems, based on an Eckhart-Young inspired objective for the Generalized Eigenvalues Problem. Such objective is prone to being implemented in a stochastic way (i.e. with minibatches of dataset samples), and can be used for various variant of CCA such as the original linear CCA, PLS and Ridge Regularized CCA, Multi-view CCA, as well as Deep CCA. The theoretical advantages of such formulation are given, and an extensive experimental benchmark is performed, showing the advantage of the introduced method on several tasks, over state of the art methods.

**Strengths:**

### Originality

I believe the work is original, as it is, up to my knowledge, the first of its kind to consider an Eckhart-Young inspired objective (and the corresponding stochastic optimization algorithm to optimize it) for CCA-type problems.

### Quality

I believe the work is of quality, with theoretical results clearly described as well as their link the the related literature, and experimental results performed extensively.


### Clarity

I believe the work is clear, with a clear description of the methods and of the state of the art. I believe the experiments are clearly described and the code provided eases the reproducibility of the methods.


### Significance


I believe the work is significant, since CCA-like methods and the related Self-Supervised learning methods have been shown to be very important these last years, in particular for modern machine learning applications, including those which use deep neural networks to learn data representations: offering a theoretically grounded view of such problems, as well as an efficient and still theoretically grounded implementation (namely, the stochastic algorithm), I believe this type of methods is very interesting to the community.

**Weaknesses:**

Although the literature is extensively cited across the paper, and well detailed, and although the experiments show a clear advantage to the presented method, I still find it a little bit hard to compare the theoretical and computational guarantees of the presented method vs. the other methods from the benchmark: I believe it might be useful, (for instance in Appendix), to provide some short structured summary (perhaps as a table for instance), on all the properties of the related methods in the literature vs. the one presented (e.g. biasedness/unbiasedness of the gradients, complexity of computing one iteration as a function of the batch-size, dimension, etc, guarantees of convergence towards global/local optimum/saddle point, etc). However, I am aware that such comparison may not always be doable (in particular because the guarantees of convergence for instance, are not always simple to express), so this is just a suggestion,  andI believe the paper in its current form may already provide the necessary information to the reader to study such aspects.

**Questions:**

I just noticed one typo below:
- In section 3.3: “we this flesh out” —> “we flesh this out”

---

> ### Author Response · Authors · 2023-11-17
>
> We sincerely appreciate your positive feedback on our work.
> Thank you for acknowledging its originality, quality, clarity, and significance, especially in the context of CCA-type problems and self-supervised learning methods.
>
> Thank you also for your constructive criticism.
> We have corrected the typo in section 3.3.
> We agree with you that it would be valuable to have a clearer comparison of properties of the presented methods with the other methods in our benchmarks.
> You are right in that it is not straightforward to decide what information is most relevant to display.
> We are still working on this, and hope to include such a summary in the appendix shortly.
>
> We hope that our detailed responses to the other reviews will reinforce your confidence in our work.

---

> > ### Comment · Reviewer_NeR6 · 2023-11-22
> > **Response to Authors**
> >
> > Dear Authors, I would like to thank you for your rebuttal. I have read your response, as well as the other reviews and responses, and I have decided to increase my confidence score.

---

> > > ### Author Response · Authors · 2023-11-22
> > >
> > > Thank you for the increased confidence score. We greatly appreciate your feedback and support.

---

### Author Response · Authors · 2023-11-17
**Common Response**

Thank you to all of the reviewers for your careful and thoughtful reviews of our submission.
We are pleased that reviewers noted our extensive experiments across a number of benchmarks.

We will use this common response to address the concern raised by two of the reviews regarding PCA pre-processing.
We will then address remaining specific concerns in the following individual responses to each reviewer.

## Why we do not perform PCA preprocessing

> we do not simplify the problem by first performing PCA on the data before applying the CCA methods

The first part of the answer is that performing full-sample PCA is computationally infeasible in our high-dimensional scenarios of interest.
Full sample PCA on data with $N$ observations in dimension $D$ has complexity $O(ND^2 + D^3)$ [1], the same scaling as for full-sample CCA.
Our motivation for developing stochastic versions of CCA was to allow application to precisely the scenarios where this complexity is infeasible.
Though one could find an approximate solution to a top-$K$ PCA subspace more efficiently (e.g. by applying our GEP formulation to PCA), this would obscure the interpretation of the experiment.

The second part of the answer is that first performing full sample PCA greatly simplifies the geometry of the CCA problem.
In fact, one of the main challenges in stochastic CCA is to maintaining orthogonality of the canonical directions with respect to the within-view covariance matrices; this corresponds to ensuring estimated generalised eigenvectors are orthogonal with respect to the matrix $B$ in the GEP formulation.
This is particularly hard in the mini-batch setting because the population version of $B$ is unknown, so it must be estimated from the mini-batches.
Preprocessing data with full-sample PCA alters this dynamic, transforming the GEP into a standard eigenvalue problem by ensuring that the full-sample version of $B$ is an identity matrix.
The experiment in [2] (and a similar experiment in [3]) therefore removes an essential part of the challenge of stochastic CCA.
Indeed, the difference in performance of the $\gamma$-eigengame algorithm between our experiments and the previous experiments in [2] illustrates just how important this change of geometry can be.

[1] Gemp, Ian, et al. "Eigengame: PCA as a nash equilibrium." arXiv preprint arXiv:2010.00554 (2020).

[2] Gemp, Ian, Charlie Chen, and Brian McWilliams. "The generalized eigenvalue problem as a Nash equilibrium." arXiv preprint arXiv:2206.04993 (2022).

[3] Meng, Zihang, Rudrasis Chakraborty, and Vikas Singh. "An online riemannian pca for stochastic canonical correlation analysis." Advances in neural information processing systems 34 (2021): 14056-14068.

---

### Meta-Review · Area_Chair_rwFW · 2023-12-07

**Metareview:**

This paper presents a method for CCA-family objectives, which were typically solved with an eigenvalue approach, with the alternative low-rank (Eckhart-Young) approach. In my opinion, the proposed approach loses a little the convex structure in the eigenvalue approach. The iterative eigenvalue solver boils down to solving alternating least squares (a sequence of convex problems in the linear case), whereas the propose approach involves 4-th order terms which are not jointly convex. Nonetheless, the current paper provides a rigorous treatment of the low-rank decomposition approach with extensive and pretty solid experimental results, and is worth publication.

**Justification For Why Not Higher Score:**

The low-rank decomposition approach involving 4-th order terms was done for PCA (although the extension in the current paper is still non-trivial).

**Justification For Why Not Lower Score:**

The paper solves a pretty technical problem in a way not considered before, and obtains good results.

---

### Decision · Program_Chairs · 2024-01-16

Accept (poster)